# FUNDAMENTAL LIMITS OF PROMPT TUNING TRANSFORMERS: UNIVERSALITY, CAPACITY AND EFFICIENCY

**Jerry Yao-Chieh Hu**[*†]    **Wei-Po Wang**[*‡]    **Ammar Gilani**[†]

**Chenyang Li**[♯]    **Zhao Song**[§]    **Han Liu**[†]

[†]Northwestern University    [‡]National Taiwan University    [♯]Fuzhou University    [§]UC Berkeley

{jhu,ammargilani2024}@u.northwestern.edu, b09202009@ntu.edu.tw
{lchenyang550,magic.linuxkde}@gmail.com, hanliu@northwestern.edu

## ABSTRACT

We investigate the statistical and computational limits of prompt tuning for transformer-based foundation models. Our key contributions are prompt tuning on *single-head* transformers with only a *single* self-attention layer: (i) is universal, and (ii) supports efficient (even almost-linear time) algorithms under the Strong Exponential Time Hypothesis (SETH). Statistically, we prove that prompt tuning on such simplest possible transformers are universal approximators for sequence-to-sequence Lipschitz functions. In addition, we provide an exponential-in-$dL$ and -in-$(1/\epsilon)$ lower bound on the required soft-prompt tokens for prompt tuning to memorize any dataset with 1-layer, 1-head transformers. Computationally, we identify a phase transition in the efficiency of prompt tuning, determined by the norm of the *soft-prompt-induced* keys and queries, and provide an upper bound criterion. Beyond this criterion, no sub-quadratic (efficient) algorithm for prompt tuning exists under SETH. Within this criterion, we showcase our theory by proving the existence of almost-linear time prompt tuning inference algorithms. These fundamental limits provide important necessary conditions for designing expressive and efficient prompt tuning methods for practitioners.

## 1 INTRODUCTION

We investigate the statistical and computational limits of prompt tuning for transformer-based foundation models. These models are gigantic transformer-based architectures (Bommasani et al., 2021), pretrained on vast datasets, are pivotal across multiple fields (Touvron et al., 2023b;a; Brown et al., 2020; Floridi and Chiriatti, 2020; Yang et al., 2023; Wu et al., 2023; Nguyen et al., 2024; Zhou et al., 2025; 2024; 2023; Ji et al., 2021; Thirunavukarasu et al., 2023; Singhal et al., 2023; Moor et al., 2023). Despite their power, the significant cost of pretraining these models often makes them prohibitive outside certain industrial labs. Thus, most practitioners resort to fine-tuning methods to tailor these models to specific needs (Zheng et al., 2024; Ding et al., 2022). However, fine-tuning large models with billions or trillions of parameters is still often resource-intensive (Minaee et al., 2024). Prompt tuning mitigates this by adapting a learnable prompt with a limited set of parameters (tokens), preserving the pretrained model weights and allowing adaptation to new tasks or data without any retraining (Lester et al., 2021; Liu et al., 2021). It saves substantial computational resources and time. However, despite its empirical successes (Gao et al., 2024; Shi and Lipani, 2024; Fu et al., 2024; Chen et al., 2023; Wang et al., 2023b; Khattak et al., 2023; Jia et al., 2022; Liu et al., 2022; 2021), the theoretical aspects of prompt tuning are still underexplored, relatively (Wang et al., 2023a; Petrov et al., 2024). This work provides a timely theoretical analysis of the statistical and computational limits of prompt tuning, aiming to explain its successes and offer principled guidance for future prompt tuning methods in terms of performance and computational cost.

Let $X, Y \in \mathbb{R}^{d \times L}$ be the input and the corresponding label sequences, respectively. For $i \in [L]$, we denote $X_{:,i} \in \mathbb{R}^d$ as the $i$-th token (column) of $X$. Let $[\cdot, \cdot]$ denote sequential concatenation.

---

[*]Equal contribution. Full version and future updates are on arXiv. Throughout this work, *universality* or *universal approximation* refers to sequence-to-sequence (seq2seq) universal approximation.

**Definition 1.1** (Prompt Tuning). Let $\tau$ be a pretrained transformer. Let $P \in \mathbb{R}^{d \times L_p}$ be a length-$L_p$ prompt weight (termed *soft-prompt*) prepended to input prompt $X$ such that $X_p := [P, X] \in \mathbb{R}^{d \times (L_p + L)}$. For any downstream task with finetuning dataset $S = \{(X^{(i)}, Y^{(i)})\}_{i \in [N]}$, the problem of prompt tuning is to find a prompt weight $P^\star$ by solving the following optimization problem

$$P^\star := \underset{P}{\arg\min} \sum_{i=1}^{N} \ell\Big(\tau\big(X_p^{(i)}\big)_{:,L_p:}, Y^{(i)}\Big), \quad \text{for some loss} \quad \ell : \mathbb{R}^{d \times L} \times \mathbb{R}^{d \times L} \to \mathbb{R}_+. \quad (1.1)$$

In this work, we aim to study Definition 1.1 statistically and computationally.

Statistically, we explore the expressive power of prompt tuning a transformer of the simplest configuration. Formally, we investigate whether it is possible to approximate any sequence-to-sequence function $f$ through prompt tuning with a pretrained single-head, single-layer transformer $\tau$ such that

$$d_\alpha\left(\tau([P^\star, \cdot])_{:,L_p:}, f\right) \le \epsilon, \quad \text{for some } \epsilon > 0, \quad (1.2)$$

where approximation error $\epsilon$ between two functions is $d_\alpha(f_1, f_2) := (\int \|f_1(X) - f_2(X)\|_\alpha^\alpha dX)^{1/\alpha}$. Here, $\|\cdot\|_\alpha$ denotes entrywise $\ell_\alpha$-norm, i.e., $\|X\|_\alpha = (\sum_{i=1}^{d} \sum_{j=1}^{L} |X_{i,j}|^\alpha)^{1/\alpha}$. Specifically, while Wang et al. (2023a, Theorem 1) report the universality[1] of prompt tuning transformers with $\mathcal{O}((L_p + L)(1/\epsilon)^d)$ attention layers with 2 heads of hidden dimension[2] 1 and $\mathcal{O}((1/\epsilon)^{d(L_p+L)})$ FFN layers with 4 MLP neurons, we ask the following question:

**Question 1.** Is it possible to improve (Wang et al., 2023a) toward the universality of prompt tuning on single-head single-layer pretrained transformers?

To answer Question 1, we first refine previous results of attention contextual mapping (Lemma 2.2) and establish a chaining reduction for bounding approximation error of prompt tuning (Section 2.3).

Computationally, we investigate the computational hardness of prompt tuning in transformer-based foundation models using fine-grained complexity theory (Williams, 2018). We observe that the computational hardness of prompt tuning ties to the quadratic time complexity of the transformer attention heads. Although designing algorithms to bypass this $\Omega(L^2)$ computation time is tempting, to the best of our knowledge, there lacks formal results to support and describe such approaches in a comprehensive fashion. To bridge this gap, we pose below questions and develop a foundational theory to characterize the complexity of prompt tuning for large transformer-based models:

**Question 2.** Is it possible to improve the $\Omega(L^2)$ time with a bounded approximation error?

**Question 3.** More aggressively, is it possible to do such computations in almost linear time $L^{1+o(1)}$?

In this work, we answer both Questions 2 and 3 for the forward inference of prompt tuning. To answer them, we explore approximate prompt tuning computations with precision guarantees. To be concrete, let $W_K, W_Q, W_V \in \mathbb{R}^{d \times d}$ be attention weights such that $Q = W_V X \in \mathbb{R}^{d \times L}$, $K = W_K X \in \mathbb{R}^{d \times L}$ and $V = W_V X \in \mathbb{R}^{d \times L}$. Recall the Attention Mechanism

$$Z = V \, \mathrm{Softmax}\left(K^\mathsf{T} Q \beta\right) = (W_V X) D^{-1} \exp\left(X^\mathsf{T} W_K^\mathsf{T} W_Q X \beta\right) \in \mathbb{R}^{d \times L}, \quad (1.3)$$

with the inverse temperature $\beta > 0$ and $D := \mathrm{diag}\left(\exp\left(X^\mathsf{T} W_K^\mathsf{T} W_Q X \beta\right) \mathbb{1}_L\right)$. Here, $\exp(\cdot)$ is entry-wise exponential function. For simplicity of presentation, we set $\beta = 1$ in this work.

Formally, we study the following approximation problem for prompt tuning inference. Let $Q_p = W_Q X_p \in \mathbb{R}^{d \times (L_p + L)}$, $K_p = W_K X_p \in \mathbb{R}^{d \times (L_p + L)}$, and $V_p = W_K X_p \in \mathbb{R}^{d \times (L_p + L)}$.

**Problem 1** (Approximate Prompt Tuning Inference APTI). Let $\delta_F > 0$ and $B > 0$. Given $Q_p, K_p, V_p \in \mathbb{R}^{d \times (L + L_p)}$ with guarantees that $\max\{\|Q_p\|_{\max}, \|K_p\|_{\max}, \|V_p\|_{\max}\} \le B$, we aim to study an approximation problem $\mathrm{APTI}(d, L, L_p, B, \delta_F)$, aiming to approximate

---

[1]Throughout this work, *universality* refers to sequence-to-sequence (seq2seq) universal approximation.
[2]For attention weights $W_V, W_K, W_Q \in \mathbb{R}^{s \times d}$, hidden dimension is $s$.

$V_p \operatorname{Softmax}\left(K_p^\mathsf{T} Q_p\right)$ with a matrix $\widetilde{Z}$ such that

$$\|\widetilde{Z} - V_p \operatorname{Softmax}\left(K_p^\mathsf{T} Q_p\right)\|_{\max} \leq \delta_F,$$

Here, for a matrix $M \in \mathbb{R}^{a \times b}$, we write $\|M\|_{\max} := \max_{i,j} |M_{i,j}|$.

In this work, we aim to investigate the computational limits of all possible efficient algorithms for $\textsc{Apti}(d, L, L_p, B, \delta_F)$ under realistic setting $\delta_F = 1/\operatorname{poly}(L)$.

**Contributions.** We study the fundamental limits of prompt tuning. Our contributions are threefold:

- **Universality.** We prove that prompt tuning transformers with the simplest configurations — single-head, single-layer attention — are universal approximators for Lipschitz sequence-to-sequence functions. Additionally, we reduce the required number of FFN layers in the prompt tuning transformer to 2. These results improve upon (Wang et al., 2023a), which requires deep transformers with $\mathcal{O}((L_p + L)(1/\epsilon)^d)$ attention layers and $\mathcal{O}((1/\epsilon)^{d(L_p+L)})$ FFN layers.

- **Memorization.** We show that prompt tuning such simple transformers (1-head, 1-layer attention and 2 FNN layers) is capable of complete memorization of datasets without any assumption on the data. Moreover, we establish an exponential-in-$dL$ and -in-$(1/\epsilon)$ lower bound on the required soft-prompt tokens for any dataset, where $d, L$ are the data dimension and sequence length, respectively, and $\epsilon$ is the approximation error. Our results improve upon those of (Wang et al., 2023a), which consider datasets with only two-token sequences and focus solely on memorizing the final token.

- **Efficiency.** We address Question 2 by identifying a phase transition behavior in efficiency based on the norm of soft-prompt-induced queries and keys (Theorem 3.1). This establishes an efficiency criterion for prompt tuning inference, enabling efficient (sub-quadratic) algorithms when the criterion is met. Additionally, we address Question 3 by pushing the limits of efficiency in prompt tuning toward nearly-linear time under this criterion (Theorem 3.2).

**Organization.** Section 2 presents a statistical analysis on prompt tuning's universality and memory capacity. Section 3 explore the computational limits of inference with prompt tuning. The appendix includes the related works (Appendix B.1) and the detailed proofs of the main text.

**Notations.** We use lower case letters to denote vectors and upper case letters to denote matrices. The index set $\{1, ..., I\}$ is denoted by $[I]$, where $I \in \mathbb{N}^+$. We write $\ell_\alpha$-norm as $\|\cdot\|_\alpha$. Throughout this paper, we denote input, label sequences as $X, Y \in \mathbb{R}^{d \times L}$ and prompt sequences as $P \in \mathbb{R}^{d \times L_p}$.

## 2 STATISTICAL LIMITS OF PROMPT TUNING: UNIVERSALITY AND CAPACITY

To better understand the expressive power of prompt tuning, we explore its universality (Sections 2.3 and 2.4) and memory capacity (Section 2.5) on a transformer of simplest configurations.

**Overview of Our Results.** Let $\mathcal{T}^{h,s,r}$ denote transformers with $h$ heads, $s$ hidden size, and $r$ MLP neurons, and let $\epsilon$ represent the approximation error tolerance. Let $X \in \mathbb{R}^{d \times L}$ and $P \in \mathbb{R}^{d \times L_p}$ be the input and soft-prompt defined in Definition 1.1, respectively. We answer Question 1 affirmatively, and present three results for transformer models with 1-head, 1-layer attention layers:

**Lemma 2.1** (1-Head, 1-Layer Attention with Any-Rank Weight Matrices Is Contextual Mapping, Informal Version of Lemma 2.2). *A 1-head, 1-layer attention mechanism with weight matrices $W_K, W_Q, W_V$ of any rank is able to associate each input sequence with a unique label sequence.*

**Theorem 2.1** (Universality of Prompt Tuning $\mathcal{T}^{1,1,4}$ Transformers with $\mathcal{O}(\epsilon^{-d(L_p+L)})$ FFN Layers, Informal Version of Theorem 2.3). *Prompt tuning transformers with 1 head, a hidden size of 1, and $\mathcal{O}(\epsilon^{-d(L_p+L)})$ FFN layers of width 4 are universal approximators for Lipschitz seq-to-seq functions.*

**Theorem 2.2** (Universality of Prompt Tuning $\mathcal{T}^{1,1,r=\mathcal{O}(\epsilon^{-d(L_p+L)})}$ Transformers with 2 FFN Layers, Informal Version of Theorem 2.4). *Prompt tuning transformers with 1 head, a hidden size of 1, and 2 FFN layers of width $\mathcal{O}(\epsilon^{-d(L_p+L)})$ are universal approximators for Lipschitz seq-to-seq functions.*

**Comparing with Prior Works.** Our results improve previous works in three aspects:

- **Any Weight Matrices.** While Kajitsuka and Sato (2024) show that a self-attention layer with rank-1 weight matrices serves as a *contextual map*, we improve this to *weight matrices of any rank*.
- **Transformers with 1-Head, 1-Layer Attention.** While Wang et al. (2023a) show that prompt tuning on transformers of $\mathcal{O}((L_p + L)(1/\epsilon)^d)$ attention layers with at least 2 attention heads, we achieve the universality of prompt tuning transformers with only *single-head-single-layer* attention.
- **Only 2 FFN Layers.** We identify a width-depth tradeoff of universality. While Wang et al. (2023a) achieve prompt tuning universality with transformers of $\mathcal{O}((1/\epsilon)^{d(L_p+L)})$ FFN layers, we show that the same universality holds with 1-head, 1-layer transformers of *only* 2 *FFN layers*.

**Technical Overview.** Our proof strategy is to characterize the joint approximation error from different components of a transformer block via a chained reduction of piece-wise constant approximations.

- **Quantized Functions and Piece-Wise Constant Approximations**
  - (P1) **Piece-Wise Constant Approximation.** We consider a class of Lipschitz functions as our target functions $f_{\text{seq2seq}}$, and employ *piece-wise constant approximations*[3]. Namely, we first quantize the input and output domain of the target functions and obtain a class of quantized target functions. These quantized target functions (denoted by $\bar{f}_{\text{seq2seq}}$) are piece-wise constant functions mapping grids of the input domain to grids of the output domain.
  - (P2) **Surrogate Prompt Tuning Transformer.** Next, we construct a *surrogate* function $h_{\text{seq2seq}}$ for the transformer. This surrogate function takes prompts (i.e., $Z_p = [P, Z] \in \mathbb{R}^{d \times (L_p+L)}$) as inputs. We approximate each quantized target function $\bar{f}_{\text{seq2seq}}$ with $L_p$-imputed output of $h_{\text{seq2seq}}$. Namely, we only use the last $L$ output tokens of $h_{\text{seq2seq}}$ to approximate $\bar{f}_{\text{seq2seq}}$. We achieve this by associating a unique prompt with each quantized target function.
  - (P3) **Prompting Tuning Transformer Approximate $h_{\text{seq2seq}}$.** Then, we construct a transformer $\tau$ on which prompt tuning approximates the surrogate function $h_{\text{seq2seq}}$ with bounded error.
- **Chained Reduction of Piece-Wise Constant Approximations**
  - A transformer layer consists of a self-attention layer $f^{(\text{Att})}$ and an FFN layer. We utilize $f^{(\text{Att})}$ as a contextual mapping. A $(\delta, \gamma)$-contextual mapping preserves the correspondence of its input-output pair up to $(\delta, \gamma)$ accuracy (see Definition 2.6 for formal definition). Furthermore, instead of just token-wise manipulation, contextual mapping allows us to capture the context of an input sequence as a whole. This allows us to quantify the *quality* of a mapping in terms of its ability to perform piece-wise approximation up to any precision.
  - Lastly, we use FFN layers to map the outputs of $f^{(\text{Att})}$ to the desired outputs within a bounded error. This results in a chained reduction of approximation errors; we observe that for each step, $\text{Error}[(P3)] \geq \text{Error}[(P2)] \geq \text{Error}[(P1)]$. Therefore, we conclude that prompt tuning on the transformer $\tau$ is a universal approximator for our target functions $f$.

## 2.1 PRELIMINARIES AND PROBLEM SETUP

We first present the ideas we build on.

Let $Z \in \mathbb{R}^{d \times L}$ denote the input embeddings of attention layer and $s$ denote the hidden dimension.

**Transformer Block.** Let $h$-head self-attention layer as a function $f^{(\text{SA})} : \mathbb{R}^{d \times L} \to \mathbb{R}^{d \times L}$,

$$f^{(\text{SA})}(Z) = Z + \sum_{i=1}^{h} W_O^i f_i^{(\text{Att})}(Z, Z) \in \mathbb{R}^{d \times L}, \tag{2.1}$$

where $W_O^i \in \mathbb{R}^{d \times s}$ and $f_i^{(\text{Att})}$ is the size-$s$ self-attention mechanism for the $i$-th head

$$f_i^{(\text{Att})}(Z_{:,k}, Z) = (W_V^i Z) \operatorname{Softmax}\left[(W_K^i Z)^\top (W_Q^i Z_{:,k})\right] \in \mathbb{R}^s.$$

Here, $f_i^{(\text{Att})} : \mathbb{R}^d \times \mathbb{R}^{d \times L} \mapsto \mathbb{R}^s$ acts token-wise, and $W_V^i, W_K^i, W_Q^i \in \mathbb{R}^{s \times d}$ are the weight matrices. Next, we define the $r$-neuron feed-forward layer function as $f^{(\text{FF})} \in \mathcal{F}^{(\text{FF})} : \mathbb{R}^{d \times L} \mapsto \mathbb{R}^{d \times L}$ and the

---

[3]A piece-wise constant approximation approximates a function $f_{\text{seq2seq}}$ by a series of constant values across different segments of its domain. This technique involves discretizing the function's domain into intervals and assigning a constant value to the function over each interval. Please see (Yun et al., 2020) for utilizing piece-wise constant approximations for the transformer's universality.

output at $k$-th token as

$$f^{(\mathrm{FF})}(Z)_{:,k} = Z_{:,k} + W^{(2)}\mathrm{ReLU}(W^{(1)}Z_{:,k} + b^{(1)}) + b^{(2)}, \qquad (2.2)$$

where $W^{(1)} \in \mathbb{R}^{r \times d}$ and $W^{(2)} \in \mathbb{R}^{d \times r}$ are weight matrices, and $b^{(1)}, b^{(2)} \in \mathbb{R}^r$ are the bias terms.

**Definition 2.1** (Transformer Block). We define a transformer block of $h$-head, $s$-size and $r$-neuron as $f^{(\mathcal{T}^{h,s,r})}(Z) = f^{(\mathrm{FF})}\left(f^{(\mathrm{SA})}(Z)\right) : \mathbb{R}^{d \times L} \mapsto \mathbb{R}^{d \times L}$.

Now, we define the transformer networks as compositions of transformer blocks.

**Definition 2.2** (Transformer Network Function Class). Let $\mathcal{T}^{h,s,r}$ denote the transformer network function class where each function $\tau \in \mathcal{T}^{h,s,r}$ consists of transformer blocks $f^{(\mathcal{T}^{h,s,r})}$ with $h$ heads of size $s$ and $r$ MLP hidden neurons: $\mathcal{T}^{h,s,r} := \{\tau : \mathbb{R}^{d \times L} \mapsto \mathbb{R}^{d \times L} \mid \tau = f^{(\mathcal{T}^{h,s,r})}(f^{(\mathcal{T}^{h,s,r})}(\cdots))\}$.

**Prompt Tuning Pretrained Transformer Models.** In this work, we consider the prompt tuning problem Definition 1.1 with a pretrained transformer network $\tau \in \mathcal{T}^{h,s,r}$.

**Problem Setup.** To answer Question 1, we focus on the universal approximation of prompt tuning pretrained transformer models. We start by stating the target functions of our approximation.

**Definition 2.3** (Target Function Class). Let $\mathcal{F}_C$ be the $C$-Lipschitz (under $p$-norm) target function class of continuous sequence-to-sequence. Let $f_{\mathrm{seq2seq}} \in \mathcal{F}_C : [0,1]^{d \times L} \mapsto [0,1]^{d \times L}$ denote continuous sequence-to-sequence functions on a compact set of sequence.

Explicitly, for any $f_{\mathrm{seq2seq}} \in \mathcal{F}_C$ and two input sequences $Z, Z' \in \mathbb{R}^{d \times L}$, we have $\|f_{\mathrm{seq2seq}}(Z) - f_{\mathrm{seq2seq}}(Z')\|_\alpha \leq C\|Z - Z'\|_\alpha$. In this work, we adopt $f_{\mathrm{seq2seq}}$ as our approximation target function. Concretely, we investigate whether it is possible to approximate any $C$-Lipschitz sequence-to-sequence function $f_{\mathrm{seq2seq}}$ through prompt tuning with a pretrained single-head, single-layer transformer model. Namely, we reformulate Question 1 into the following problem.

**Problem 2.** Is it possible to find a pretrained transformer model $\tau \in \mathcal{T}^{1,1,r}$ such that, for any $f_{\mathrm{seq2seq}} \in \mathcal{F}_C$, prompt tuning $\tau$ satisfies $d_\alpha\left(\tau([P,\cdot])_{:,L_p:}, f_{\mathrm{seq2seq}}\right) \leq \epsilon$ for some $\epsilon > 0$? Here,

$$d_\alpha(f_1, f_2) := \left(\int \|f_1(Z) - f_2(Z)\|_\alpha^\alpha \mathrm{d}Z\right)^{1/\alpha},$$

measures the difference between functions $f_1$ and $f_2$ in the token-wise $\ell_\alpha$-norm.

## 2.2 Any-Rank Single-Layer Attention is a Contextual Mapping Function

In this subsection, we present new results on the contextual mapping property of attention. These results allow us to use feed-forward neural networks to map each input sequence to its corresponding label sequence, thereby achieving universal approximation in Section 2.3.

**Background: Contextual Mapping.** As stated in the previous technical overview, a key element of our proof is the concept of contextual mapping in attention (Kajitsuka and Sato, 2024; Yun et al., 2020). Contextual mapping enables transformers to move beyond simple token-wise manipulation and capture the full context of a sequence. Through this, identical tokens within different input sequences become distinguishable. Let $Z, Y \in \mathbb{R}^{d \times L}$ be the input embeddings and output label sequences, respectively. Let $Z_{:,i} \in \mathbb{R}^d$ be the $i$-th token (column) of each $Z$ embedding sequence.

**Definition 2.4** (Vocabulary). We define the $i$-th vocabulary set for $i \in [N]$ by $\mathcal{V}^{(i)} = \bigcup_{k \in [L]} Z_{:,k}^{(i)} \subset \mathbb{R}^d$, and the whole vocabulary set $\mathcal{V}$ is defined by $\mathcal{V} = \bigcup_{i \in [N]} \mathcal{V}^{(i)} \subset \mathbb{R}^d$.

Note that while "vocabulary" typically refers to the tokens' codomain, here it refers to the set of all tokens within a single sequence. To facilitate our analysis, we introduce the idea of input token separation following (Kajitsuka and Sato, 2024; Kim et al., 2022; Yun et al., 2020).

**Definition 2.5** (Tokenwise Separateness). Let $Z^{(1)}, \ldots, Z^{(N)} \in \mathbb{R}^{d \times L}$ be embeddings. Then, $Z^{(1)}, \ldots, Z^{(N)}$ are called tokenwise $(\gamma_{\min}, \gamma_{\max}, \delta)$-separated if the following conditions hold.

(i) For any $i \in [N]$ and $k \in [L]$, $\|Z_{:,k}^{(i)}\| > \gamma_{\min}$ holds.

(ii) For any $i \in [N]$ and $k \in [L]$, $\|Z_{:,k}^{(i)}\| < \gamma_{\max}$ holds.

(iii) For any $i, j \in [N]$ and $k, l \in [L]$ if $Z_{:,k}^{(i)} \neq Z_{:,l}^{(j)}$, then $\|Z_{:,k}^{(i)} - Z_{:,l}^{(j)}\| > \delta$ holds.

Note that when only conditions (ii) and (iii) hold, we denote this as $(\gamma, \delta)$-separateness. Moreover, if only condition (iii) holds, we denote it as $(\delta)$-separateness.

To clarify condition (iii), we consider cases where there are repeated tokens between different input sequences. Next, we define contextual mapping. Contextual mapping describes a function's ability to capture the context of each input sequence as a whole and assign a unique ID to each input sequence.

**Definition 2.6** (Contextual Mapping). A function $q : \mathbb{R}^{d \times L} \to \mathbb{R}^{d \times L}$ is said to be a $(\gamma, \delta)$-contextual mapping for a set of embeddings $Z^{(1)}, \ldots, Z^{(N)} \in \mathbb{R}^{d \times L}$ if the following conditions hold:

1. **Contextual Sensitivity $\gamma$.** For any $i \in [N]$ and $k \in [L]$, $\|q(Z^{(i)})_{:,k}\| < \gamma$ holds.

2. **Approximation Error $\delta$.** For any $i, j \in [N]$ and $k, l \in [L]$ such that $\mathcal{V}^{(i)} \neq \mathcal{V}^{(j)}$ or $Z_{:,k}^{(i)} \neq Z_{:,l}^{(j)}$, $\|q(Z^{(i)})_{:,k} - q(Z^{(j)})_{:,l}\| > \delta$ holds.

Note that $q\left(Z^{(i)}\right)$ for $i \in [N]$ is called a *context ID* of $Z^{(i)}$.

**Any-Rank Attention is Contextual Mapping.** Now we present the result showing that a 1-head, 1-layer (softmax-)attention block with any-rank weight matrices is a contextual mapping.

**Lemma 2.2** (Any-Rank Attention as a $(\gamma, \delta)$-Contextual Mapping, Modified from Theorem 2 of (Kajitsuka and Sato, 2024)). Let $Z^{(1)}, \ldots, Z^{(N)} \in \mathbb{R}^{d \times L}$ be $(\gamma_{\min}, \gamma_{\max}, \epsilon)$-tokenwise separated embeddings, with the vocabulary set $\mathcal{V} = \bigcup_{i \in [N]} \mathcal{V}^{(i)} \subset \mathbb{R}^d$. Additionally, assume no duplicate word tokens in each sequence, i.e., $Z_{:,k}^{(i)} \neq Z_{:,l}^{(i)}$ for any $i \in [N]$ and $k, l \in [L]$. Then, there exists a 1-layer, single-head attention mechanism with weight matrices $W^{(O)} \in \mathbb{R}^{d \times s}$ and $W_V, W_K, W_Q \in \mathbb{R}^{s \times d}$ that serves as a $(\gamma, \delta)$-contextual mapping for the embeddings $Z^{(1)}, \ldots, Z^{(N)}$, where:

$$\gamma = \gamma_{\max} + \frac{\epsilon}{4} \quad \text{and} \quad \delta = \exp\left(-5\epsilon^{-1}|\mathcal{V}|^4 d\kappa\gamma_{\max} \log L\right), \quad \text{with} \quad \kappa := \gamma_{\max}/\gamma_{\min}.$$

*Proof Sketch.* We generalize (Kajitsuka and Sato, 2024, Theorem 2) where all weight matrices have to be rank-1. We eliminate the rank-1 requirement by constructing the weight matrices as an outer product sum $\sum_i^\rho u_i v_i^\top$, where $u_i \in \mathbb{R}^s$, $v_i \in \mathbb{R}^d$. This extends (Kajitsuka and Sato, 2024, Theorem 2) holds for attention with weights of any rank. Please see Appendix E.1 for a detailed proof. □

Lemma 2.2 indicates that any-rank self-attention function distinguishes input tokens $Z_{:,k}^{(i)} = Z_{:,l}^{(j)}$ such that $\mathcal{V}^{(i)} \neq \mathcal{V}^{(j)}$. In other words, it distinguishes two identical tokens within a different context.

**Remark 2.1** (Comparing with Existing Works). In comparison, Kajitsuka and Sato (2024) provide a proof for the case where all self-attention weight matrices $W_V, W_K, W_Q \in \mathbb{R}^{s \times d}$ are rank-1 strictly. However, this is almost impossible in practice for any pre-trained transformer-based models. In contrast, by considering self-attention weight matrices of rank $\rho$, where $1 \leq \rho \leq \min(d, s)$, we show that single-head, single-layer self-attention with matrices of any rank is a contextual mapping, pushing the universality of (prompt-tuning) transformers towards more practical scenarios.

Next, we utilize Lemma 2.2 to prove the universality and memory capacity of prompt tuning on transformer networks with single layer self-attention.

## 2.3 UNIVERSALITY OF PROMPT TUNING $\mathcal{T}_A^{1,1,4}$ WITH $\mathcal{O}((1/\epsilon)^{d(L_p+L)})$ FFN LAYERS

In this section, we prove the universality of prompt tuning by showing that there exists a simple transformer of single-layer self-attention $\tau \in \mathcal{T}_A^{1,1,4}$ such that for any $f_{\text{seq2seq}} \in \mathcal{F}_C$, prompt tuning on $\tau$ approximates this function up to some error $\epsilon > 0$. Specifically, $\tau \in \mathcal{T}_A^{1,1,4}$ consists of a single-head, single-layer, size-one self-attention function $f^{(\text{SA})} \in \mathcal{F}^{(\text{SA})}$, and $\mathcal{O}((1/\epsilon)^{d(L_p+L)})$ feed-forward layers $f^{(\text{FF})} \in \mathcal{F}^{(\text{FF})}$, each with 4 MLP hidden neurons. Formally, we write

$$\mathcal{T}_A^{1,1,4} := \{\tau : \mathbb{R}^{d \times L} \mapsto \mathbb{R}^{d \times L} \mid \tau = f_{\ell_1}^{(\text{FF})} \circ \ldots \circ f_1^{(\text{FF})} \circ f^{(\text{SA})} \circ f_{\ell_2}^{(\text{FF})} \circ \ldots \circ f_1^{(\text{FF})}\}. \quad (2.3)$$

**Proof Strategy.** We employ a chained reduction of piece-wise constant approximations:

(A1) We start by quantizing the input and output domain of $f_{\text{seq2seq}} \in \mathcal{F}_C$ into a quantized function

$$\bar{f}_{\text{seq2seq}} : \mathcal{G}_{\delta,L} \mapsto \mathcal{G}_{\delta,L}, \quad \text{where} \quad \mathcal{G}_{\delta,L} = \{0, \delta, 2\delta, \ldots, 1 - \delta\}^{d \times L}.$$

Here, $\bar{f}_{\text{seq2seq}}, \overline{\mathcal{F}}_C$ denote the quantized function and function class. In essence, this is performing a piece-wise constant approximation with bounded error $\delta$.

(A2) Next, we construct a surrogate quantized sequence-to-sequence function

$$h_{\text{seq2seq}} : \mathcal{G}_{\delta,(L_p+L)} \to \mathcal{G}_{\delta,(L_p+L)}, \text{ where } \mathcal{G}_{\delta,(L_p+L)} = \{0, \delta, 2\delta, \ldots, 1 - \delta\}^{d \times (L_p+L)}.$$

Here $h_{\text{seq2seq}}$ takes prompts and embeddings $Z_p = [P, Z]$ as inputs. Crucially, its $L_p$-imputed output approximates any $\bar{f}_{\text{seq2seq}} \in \overline{\mathcal{F}}_C$ by using various soft prompts $P$.

(A3) Finally, we show that there exist transformers $\tau \in \mathcal{T}_A^{1,1,4}$ approximating $h_{\text{seq2seq}}$ to any precision. By simple reduction from $h_{\text{seq2seq}}, \bar{f}_{\text{seq2seq}}$ and $f_{\text{seq2seq}}$, we achieve the universality of prompt tuning on $\mathcal{T}_A^{1,1,4}$ with $\mathcal{O}((1/\epsilon)^{d(L_p+L)})$ FFN layers, where $\epsilon$ is the approximation error.

**Remark 2.2.** We remark that while (A1) shares some similarity with (Wang et al., 2023a) by the nature of quantization approach to transformer's universality (Yun et al., 2020), (A2) and (A3) differ significantly in techniques and results. See the opening of this section for an overview.

For (A1) and (A2), we introduce the next lemma, and approximate the quantized $\bar{f}_{\text{seq2seq}}$ with a $L_p$-imputed version of some quantized sequence-to-sequence function

$$h_{\text{seq2seq}} : \mathcal{G}_{\delta,(L_p+L)} \to \mathcal{G}_{\delta,(L_p+L)}, \text{ where } \mathcal{G}_{\delta,(L_p+L)} = \{0, \delta, 2\delta, \ldots, 1 - \delta\}^{d \times (L_p+L)}.$$

**Lemma 2.3** (Universality of Prompt Tuning Surrogate Function $h_{\text{seq2seq}}$). Let $\mathcal{F}_C$ be a class of $C$-Lipschitz sequence-to-sequence functions, where each function $f_{\text{seq2seq}} : [0,1]^{d \times L} \to [0,1]^{d \times L}$. Define the discrete function space $\mathcal{G}_{\delta,(L_p+L)} = \{0, \delta, 2\delta, \ldots, 1 - \delta\}^{d \times (L_p+L)}$. Then, there exists a function $h_{\text{seq2seq}} : \mathcal{G}_{\delta,(L_p+L)} \to \mathcal{G}_{\delta,(L_p+L)}$ such that for any $f_{\text{seq2seq}} \in \mathcal{F}_C$, there exists a prompt $P \in \mathbb{R}^{d \times L_p}$ satisfying

$$d_p\left(h([P, \cdot])_{:,L_p:}, f_{\text{seq2seq}}\right) \le \epsilon/2,$$

where the prompt sequence length $L_p$ satisfies $L_p \ge L\lambda$, with $\lambda = (2\epsilon^{-1}C(dL)^{1/\alpha})^{dL}$.

*Proof Sketch.* Our proof consists of three steps. Firstly, we approximate each function in $\mathcal{F}_C$ by a piece-wise constant function in $\overline{\mathcal{F}}_C$. $\overline{\mathcal{F}}_C$ is constructed by quantizing the input and output domain of $\mathcal{F}_C$. This gives us a function class of limited size, so that further discussion is feasible. Secondly, we construct a quantized prompt set $\mathcal{P}$. We correspond each quantized function $\bar{f}_{\text{seq2seq}}^{(i)} \in \overline{\mathcal{F}}_C$ to a prompt $P^{(i)} \in \mathcal{P}$. Lastly, we build a sequence-to-sequence function

$$h_{\text{seq2seq}} : \mathcal{G}_{\delta,(L_p+L)} \to \mathcal{G}_{\delta,(L_p+L)} \quad \text{with} \quad \mathcal{G}_{\delta,(L_p+L)} = \{0, \delta, 2\delta, \ldots, 1 - \delta\}^{d \times (L_p+L)},$$

that takes a soft-prompt $P$ and embeddings $Z$ as input. Most importantly, this function $h_{\text{seq2seq}}$ behaves like $\bar{f}_{\text{seq2seq}}^{(i)}$ when taking the corresponding prompt $P^{(i)}$. Namely, it satisfies $h_{\text{seq2seq}}([P^{(i)}, \cdot])_{:,L_p:} = \bar{f}_{\text{seq2seq}}^{(i)}(\cdot)$. Please see Appendix F.1 for a detailed proof. $\qquad\square$

For (A3), we present the next lemma demonstrating that $\tau \in \mathcal{T}_A^{1,1,4}$ approximates $h_{\text{seq2seq}}$ up to any desired precision. The technical contribution involves using the contextual mapping property of any-rank 1-layer, 1-head attention (Lemma 2.2) to preserve the piece-wise constant approximation.

**Lemma 2.4** (Transformer $\tau \in \mathcal{T}_A^{1,1,4}$ Approximate $h_{\text{seq2seq}}$ to Any Precision). For any given quantized sequence-to-sequence function $h_{\text{seq2seq}} : \mathcal{G}_{\delta,(L_p+L)} \to \mathcal{G}_{\delta,(L_p+L)}$ with $\mathcal{G}_{\delta,(L_p+L)} =$

$\{0, \delta, 2\delta, \ldots, 1 - \delta\}^{d \times (L_p + L)}$, there exists a transformer $\tau \in \mathcal{T}_A^{1,1,4}$ with positional embedding $E \in \mathbb{R}^{d \times (L_p + L)}$, such that $\tau = h([P, \cdot])_{:, L_p:}$.

*Proof.* Please see Appendix F.2 for a detailed proof. □

Altogether, we state our main result: the universality of prompt tuning for a $\tau \in \mathcal{T}_A^{1,1,4}$ transformer.

**Theorem 2.3** (Universality of Prompt Tuning for $\tau \in \mathcal{T}_A^{1,1,4}$ Transformers). Let $1 \le p < \infty$ and $\epsilon > 0$. There exists a transformer $\tau \in \mathcal{T}_A^{1,1,4}$ with a single self-attention layer such that for any $f_{\text{seq2seq}} \in \mathcal{F}_C$, there exists a prompt $P \in \mathbb{R}^{d \times L_p}$ satisfying $d_\alpha \left( \tau([P, \cdot])_{:, L_p}, f_{\text{seq2seq}} \right) \le \epsilon$.

*Proof Sketch.* By Lemmas 2.3 and 2.4, we obtain a $\tau \in \mathcal{T}_A^{1,1,4}$, with soft-prompt $P \in \mathcal{G}_{\delta, L_p}$, such that for any $f_{\text{seq2seq}} \in \mathcal{F}_C, d_\alpha \left( \tau([P, \cdot])_{:, L_p:}, f_{\text{seq2seq}} \right) \le \epsilon$. See Appendix F.3 for a detailed proof. □

Intuitively, Theorem 2.3 indicates that even the simplest transformer with 1-head, 1-layer attention has enough expressive power through prompt tuning to approximate any Lipschitz seq2seq function.

## 2.4 WIDTH-DEPTH TRADEOFF: UNIVERSALITY OF PROMPT TUNING $\mathcal{T}^{1,1,r=\mathcal{O}\left((1/\epsilon)^{d(L_p+L)}\right)}$ ONLY NEEDS 2 FFN LAYERS

In Section 2.3, we achieve the universality of prompt tuning simple transformers with many FFN layers. In this section, we explore the possibility of further simplifying such transformer blocks by reducing the number of FFN layers. Surprisingly, we show that two FFN layers are sufficient.

We begin by determining the minimum number of FFN layers required for $\tau \in \mathcal{T}_A^{1,1,4}$ transformers to achieve universality through prompt tuning. For clarity, we denote the transformer with four MLP neurons as $\mathcal{T}_A$ (i.e., (2.3)) and the transformer with $r$ MLP neurons as $\mathcal{T}_B$.

**Lemma 2.5** (Required Number of FFN Layers). A transformer $\tau \in \mathcal{T}_A^{1,1,4}$, as defined in (2.3), requires $\mathcal{O}((1/\epsilon)^{d(L_p+L)})$ FFN layers to be a universal approximator through prompt tuning.

*Proof.* Please see Appendix G.1 for a detailed proof. □

Next, we prove the universality of prompt tuning on another simple transformer block $\mathcal{T}_B^{1,1,r}$ with smaller FFN depth than $\mathcal{T}_A^{1,1,4}$ from Section 2.3. This suggests a trade-off between the depth and width of the transformer. Let transformers $\tau \in \mathcal{T}_B^{1,1,r}$ consist of a single-head, single-layer, size-one self-attention $f^{(\text{SA})}$ and 2 feed-forward layers, $f_1^{(\text{FF})}$ and $f_2^{(\text{FF})}$, each with $r$ MLP hidden neurons:

$$\mathcal{T}_B^{1,1,r} := \{\tau : \mathbb{R}^{d \times L} \mapsto \mathbb{R}^{d \times L} \mid \tau = f_2^{(\text{FF})} \circ f^{(\text{SA})} \circ f_1^{(\text{FF})}\}.$$

**Proof Strategy.** We follow a similar proof strategy as in Section 2.3, but with a key difference: we use the construction technique from (Kajitsuka and Sato, 2024) to build a transformer with single-head, single-layer, size-one self-attention, and two FFN layers. We achieve this by summing multiple shifted ReLU functions to map inputs to desired outputs with precision guarantees. Moreover, this approach reduces the number of FFN layers by increasing the number of neurons in the MLP.

**Theorem 2.4** (Prompt Tuning Transformers with Single-Head, Single-Layer Attention and Two Feed-Forward Layers). Let $1 \le p < \infty$ and $\epsilon > 0$. There exists a transformer $\tau \in \mathcal{T}_B^{1,1,r}$ with a single self-attention layer and $r = \mathcal{O}\left((1/\epsilon)^{d(L_p+L)}\right)$ MLP neurons, such that for any $f_{\text{seq2seq}} \in \mathcal{F}_C$, there exists a prompt $P \in \mathbb{R}^{d \times L_p}$ satisfying: $d_p \left( \tau([P, \cdot])_{:, L_p}, f_{\text{seq2seq}} \right) \le \epsilon$.

*Proof.* Please see Appendix G.2 for a detailed proof. □

## 2.5 MEMORY CAPACITY OF PROMPT TUNING

Based on our universality results, we show the memory capacity of prompt tuning on simple transformer networks with single-head single-layer self-attention. We start with the definition.

**Definition 2.7** (Prompt Tuning Memorization). Given a dataset $S = \{(X^{(i)}, Y^{(i)})\}_{i=1}^{N}$ with $X^{(i)}, Y^{(i)} \in \mathbb{R}^{d \times L}$, a pretrained transformer $\tau \in \mathcal{T}$ memorizes $S$ through prompt tuning if there exists a prompt $P \in \mathbb{R}^{d \times L_p}$ such that: $\max_{i \in [N]} \left\| \tau([P, X^{(i)}])_{:, L_p} - Y^{(i)} \right\|_\alpha \leq \epsilon$ for all $i \in [N]$.

We now prove the existence of a transformer $\tau \in \mathcal{T}_B^{1,1,r}$ that memorizes any dataset $S$ through prompt tuning. We remark that this result is easy to extend to $\tau \in \mathcal{T}_A^{1,1,4}$ transformers.

**Theorem 2.5** (Memorization Capacity of Prompt Tuning). Consider a dataset $S = \{(X^{(i)}, Y^{(i)})\}_{i=1}^{N}$, where $X^{(i)}, Y^{(i)} \in [0, 1]^{d \times L}$. Assume the corresponding embedding sequences $Z^{(1)}, \ldots, Z^{(N)}$ are generated from a $C$-Lipschitz function. Then, there exists a single-layer, single-head attention transformer $\tau \in \mathcal{T}_B^{1,1,r}$ with $r = \mathcal{O}\left((1/\epsilon)^{d(L_p+L)}\right)$ and a soft-prompt $P \in \mathbb{R}^{d \times L_p}$ such that,

$$\left\| \tau([P, Z^{(i)}])_{:, L_p} - Y^{(i)} \right\|_\alpha \leq \epsilon, \quad \text{for any } i \in [N],$$

where $L_p \geq L\lambda$ with $\lambda = \left(2\epsilon^{-1}C(dL)^{1/\alpha}\right)^{dL}$.

*Proof Sketch.* We first find the underlying sequence-to-sequence function of the dataset $S$, denoted by $f_{\text{seq2seq}}^\star : [0, 1]^{d \times L} \mapsto [0, 1]^{d \times L}$, such that for any $i \in [N]$, $f_{\text{seq2seq}}^\star\left(Z^{(i)}\right) = Y^{(i)}$. Next, we complete the proof by utilizing the results of Theorem 2.4 to construct a transformer $\tau \in \mathcal{T}_B^{1,1,r}$ that is capable of approximating $f_{\text{seq2seq}}^\star$ through prompt tuning. Please see Appendix H.1 for a detailed proof. $\square$

**Remark 2.3.** Theorem 2.5 shows that a simple transformer is capable of memorizing any dataset through prompt tuning, when configured carefully. In contrast, (Wang et al., 2023a, Theorem 3) is limited to datasets with only two tokens per example and defines memorization as memorizing only the last token. Additionally, we provide a lower bound on the prompt sequence length required to memorize any dataset, based on its dimensions and the desired accuracy.

**Remark 2.4.** In (Wang et al., 2023a, Theorem 2), the authors construct a dataset and prove it to be unmemorizable by prompt tuning a transformer. However, their analysis differs from ours as it assumes full-rank self-attention weight matrices and a specific feed-forward layer structure. Their specialized dataset design relies on the invertibility of the weight matrices and a weak feed-forward layer, preventing the transformer from mapping contextual embeddings to the correct labels. We discuss these limitations in the expressive power of prompt tuning in Appendix J. In contrast, we prove that a transformer with single-layer self-attention and weight matrices of any rank is capable of achieving memorization through prompt tuning.

## 3 COMPUTATIONAL LIMITS OF PROMPT TUNING

We analyze the computational limits of inference of prompt tuning Problem 1 using fine-grained complexity theory. Specifically, recall that $X_p = [P, X] \in \mathbb{R}^{d \times (L_p+L)}$ with $Q_p = W_Q X_p \in \mathbb{R}^{d \times (L_p+L)}$, $K_p = W_K X_p \in \mathbb{R}^{d \times (L_p+L)}$, and $V_p = W_K X_p \in \mathbb{R}^{d \times (L_p+L)}$. We study approximate prompt tuning inference with precision guarantees under $\delta_F = 1/\text{poly}(L_p + L)$.

**Problem 1** (Approximate Prompt Tuning Inference APTI). Let $\delta_F > 0$ and $B > 0$. Given $Q_p, K_p, V_p \in \mathbb{R}^{d \times (L+L_p)}$ with guarantees that $\max\{\|Q_p\|_{\max}, \|K_p\|_{\max}, \|V_p\|_{\max}\} \leq B$, we aim to study an approximation problem $\text{APTI}(d, L, L_p, B, \delta_F)$, aiming to approximate $V_p \text{Softmax}\left(K_p^\intercal Q_p\right)$ with a matrix $\widetilde{Z}$ such that

$$\|\widetilde{Z} - V_p \text{Softmax}\left(K_p^\intercal Q_p\right)\|_{\max} \leq \delta_F,$$

Here, for a matrix $M \in \mathbb{R}^{a \times b}$, we write $\|M\|_{\max} \coloneqq \max_{i,j} |M_{i,j}|$.

### 3.1 PRELIMINARIES: STRONG EXPONENTIAL TIME HYPOTHESIS (SETH)

Our hardness results are built on a common conjecture. Impagliazzo and Paturi (2001) introduce the Strong Exponential Time Hypothesis (SETH) as a stronger form of the P $\neq$ NP conjecture. It suggests

that our current best `SAT` algorithms are optimal and is a popular conjecture for proving fine-grained lower bounds for a wide variety of algorithmic problems (Cygan et al., 2016; Williams, 2018).

**Hypothesis 1** (SETH). For every $\epsilon > 0$, there is a positive integer $k \geq 3$ such that $k$-`SAT` on formulas with $n$ variables cannot be solved in $\mathcal{O}(2^{(1-\epsilon)n})$ time, even by a randomized algorithm.

Below, we rely on SETH to facilitate the fine-grained reduction for lower bound result (Theorem 3.1).

### 3.2 Efficiency Criterion for Prompt Tuning Inference

We answer Question 2 affirmatively by identifying a phase transition behavior in the efficiency of all possible algorithms for Prompt Tuning Inference problem APTI (Problem 1), based on on the norm of $Q_p = W_Q X_p$, $K_p = W_K X_p$, and $V_p = W_V X_p$ with $X_p = [P, X] \in \mathbb{R}^{d \times (L_p + L)}$.

**Theorem 3.1** (Norm-Based Efficiency Phase Transition). Let $\|Q_p\|_{\max} \leq B$, $\|K_p\|_{\max} \leq B$ and $\|V_p\|_{\max} \leq B$ with $B = \mathcal{O}(\sqrt{\log(L_p + L)})$. Assuming Hypothesis 1, for every $q > 0$, there are constants $C, C_a, C_b > 0$ such that: there is no $\mathcal{O}((L_p + L)^{2-q})$-time (sub-quadratic) algorithm for the problem APTI$(L, L_p, d = C \log(L_p + L), B = C_b \sqrt{\log(L_p + L)}, \delta_F = (L_p + L)^{-C_a})$.

*Proof Sketch.* Our proof strategy involves connecting APIT to the hardness of attention inference (ATTC in (Alman and Song, 2023)) via a straightforward reduction. We achieve this by establishing a correspondence between APIT and ATTC, then applying a reduction with tighter error bounds using prompt tuning imputation (i.e., $\left|[\cdot]:, L_p :\right|_{\max} \leq \|\cdot\|_{\max}$). See Appendix I.1 for a detailed proof.  □

**Remark 3.1.** Theorem 3.1 suggests an efficiency threshold for the upper bound of $\|Q_p\|_{\max}$, $\|K_p\|_{\max}$, $\|V_p\|_{\max}$: $B = \mathcal{O}(\sqrt{\log(L_p + L)})$. Only below this threshold are efficient algorithms for Problem 1 possible , i.e. solving APIT in $(L_p + L)^{2-\Omega(1)}$ (sub-quadratic) time is possible.

### 3.3 Prompt Tuning Can Be as Fast as Almost-Linear Time

We answer Question 3 affirmatively by proving the existence of almost-linear time efficient algorithms for Prompt Tuning Inference problem APTI (Problem 1) based on low-rank approximation.

**Theorem 3.2** (Almost-Linear Prompt Tuning Inference). The prompt tuning inference problem APTI$(L, L_p, d = \mathcal{O}(\log(L_p + L)), B = o(\sqrt{\log(L_p + L)}), \delta_F = 1/\text{poly}(L_p + L))$ can be solved in time $\mathcal{T}_{\mathrm{mat}}((L_p + L), (L_p + L)^{o(1)}, d) = (L_p + L)^{1+o(1)}$.

*Proof Sketch.* We prove this using low-degree polynomial approximation of transformer attention. Consider a matrix $A \in \mathbb{R}^{p \times q}$ and a function $f : \mathbb{R} \to \mathbb{R}$. We define $f(A) : \mathbb{R}^{p \times q} \to \mathbb{R}^{p \times q}$ as the matrix obtained by applying $f$ to each entry of $A$. The goal of the polynomial method is to identify a low-rank approximation of $f(A)$. This method is effective if $A$ has a low rank and $f$ can be approximated by a low-degree polynomial, allowing $f(A)$ to also be represented as a low-rank matrix. This low-rank approximation can be efficiently computed in almost linear time via low-rank decomposition (Li et al., 2025; Alman and Yu, 2024; Hu et al., 2024b; Alman and Song, 2023; Aggarwal and Alman, 2022). Alman and Song (2023) provide bounds on the polynomial degrees necessary for approximating softmax attention with low rank. Utilizing these results and the structural properties of prompt tuning imputation (i.e., $\left\|[\cdot]_{:,L_p:}\right\|_{\max} \leq \|\cdot\|_{\max}$), we construct a low-rank approximation for the prompt tuning inference problem APTI. See Appendix I.2 for a proof.  □

Theorem 3.2 provides a formal example of the efficient criterion Theorem 3.1 for APTI using low-rank approximation within a controllable approximation error. This is applicable under Theorem 3.1 when the efficiency criterion is met. Specifically, to achieve nearly-linear $(L_p + L)^{1+o(1)}$ time prompt tuning inference with bounded error $\epsilon = 1/\text{poly}(L_p + L)$, we require $B = o(\sqrt{\log(L_p + L)})$.

**Concise Summary.** We study the statistical and computational limits of prompt-tuning transformers and identify key necessary conditions for designing expressive and efficient prompt-tuning methods. Due to page limits, we defer concluding remarks and practical implications to Appendix A, related work and limitations to Appendix B, and additional results on sequence-to-sequence approximation with one-layer, one-head attention using generic weight matrices — an extension of (Kajitsuka and Sato, 2024) to a more generic setting — to Appendix C.

## ACKNOWLEDGMENTS

JH would like to thank Tokio Kajitsuka, Jigyasa Kumari, Mimi Gallagher, Sara Sanchez, Dino Feng and Andrew Chen for enlightening discussions; Weimin Wu, Yi-Chen Lee, Yu-Chao Huang, Maojiang Su, and Hude Liu for collaborations on related topics; and the Red Maple Family for their support. The authors also thank the authors of (Wang et al., 2023a) for their clarifications, and thank the anonymous reviewers and program chairs for constructive comments.

JH is partially supported by the Walter P. Murphy Fellowship. HL is partially supported by NIH R01LM1372201, AbbVie and Dolby. The content is solely the responsibility of the authors and does not necessarily represent the official views of the funding agencies.

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

# Appendix

## A    DISCUSSION AND CONCLUDING REMARKS

We study the fundamental limits of prompt tuning transformer-based pretrained models (i.e., foundation models) in two aspects: statistical and computational. Statistically, we show the universality of prompt tuning transformer models with 1-head, 1-layer attention layers (Theorem 2.3 and Theorem 2.4). Recall that $d$ is the token dimension, $L$ is the input sequence length, $L_p$ is the soft-prompt length, and $\epsilon$ is the approximation error. Our results significantly relax previous requirements for thick layers, reducing from $O((L_p + L)(1/\epsilon)^d)$ layers to 1 attention layer, and from $\mathcal{O}((1/\epsilon)^{d(L_p+L)})$ layers to 2 FFN layers for prompt tuning universality. In addition, we prove the memorization capacity of prompt tuning and derive an exponential-in-$dL$ and -in-$1/\epsilon$ lower bound on required soft-prompt tokens (Theorem 2.5). Different from (Wang et al., 2023a) where the analysis of capacity is solely on datasets of two-token sequences and focuses on only memorizing the last token, we demonstrate a complete memorization of prompt tuning on any general dataset. Computationally, we establish an efficient criterion of all possible prompt tuning inference for the norm of soft-prompt induced keys and queries (Theorem 3.1). In addition, we showcase our theory by proving the existence of nearly-linear time prompt tuning algorithms (Theorem 3.2).

**Practical Implications from Statistical Limits (Section 2).** We analyze the universality of prompt tuning transformers with minimal structures and its memorization capacity on general datasets.

- **Universality (Theorem 2.4).** Our results show that the universality of prompt tuning pretrained transformer is achievable on as simple as a single-layer, single-head attention transformer. This demonstrates that universality in prompt-tuning isn't limited to large, complex foundation models.

- **Width-Depth Tradeoff (Section 2.4).** Our results highlight a trade-off in the design choices for the depth and width of FFN (MLP) layers: (i) $\mathcal{O}((1/\epsilon)^{d(L+L_p)})$ FFN layers of width 4 or (ii) 2 FFN layers of width $\mathcal{O}((1/\epsilon)^{d(L+L_p)})$. In practice, (i) and (ii) differ in memory usage, parallelization, and optimization preferences, leading to distinct application scenarios.

- **Memorization (Section 2.5).** Our memorization results apply to general datasets, whereas prior results are limited to specialized cases. This makes our results go beyond specialized theoretical analysis and align more with practical applications with a suggested *long* soft-prompt length.

**Practical Implications from Computational Limits (Section 3).** We analyze the $\mathcal{O}(L^2)$ bottleneck of prompt tuning transformers and provides useful guidance for designing efficient prompt tuning (approximation) methods with precision guarantees. Let $Q_p = W_Q X_p$, $K_p = W_K X_p$, and $V_p = W_V X_p$ with $X_p = [P, X] \in \mathbb{R}^{d \times (L_p + L)}$. Here $L$ and $L_p$ are the input and soft-prompt length.

- **Self- and Cross-Attention.** Our computational results apply to both self-attention and cross-attention prompt tuning. This is because the norm bound conditions depend on $\max\{|Q_p|, |K_p|, |V_p|\}$, which are valid for both self- and cross-attention inputs.

- **Necessary Conditions for Subquadratic Prompt Tuning (Theorem 3.1).** Our result suggests proper normalization on soft-prompt and weight matrices are required to ensure subquadratic prompt tuning inference, i.e., $\max\{\|Q_p\|_{\max}, \|K_p\|_{\max}, \|V_p\|_{\max}\} \leq \mathcal{O}(\sqrt{\log(L_p + L)})$.

- **Necessary Conditions for Almost Linear Time Prompt Tuning (Theorem 3.2).** Our result suggests more strict normalization on soft-prompt and weight matrices are required to ensure almost linear time prompt tuning inference, i.e., $\max\{\|Q_p\|_{\max}, \|K_p\|_{\max}, \|V_p\|_{\max}\} \leq o(\sqrt{\log(L_p + L)})$.

Suitable normalizations for the above can be implemented using pre-activation layer normalization (Xiong et al., 2020; Wang et al., 2019) to control $\|X_p\|_{\max}$, or outlier-free attention activation functions (Hu et al., 2024a) to control $\|W_K\|_{\max}, \|W_Q\|_{\max}, \|W_V\|_{\max}$.

# B   RELATED WORKS, LIMITATIONS AND BROADER IMPACT

## B.1   RELATED WORKS

**Context-based Fine-tuning and Soft-prompt Tuning.**   Recently, resource-efficient fine-tuning strategies (Ding et al., 2023; 2022), such as LoRA (Pan et al., 2024; Hayou et al., 2024; Hu et al., 2025; 2022), emerge as powerful alternatives to conventional full fine-tuning. In contrast, context-based fine-tuning techniques, like hard-prompt tuning (Wen et al., 2024), in-context learning (Xu et al., 2024; Shi et al., 2024; Wei et al., 2023; Dong et al., 2022; Brown et al., 2020), and prefix-tuning (Liang et al., 2024; Li and Liang, 2021), adapt pretrained models to specific tasks without modifying underlying model parameters (Brown et al., 2020; Li and Liang, 2021; Liu et al., 2022). One of the most effective methods is soft-prompt tuning (Liu et al., 2023), which uses real-valued embeddings to guide model outputs. This approach leverages the expressive power of continuous spaces to fine-tune responses, avoiding extensive parameter updates and making it both efficient and less resource-intensive than traditional fine-tuning methods (Lester et al., 2021; Liu et al., 2022).

**Universality of Transformers.**   The universality of transformers refers to their ability to serve as universal approximators. This means that transformers theoretically model any sequence-to-sequence function to a desired degree of accuracy. Yun et al. (2020) show that transformers universally approximate sequence-to-sequence functions by stacking numerous layers of feed-forward functions and self-attention functions. In a different approach, Jiang and Li (2023) affirm the universality of transformers by utilizing the Kolmogorov-Albert representation Theorem. Furthermore, Alberti et al. (2023) demonstrate universal approximation for architectures that incorporate non-standard attention mechanisms. Most recently, Kajitsuka and Sato (2024) show that transformers with one self-attention layer are a universal approximator. Of independent interest, recent work by Havrilla and Liao (2024) examines the generalization and approximation of transformers under Hölder smoothness and low-dimensional subspace assumptions.

Our paper is motivated by and builds upon works of Yun et al. (2020); Kajitsuka and Sato (2024). Specifically, we study the universality of prompt tuning transformers using the analysis framework by Yun et al. (2020). Furthermore, we extend the contextual mapping property of 1-rank attention by Kajitsuka and Sato (2024) to any-rank attention. This allows us to establish the universality of prompt tuning transformers in the simplest configuration — single-layer, single-head attention.

**Analysis on Prompt Tuning.**   Prompt tuning has been successful in various applications. However, the theoretical analysis of it is less developed. Petrov et al. (2023) discuss different kinds of context-based learning, and experimentally show when prompt tuning is successful in adapting to new tasks. In this work, we tackle the prompt tuning problem from a theoretical perspective. Oymak et al. (2023) identify the cases where the attention layer with prompt tuning is more expressive than a self-attention layer. They utilize prompt tokens dependent on weight matrices. In addition, they require weight matrices to be full rank. Conversely, our study explores the expressive power of prompt tuning under more general conditions, without relying on such assumptions. Wang et al. (2023a) show the universality of prompt tuning transformers with an increasing number of layers in proportion to the input data dimension and the quantization grid. Petrov et al. (2024) prove the universality of prompt tuning on transformers with the number of layers linear in the input sequence length. Liang et al. (2024) study the convergence guarantee for prompt tuning with ultra-long soft-prompt in the Neural Tangent Kernel region (NTK). On the other hand, we focus on approximation and computation properties of prompt tuning transformers with single-layer-single-head self-attention.

Our work builds on (Wang et al., 2023a), as both quantize the input and output domains of sequence-to-sequence functions to establish universal approximation. However, this work differs in three aspects. First, while Wang et al. (2023a) require transformers with a number of layers proportional to the input data dimension and two attention heads, we demonstrate the universality of prompt tuning with the simplest transformer: a single-layer, single-head attention transformer. Second, we present the first study to show complete data memorization through prompt tuning, providing a lower bound on the required soft-prompt tokens for a single-layer, single-head transformer to memorize any dataset. Lastly, we provide the first comprehensive analysis of the computational limits, proving the existence of nearly-linear time prompt tuning inference algorithms.

**Memory Capacity of Transformer.**   Even though there has not been much analysis on the memory capacity of prompt tuning, there are many works on the memorization of transformers itself. Kim et al. (2022) prove $2n$ self-attention blocks are sufficient for the memorization of finite samples, where $n$ denotes the sequence length of data. Mahdavi et al. (2023) show that a multi-head-attention with $h$ heads is able to memorize $\mathcal{O}(hn)$ examples. Kajitsuka and Sato (2025; 2024) prove the memorization capacity for a single-layer transformer. They demonstrate that for $N$ sequence-to-sequence data examples, each with dimension $d \times n$, the number of parameters required for memorization is $\mathcal{O}(nNd + d^2)$. Another area of research introduces a distinct type of memory capacity for transformers by linking transformer attention mechanisms with dense associative memory models, specifically modern Hopfield networks (Bietti et al., 2024; Hu et al., 2024a;b;c; 2023; Wu et al., 2024a;b; Ramsauer et al., 2020).

The closest work to ours is (Wang et al., 2023a), where they discuss the required prompt tokens for prompt tuning on memorizing a special sequence-to-sequence dataset. In the special dataset, the examples are required to have exactly two tokens each. In addition, they discussed the memorization of only the last token of each data sequence. In contrast, we provide the first analysis on general cases where prompt tuning memorizes the whole sequence for each example in a general dataset with no assumption on the data. In addition, our work is the first to provide the lower bound on the required soft-prompt tokens for memorization.

## B.2   LIMITATIONS AND BROADER IMPACT

**Limitations.**   By the formal nature of this work, our results do not lead to practical implementations. However, we anticipate that our findings will offer valuable insights for future prompt tuning methods.

Moreover, our memorization findings indicate an exponential dependence on the data sequence length $L$ and approximation precision $1/\epsilon$. Although resource-efficient, this exponential dependence implies that prompt tuning pretrained transformers may not be an optimal method for encoding or memorizing information. This leads to two fundamental possibilities:

- While not investigated in this work, there may be an information-theoretic lower bound that highlights the limitations of our current memory capacity results for prompt tuning.

- If we prove that no upper bound can match this lower bound, it would reveal a fundamental limitation of prompt tuning: it is not an information-efficient learning method (or machine).

We plan to investigate these issues in future work.

**Broader Impact.**   This theoretical work aims to shed light on the foundations of large transformer-based models and is not expected to have negative social impacts.

## C  ADDITIONAL THEORETICAL RESULTS: UNIVERSALITY OF TRANSFORMERS WITH 1-LAYER, 1-HEAD, ANY-RANK SELF-ATTENTION

Lemma 2.2 shows that any-rank single-layer, single-head attention is contextual mapping. A direct consequence is the universality of transformers with 1-layer, 1-head, *any-rank*[4] self-attention following Kajitsuka and Sato (2024). We believe this result may be of independent interest.

**Theorem C.1.** Let $1 \leq \alpha < \infty$ and $\epsilon > 0$. For any $f_{\text{seq2seq}} \in \mathcal{F}_C$, there exists a transformer with single-layer, single-head attention and any-rank weight matrices $\tau \in \mathcal{T}_A^{1,1,4}$ (or $\tau \in \mathcal{T}_B^{1,1,r}$ with $r = \mathcal{O}((1/\epsilon)^{dL})$) with positional embedding $E \in \mathbb{R}^{d \times L}$ such that $d_\alpha(\tau, f_{\text{seq2seq}}) \leq \epsilon$.

*Proof Sketch.* This proof is inspired by (Yun et al., 2020) and similar to the proof of Lemma F.2.

There are mainly three steps:

1. Given an input data $X \in \mathbb{R}^{d \times L}$, we first apply positional encoding $E$, which is given as

$$E = \begin{bmatrix} 0 & 1 & 2 & \dots & L-1 \\ 0 & 1 & 2 & \dots & L-1 \\ \vdots & \vdots & \vdots & \ddots & \vdots \\ 0 & 1 & 2 & \dots & L-1 \end{bmatrix}.$$

   Then a series of feed-forward layers in the modified Transformer network quantizes $X + E$ to a quantized sequence $M \in \bar{\mathcal{G}}_{\delta,L}$. Here, we define the grid

$$\bar{\mathcal{G}}_{\delta,L} := [0 : \delta : 1 - \delta]^d \times [1 : \delta : 2 - \delta]^d \times \cdots \times [L-1 : \delta : L - \delta]^d,$$

   where $[a : \varepsilon : b] := \{a, a + \varepsilon, a + 2\varepsilon, \dots, b - \varepsilon, b\}$. Note that with the positional encoding, our contextual mapping through self-attention won't be limited to permutation equivalent functions.

2. Next, by utilizing Lemma 2.2, the single self-attention layer in the modified transformer takes the input $M$ and implements a contextual mapping $q : \mathbb{R}^{d \times L} \mapsto \mathbb{R}^{d \times L}$.

3. Finally, a series of feed-forward layers map elements of the contextual embedding $q(M)$ to the desired output value of $f_{\text{seq2seq}}(X)$.

We remark that Step 2 distinguishes us from prior works by utilizing the fact that any-rank attention is a contextual mapping Lemma 2.2. This improves the result of (Kajitsuka and Sato, 2024), which requires an attention layer of rank one. ☐

*Proof of Theorem C.1.* First, we apply the positional encoding $E \in \mathbb{R}^{d \times L}$ on the input sequence $X \in \mathbb{R}^{d \times L}$, so that each token has a different domain. The positional encoding $E$ is given as

$$E = \begin{bmatrix} 0 & 1 & 2 & \dots & L-1 \\ 0 & 1 & 2 & \dots & L-1 \\ \vdots & \vdots & \vdots & \ddots & \vdots \\ 0 & 1 & 2 & \dots & L-1 \end{bmatrix}.$$

We next use feed-forward layers $f^{(\text{FF})}$ to implement a quantization map to quantize the input $X + E$ in to its discrete version $M \in \bar{\mathcal{G}}_{\delta,L}$. The grid $\bar{\mathcal{G}}_{\delta,L}$ is defined as

$$\bar{\mathcal{G}}_{\delta,L} := [0 : \delta : 1 - \delta]^d \times [1 : \delta : 2 - \delta]^d \times \cdots \times [L-1 : \delta : L - \delta]^d,$$

---

[4]By any-rank attention, we refer to an attention head with generic weights of arbitrary rank.

where $[a : \varepsilon : b] := \{a, a + \varepsilon, a + 2\varepsilon, \ldots, b - \varepsilon, b\}$. Note that the first column of $X + E$ is in $[0, 1]^d$, the second is in $[1, 2]^d$, and so on. Here, we write the quantization mapping as

$$[0, 1]^d \times \cdots \times [L - 1, L]^d \mapsto [0 : \delta : 1 - \delta]^d \times \cdots \times [L - 1 : \delta : L - \delta]^d.$$

Inspired by the construction recipe by (Yun et al., 2020), this task is realized by $dL/\delta$ feed-forward layers. We add $dL/\delta$ layers of $f^{(\mathrm{FF})}$ with the following form, for $k = 0, \delta, \ldots, L - \delta$ and $i = 1, \ldots, d$ :

$$Z \mapsto Z + e^{(i)} \phi \left( \left( e^{(i)} \right)^T Z - k\delta \mathbf{1}_n^T \right), \phi(t) = \begin{cases} 0 & t < 0 \text{ or } t \geq \delta \\ -t + 1 & 0 \leq t < \delta \end{cases}, \tag{C.1}$$

where $e^{(1)} = (1, 0, 0, ..., 0) \in \mathbb{R}^d$ and $\phi(t) \in \Phi$ is an entrywise function, where the set of activation functions $\Phi$ consists of all piece-wise linear functions with at least one piece being constant and at most three pieces. Furthermore, any activation function $\phi \in \Phi$ is realized by 4 MLP neurons. Each layer in the form of (C.1) quantizes $X_{i,:}$ (the $i$-th row) in $[k\delta, k\delta + \delta)$ to $k\delta$. We denote output after the feed-forward layers as $M \in \bar{\mathcal{G}}_{\delta, L}$.

Next, in order to utilize Lemma 2.2, we observe that the quantized output $M$ from the previous step has no duplicate tokens, since each column has a unique domain. Also, we see that $M$ is token-wise $\left( \sqrt{d}, \sqrt{d}(L - \delta), \sqrt{d}\delta \right)$-separated. This is easily observed as we have, for any $k, l \in L$,

$$\|M_{:,k}\| > \sqrt{d},$$
$$\|M_{:,k}\| < \sqrt{d}(L - \delta),$$
$$\|M_{:,k} - M_{:,l}\| > \sqrt{d}\delta.$$

As a result, with Lemma 2.2, we arrive at a $(\Gamma, \Delta)$-contextual mapping $q : \mathbb{R}^{d \times L} \mapsto \mathbb{R}^{d \times L}$ where

$$\Gamma = \sqrt{d}(L - \delta) + \frac{\sqrt{d}\delta}{4} = \sqrt{d}(L - \frac{3\delta}{4}),$$
$$\Delta = \exp\left(-5|\mathcal{V}|^4 d \ln(n) L^2/\delta\right).$$

Now we have successfully mapped each input sequence $X + E$ to unique contextual embeddings $q(M) \in \mathbb{R}^{d \times L}$. We next associate each unique embeddings to a corresponding expected output of $f_{\mathrm{seq2seq}}(X)$.

We use feed-forward layers to map each token of $q(M)$ to the desired $[0, 1]^d$. As in (Yun et al., 2020, C.3), with a method similar to (C.1), we need one layer for each unique value of $q(M)$ for each $M \in \bar{\mathcal{G}}_{\delta, L}$. There are in total $(1/\delta)^{dL}$ possibilities of $M$ and each corresponds to some output of $h_{\mathrm{seq2seq}}([P, \cdot])$. Since we only focus on the last $L$ tokens of output, we require $\mathcal{O}\left(L(1/\delta)^{dL}\right) = \mathcal{O}\left(\delta^{-dL}\right)$ layers to map these distinct numbers to expected outputs.

This completes the proof for transformers $\tau \in \mathcal{T}_A^{1,1,4}$. The proof for transformers $\tau \in \mathcal{T}_B^{1,1,r}$ follows the same recipe, and we refer to the proof of Lemma G.2 for details. $\qquad \square$

## D BACKGROUND: BOLTZMANN OPERATOR AND ATTENTION MECHANISM

Here, we present some auxiliary definitions and lemmas to prepare our proofs.

To demonstrate that a single-layer self-attention mechanism with matrices of any rank acts as a contextual map, we follow (Kajitsuka and Sato, 2024; Asadi and Littman, 2017). Specifically, we utilize the connection between self-attention mechanisms and the Boltzmann operator Boltz.

In this section, we introduce non-original but still necessary auxiliary lemmas. We defer the proofs to Appendix K for completeness. Below, we start with the definition of the Boltzmann operator Boltz.

**Boltzmann Operator.** Following (Asadi and Littman, 2017; Kajitsuka and Sato, 2024), we associate the $\mathrm{Softmax}$ function with the Boltzmann operator Boltz defined below:

**Definition D.1** (Softmax and Boltz). Let $z = (z_1, \ldots, z_n) \in \mathbb{R}^n$ and the function $\mathrm{Softmax} : \mathbb{R}^n \to \mathbb{R}^n$ operate element-wise: $\mathrm{Softmax}(z)_i = \exp(z_i) / \sum_{j=1}^{n} \exp(z_j)$. Denote $p = (p_1, \ldots, p_n) := \mathrm{Softmax}(z) \in \mathbb{R}^n$ with $p_i = \mathrm{Softmax}(z)_i$. The Boltzmann operator $\mathrm{Boltz} : \mathbb{R}^n \mapsto \mathbb{R}$ is defined as

$$\mathrm{Boltz}(z) = z^\top \mathrm{Softmax}(z) = z^\top p = \sum_{i=1}^{n} z_i p_i. \tag{D.1}$$

To give a brief overview to this section, in Appendix D.1, we first introduced the essential properties of Boltz. Next, in Appendix D.2, we utilized these properties to further illustrate the Boltz operator's ability to maintain the separation between inputs.

In the following, we present the essential properties of Boltz in Appendix D.1.

### D.1 ESSENTIAL PROPERTIES OF BOLTZMANN OPERATOR

Before characterizing the Boltzmann operator Boltz, we introduce some useful functions and essential properties of Boltz from (Kajitsuka and Sato, 2024) to facilitate our proofs.

We first recall the partition function and the (Gibbs) entropy function from statistical physics,

$$\mathcal{Z}(z) = \sum_{i=1}^{n} \exp(z_i), \quad \text{and} \quad \mathcal{S}(p) = -\sum_{i=1}^{n} p_i \ln(p_i). \tag{D.2}$$

Then, the next lemma presents the relation between the Boltzmann operator Boltz, partition function $\mathcal{Z}$ and entropy $\mathcal{S}$.

**Lemma D.1** (Boltz, $\mathcal{Z}$ and $\mathcal{S}$). With the definitions given above and a vector $z = (z_1, \ldots, z_n) \in \mathbb{R}^n$, the Boltzmann operator Boltz also takes the form

$$\mathrm{Boltz}(z) = -\mathcal{S}(p) + \ln \mathcal{Z}(z).$$

*Proof.* See Appendix K.1 for a detailed proof. $\qquad \square$

Next, we recall that Boltz decreases monotonically when the maximum entry is sufficiently distant from the other entries.

**Lemma D.2** (Monotonically Decrease, Lemma 4 of (Kajitsuka and Sato, 2024)). Given a vector $z = (z_1, \ldots, z_n) \in \mathbb{R}^n$, the Boltzmann operator $\mathrm{Boltz}(z)$ monotonically decreases in the direction of $z_i$ when $\max_{j \in [n]} z_j - z_i > \ln n + 1$, that is,

$$\frac{\partial}{\partial z_i} \mathrm{Boltz}(z) = p_i \left(1 + \ln p_i + \mathcal{S}(p)\right) < 0.$$

*Proof.* See Appendix K.2 for a detailed proof. $\qquad \square$

The next lemma shows the concavity of $\mathrm{Boltz}$ when the max entry and the rest of the entries are distant enough.

**Lemma D.3** (Concave, Lemma 5 of (Kajitsuka and Sato, 2024)). Given a vector $z = (z_1, \ldots, z_n) \in \mathbb{R}^n$, the Boltzmann operator $\mathrm{Boltz}(z)$ is concave with respect to $z_i$ when $\max_{j \in [n]} z_j - z_i > \ln n + 3$, that is,

$$\frac{\partial^2}{\partial z_i^2} \mathrm{Boltz}(z) < 0.$$

*Proof.* See Appendix K.3 for a detailed proof. □

To ease the later calculation and better understand the characteristics of the Boltzmann operator, the next lemma shows the bounds of the output of $\mathrm{Boltz}$ when given inputs with certain constraints.

**Lemma D.4** (Lower Bound of Boltz with $(\delta)$-Separated Input). Given a tokenwise $(\delta)$-separated vector $z = (z_1, \ldots, z_n) \in \mathbb{R}^n$ with $n \geq 2$ and $\delta > \ln n + 1$. Also let the entries of $z$ be sorted in a decreasing order with no duplicate entry, that is, for any $i, j \in [n], i < j$ ,

$$z_i - z_j > \delta.$$

Then Boltzmann operator $\mathrm{Boltz}(z)$ is lower bounded by

$$\mathrm{Boltz}(z) > \mathrm{Boltz}(z')$$

where $z' = (z_1, z_1 - \delta, \ldots, z_1 - \delta)$ .

*Proof.* See Appendix K.4 for a detailed proof. □

Next, we present another property of $\mathrm{Boltz}$, which states that when two vectors share the same first $n$ entries but differ in dimension, the output of $\mathrm{Boltz}$ for the lower-dimensional vector will be larger.

**Lemma D.5** (Boltz Value Comparison). Given two tokenwise $(\delta)$-separated vectors $z = (z_1, \ldots, z_n) \in \mathbb{R}^n$, $z' = (z'_1, \ldots, z'_m) \in \mathbb{R}^m$ with $m > n \geq 2$ and $\delta > \ln n + 1$. Also let the entries of $z, z'$ be sorted in a decreasing order with no duplicate entry. In addition, let the first $n$ entries of $z'$ be $z$ , that is,

$$(z'_1, \ldots, z'_n) = z.$$

Then, we have

$$\mathrm{Boltz}(z) > \mathrm{Boltz}(z').$$

*Proof.* See Appendix K.5 for a detailed proof. □

With a solid understanding of $\mathrm{Boltz}$ established, we leverage its properties to demonstrate that $\mathrm{Boltz}$ preserves the separation between two distinct input tokens.

### D.2 Distance Preservation of Boltzmann Operator

In this section, by utilizing the above properties, we show that when given well separated input tokens, the output of $\mathrm{Boltz}$ is also separated. We start by examining specific cases with more stringent constraints on the inputs, and subsequently expand our discussion to more general scenarios.

We first discuss the case when the two input vector has no same entries.

**Lemma D.6** (Input of Complete Different Entries, Lemma 7 of (Kajitsuka and Sato, 2024)). Let $n \geq 2$ and consider two vectors $a = (a_1, \ldots, a_n)$, $b = (b_1, \ldots, b_n) \in \mathbb{R}^n$. In addition, assume the following conditions hold:

- Decreasing order entries: The entries of $a$ and $b$ are sorted in strictly decreasing order,

$$a_1 > a_2 > \cdots > a_n \quad \text{and} \quad b_1 > b_2 > \cdots > b_n.$$

- Tokenwise ($\delta$)-separateness: For any $i, j \in [n]$, if $a_i \neq b_j$

$$|a_i - b_j| > \delta,$$

and if $i < j$,

$$a_i - a_j > \delta,$$
$$b_i - b_j > \delta,$$

where $\delta \geq 4 \ln n$.

- Initial dominance: The largest element in a is strictly greater than the largest element in b,

$$a_1 > b_1.$$

Under these assumptions, we have

$$\mathrm{Boltz}(a) - \mathrm{Boltz}(b) > (\ln n)^2 e^{-(a_1 - b_1)}.$$

*Proof Sketch.* To find the lower bound of $\mathrm{Boltz}(a) - \mathrm{Boltz}(b)$, we first find some lower bound of $\mathrm{Boltz}(a)$ and some upper bound of $\mathrm{Boltz}(b)$ that ease the computation. From Lemma D.4, we have that $\mathrm{Boltz}(a) > \mathrm{Boltz}(a')$ where $a' = (a_1, a_1 - \delta, \ldots, a_1 - \delta)$. In addition, by definition of Boltz the upper bound of $\mathrm{Boltz}(b)$ is $\mathrm{Boltz}(b) \leq b_1$. As a result, we evaluate $\mathrm{Boltz}(a') - b_1$ to complete the proof. See Appendix K.6 for a detailed proof. □

Next, we show that when two inputs are different only by one last entry, their $\mathrm{Boltz}$ outputs are still different with a certain distance.

**Lemma D.7** (Input of One Entry Difference, Lemma 6 of (Kajitsuka and Sato, 2024)). Consider $n \geq 2$, and two vectors $a = (a_1, \ldots, a_{n-1}, a_n)$, $b = (b_1, \ldots b_{n-1}, b_n) \in \mathbb{R}^n$. In addition, assume the following conditions hold:

- Identical first $n - 1$ entries: The first $n - 1$ entries of $a$ is the same as $b$,

$$a_i = b_i \forall i \in [n-1].$$

- Strict inequality for last entry: The last entry of $a$ is strictly greater than that of $b$,

$$a_n > b_n.$$

- Well separated: The last entry $a_n$ is sufficiently smaller than the maximum of the first $n - 1$ entries of $a$,

$$\max_{i \in [n-1]} a_i - a_n > \ln n + 3.$$

Then the difference of $\mathrm{Boltz}(a)$ between $\mathrm{Boltz}(b)$ is lower bounded as

$$\mathrm{Boltz}(b) - \mathrm{Boltz}(a) > (a_n - b_n)(\delta + a_n - b_n - \ln n - 1) \cdot \frac{e^{b_n}}{\sum_{i=1}^n e^{b_i}}.$$

*Proof.* See Appendix K.7 for a detailed proof. □

Now, we consider a more general case, where the top $k$ entries are the same.

**Lemma D.8** (Input of Matching Top $k$ Entries, Lemma 7 of (Kajitsuka and Sato, 2024)). Let $n \geq 2$ and consider two vectors $a = (a_1, \ldots, a_n)$, $b = (b_1, \ldots, b_n) \in \mathbb{R}^n$. In addition, assume the following conditions hold:

- Decreasing order entries: The entries of $a$ and $b$ are sorted in strictly decreasing order,

$$a_1 > a_2 > \cdots > a_n \quad \text{and} \quad b_1 > b_2 > \cdots > b_n.$$

- Tokenwise $(\delta)$-separateness: For any $i, j \in [n]$, if $a_i \neq b_j$

$$|a_i - b_j| > \delta,$$

and if $i < j$,

$$a_i - a_j > \delta,$$
$$b_i - b_j > \delta,$$

where $\delta \geq 4 \ln n$.

- Identical first $k$ entries: Let $a, b$ have the same top-$k$ entries for $k \in [n-1]$, which is

$$(a_1, \ldots, a_k) = (b_1, \ldots, b_k)$$

- $(k+1)$-th dominance: The largest element in a is strictly greater than the largest element in b,

$$a_{k+1} > b_{k+1}.$$

Under these assumptions, we have

$$|\text{Boltz}(a) - \text{Boltz}(b)| > \ln^2(n) \cdot e^{-(a_1 - b_{k+1})}.$$

*Proof Sketch.* As the top-$k$ entries of $a, b$ are the same, and all entries are $(\delta)$-separated while sorted in a decreasing order, when $a_{k+1} > b_{k+1}$, we have

$$\text{Boltz}(b) > \text{Boltz}(a).$$

To understand the intuition behind this, first recognize that Boltz calculates a weighted sum of elements, assigning higher weights to larger entries. Additionally, the total sum of all weights equals one. Consequently, when all entries are distinct and arranged in descending order, a larger $(k+1)$-th entry, shares more weight from the top $k$ greatest terms, compared to a smaller $(k+1)$-th entry. This results in a lower weighted sum.

Next, we compute the value of $\text{Boltz}(b) - \text{Boltz}(a)$. By Lemma D.5, we have that $\text{Boltz}(a)$ is upper bounded by $\text{Boltz}(a_{\text{up}})$, where

$$a_{\text{up}} = (a_1, a_2, \ldots, a_k, a_{k+1}).$$

Also, similar to Lemma D.4, $\text{Boltz}(b)$ is lower bounded by $\text{Boltz}(b_{\text{lo}})$, where

$$b_{\text{lo}} = (a_1, a_2, \ldots, a_k, b_{k+1}, b_{k+1}, \ldots, b_{k+1}).$$

Computing $\text{Boltz}(b_{\text{lo}}) - \text{Boltz}(a_{\text{up}})$ is easier than directly calculating $\text{Boltz}(b) - \text{Boltz}(a)$ as we are able to decompose $\text{Boltz}(b_{\text{lo}})$ and utilize Lemma D.7 to arrive at the final bound. See Appendix K.8 for a detailed proof. □

Finally, by utilizing the results above, we show that the Boltzmann operator is a mapping that projects input sequences to scalar values while preserving some distance.

**Lemma D.9** (Boltz Preserves Distance, Lemma 1 of (Kajitsuka and Sato, 2024)). Given $(\gamma, \delta)$-tokenwise separated vectors $z^{(1)}, \ldots, z^{(N)} \in \mathbb{R}^n$ with no duplicate entries in each vector, that is

$$z_s^{(i)} \neq z_t^{(i)},$$

where $i \in [N]$ and $s, t \in [n], s \neq t$. Also, let

$$\delta \geq 4 \ln n.$$

Then, the outputs of the Boltzmann operator are $(\gamma, \delta')$-separated:

$$\left| \mathrm{Boltz}\left(z^{(i)}\right) \right| \leq \gamma, \tag{D.3}$$

$$\left| \mathrm{Boltz}\left(z^{(i)}\right) - \mathrm{Boltz}\left(z^{(j)}\right) \right| > \delta' = \ln^2(n) \cdot e^{-2\gamma} \tag{D.4}$$

for all $i, j \in [N], i \neq j$.

*Proof.* See Appendix K.9 for a detailed proof. □

We have now established that the $\mathrm{Boltz}$ operator has the property of preserving the distances between inputs.

# E   PROOFS OF SECTION 2.2

In this section, by relating Softmax with Boltz, we show that the one layer of single head self-attention with weight matrices of any rank is a contextual mapping.

We first introduce a helper lemma.

**Lemma E.1** (Lemma 13 of (Park et al., 2021)). For any finite subset $\mathcal{X} \subset \mathbb{R}^d$, there exists at least one unit vector $u \in \mathbb{R}^d$ such that

$$\frac{1}{|\mathcal{X}|^2}\sqrt{\frac{8}{\pi d}}\|x - x'\| \leq |u^\top (x - x')| \leq \|x - x'\|$$

for any $x, x' \in \mathcal{X}$.

*Proof.* See Appendix K.10 for a detailed proof.   □

## E.1   PROOFS OF LEMMA 2.2

With Lemma E.1, we develop a method to configure weight matrices of a self-attention layer.

**Lemma E.2** (Construction of Weight Matrices). Given a dataset with a $(\gamma_{\min}, \gamma_{\max}, \epsilon)$-separated finite vocabulary $\mathcal{V} \subset \mathbb{R}^d$, there exist rank-$\rho$ weight matrices $W_K, W_Q \in \mathbb{R}^{s \times d}$ such that

$$\left|(W_K v_a)^\top (W_Q v_c) - (W_K v_b)^\top (W_Q v_c)\right| > \delta,$$

for any $\delta > 0$, any $\min(d, s) \geq \rho \geq 1$, and any $v_a, v_b, v_c \in \mathcal{V}$ with $v_a \neq v_b$. Specifically, the matrices are constructed as follows:

$$W_K = \sum_{i=1}^{\rho} p_i q_i^\top \in \mathbb{R}^{s \times d}, \quad W_Q = \sum_{j=1}^{\rho} p_j' q_j'^\top \in \mathbb{R}^{s \times d},$$

where $q_1, q_1' \in \mathbb{R}^d$ are unit vectors satisfying Lemma E.1, and $p_1, p_1' \in \mathbb{R}^s$ satisfy

$$\left|p_1^\top p_1'\right| \geq 5(|\mathcal{V}| + 1)^4 d \frac{\delta}{\epsilon \gamma_{\min}}.$$

*Proof of Lemma E.2.* We build our proof upon (Kajitsuka and Sato, 2024).

We start the proof by applying Lemma E.1 to $\mathcal{V} \cup \{0\}$. We obtain at least one unit vector $q \in \mathbb{R}^d$ such that for any $v_a, v_b \in \mathcal{V} \cup \{0\}$ and $v_a \neq v_b$, we have

$$\frac{1}{(|\mathcal{V}| + 1)^2 d^{0.5}}\|v_a - v_b\| \leq |q^\top (v_a - v_b)| \leq \|v_a - v_b\|.$$

By choosing $v_b = 0$, we have that for any $v_c \in \mathcal{V}$

$$\frac{1}{(|\mathcal{V}| + 1)^2 d^{0.5}}\|v_c\| \leq |q^\top v_c| \leq \|v_c\|. \tag{E.1}$$

For convenience, we denote the set of all unit vector $q$ that satisfies (E.1) as $\mathcal{Q}$, where

$$\mathcal{Q} := \left\{q \in \mathbb{R}^d \mid \frac{1}{(|\mathcal{V}| + 1)^2 d^{0.5}}\|v_c\| \leq |q^\top v_c| \leq \|v_c\|\right\}.$$

Next, we choose some arbitrary vector pairs $p_1, p_1' \in \mathbb{R}^s$ that satisfy

$$\left| p_1^\top p_1' \right| \geq (|\mathcal{V}| + 1)^4 d \frac{\delta}{\epsilon \gamma_{\min}}. \tag{E.2}$$

We construct the weight matrices by setting

$$W_K = \sum_{i=1}^{\rho} p_i q_i^\top \in \mathbb{R}^{s \times d},$$

$$W_Q = \sum_{j=1}^{\rho} p_j' q_j'^\top \in \mathbb{R}^{s \times d},$$

where $p_1, p_1'$ satisfies (E.2) and $q_1, q_1' \in \mathcal{Q}$. We arrive at

$$\left| (W_K v_a)^\top (W_Q v_c) - (W_K v_b)^\top (W_Q v_c) \right|$$

$$= \left| (v_a - v_b)^\top (W_K)^\top (W_Q v_c) \right|$$

$$= \left| (v_a - v_b)^\top \left( \sum_{i=1}^{\rho} q_i p_i^\top \right) \left( \sum_{j=1}^{\rho} p_j' q_j'^\top v_c \right) \right|$$

$$= \left| \left( \sum_{i=1}^{\rho} (v_a - v_b)^\top q_i p_i^\top \right) \left( \sum_{j=1}^{\rho} p_j' q_j'^\top v_c \right) \right|$$

$$= \left| \sum_{i=1}^{\rho} \sum_{j=1}^{\rho} (v_a - v_b)^\top q_i p_i^\top p_j' q_j'^\top v_c \right|$$

$$= \left| (v_a - v_b)^\top q_1 \right| \cdot \left| p_1^\top p_1' \right| \cdot \left| q_1'^\top v_c \right|$$

$$\qquad \text{(We choose } \left| p_1^\top p_1' \right| \text{ large enough so it's the dominating term in the sum.)}$$

$$\geq \frac{1}{(|\mathcal{V}| + 1)^2 d^{0.5}} \|v_a - v_b\| \cdot (|\mathcal{V}| + 1)^4 d \frac{\delta}{\epsilon \gamma_{\min}} \cdot \frac{1}{(|\mathcal{V}| + 1)^2 d^{0.5}} \|v_c\| \qquad \text{(By (E.1) and (E.2))}$$

$$> \delta. \qquad \text{(By } (\gamma_{\min}, \gamma_{\max}, \epsilon)\text{-separateness of } \mathcal{V})$$

This completes the proof. $\qquad \square$

Now we present the result showing that a softmax-based 1-layer attention block is a contextual mapping.

**Lemma E.3** (Lemma 2.2 Restated). Let $Z^{(1)}, \ldots, Z^{(N)} \in \mathbb{R}^{d \times L}$ be $(\gamma_{\min}, \gamma_{\max}, \epsilon)$-tokenwise separated embeddings, with the vocabulary set $\mathcal{V} = \bigcup_{i \in [N]} \mathcal{V}^{(i)} \subset \mathbb{R}^d$. Additionally, assume no duplicate word tokens in each sequence, i.e., $Z_{:,k}^{(i)} \neq Z_{:,l}^{(i)}$ for any $i \in [N]$ and $k, l \in [L]$. Then, there exists a 1-layer, single-head attention mechanism with weight matrices $W^{(O)} \in \mathbb{R}^{d \times s}$ and $W_V, W_K, W_Q \in \mathbb{R}^{s \times d}$ that serves as a $(\gamma, \delta)$-contextual mapping for the embeddings $Z^{(1)}, \ldots, Z^{(N)}$, where:

$$\gamma = \gamma_{\max} + \frac{\epsilon}{4} \quad \text{and} \quad \delta = \exp\left( -5\epsilon^{-1} |\mathcal{V}|^4 d \kappa \gamma_{\max} \log L \right), \quad \text{with} \quad \kappa := \gamma_{\max} / \gamma_{\min}.$$

**Remark E.1** (Comparing with Existing Works). In comparison with (Kajitsuka and Sato, 2024), they provided a proof for the case where all self-attention weight matrices $W_V, W_K, W_Q \in \mathbb{R}^{s \times d}$ are strictly rank-1. However, this is almost impossible for any pre-trained transformer based models. Here, by considering self-attention weight matrices of rank-$\rho$ where $\min(d, s) \geq \rho \geq 1$, we are able to show that singe-head-single-layer self-attention with matrices of any rank is a contextual mapping.

**Remark E.2.** In (Kajitsuka and Sato, 2024), $\gamma$ and $\delta$ are chosen as follows:

$$\Gamma = \gamma_{\max} + \frac{\epsilon}{4}, \quad \Delta = \frac{2(\ln L)^2 \epsilon^2 \gamma_{\min}}{\gamma_{\max}^2 (|\mathcal{V}|+1)^4 (2\ln L + 3)\pi d} \exp\left(-(|\mathcal{V}|+1)^4 \frac{(2\ln L + 3)\pi d \gamma_{\max}^2}{4\epsilon \gamma_{\min}}\right).$$

Since the exponential term dominates the polynomial terms, in Lemma 2.2, we simplify $\Delta$ to $\exp\left(-\Theta(\epsilon^{-1}|\mathcal{V}|^4 d\kappa \gamma_{\max} \ln L)\right)$.

*Proof Sketch.* We generalize the results of (Kajitsuka and Sato, 2024, Theorem 2) where all weight matrices have to be rank-1. We eliminate the rank-1 requirement, and extend the lemma for weights of any rank $\rho$. This is achieved by constructing the weight matrices as a outer product sum $\sum_i^\rho u_i v_i^\top$, where $u_i \in \mathbb{R}^s, v_i \in \mathbb{R}^d$. Specifically, we divide the proof into two parts:

- We first construct a softmax-based self-attention that maps different input tokens to unique contextual embeddings, by configuring weight matrices according to Lemma E.2.

- Secondly, for the identical tokens within a different context, we utilize the tokenwise separateness guaranteed by Lemma E.2 and Lemma D.9 which shows Boltz preserves some separateness.

As a result, we prove that the self-attention function distinguishes input embeddings $Z_{:,k}^{(i)} = Z_{:,l}^{(j)}$ such that $\mathcal{V}^{(i)} \neq \mathcal{V}^{(j)}$. $\square$

*Proof of Lemma 2.2.* We build our proof upon (Kajitsuka and Sato, 2024). We construct a self-attention layer that is a contextual mapping. There are mainly two things to prove. We first show that the attention later we constructed maps different tokens to unique ids. Secondly, we prove that the self-attention function distinguishes duplicate input tokens within different context. For the first part, we show that our self-attention layer satisfies:

$$\|\Psi\| = \left\| W_O \left( W_V Z^{(i)} \right) \mathrm{Softmax} \left[ \left( W_K Z^{(i)} \right)^\top \left( W_Q Z_{:,k}^{(i)} \right) \right] \right\| < \frac{\epsilon}{4}, \tag{E.3}$$

for $i \in [N]$ and $k \in [n]$. Since with (E.3), it is easy to show that

$$\left\| \mathcal{F}_S^{(SA)} \left( Z^{(i)} \right)_{:,k} - \mathcal{F}_S^{(SA)} \left( Z^{(j)} \right)_{:,l} \right\| = \left\| Z_{:,k}^{(i)} - Z_{:,l}^{(j)} + \left( \Psi^{(i)} - \Psi^{(j)} \right) \right\| \tag{E.4}$$

$$\geq \left\| Z_{:,k}^{(i)} - Z_{:,l}^{(j)} \right\| - \left\| \Psi^{(i)} - \Psi^{(j)} \right\|$$

$$\geq \left\| Z_{:,k}^{(i)} - Z_{:,l}^{(j)} \right\| - \left\| \Psi^{(i)} \right\| - \left\| \Psi^{(j)} \right\|$$

$$> \epsilon - \frac{\epsilon}{4} - \frac{\epsilon}{4} = \frac{\epsilon}{2}, \qquad \text{(By $\epsilon$-separatedness of $Z$ and E.3)}$$

for any $i, j \in [N]$ and $k, l \in [n]$ such that $Z_{:,k}^{(i)} \neq Z_{:,l}^{(j)}$. Now, we prove (E.3) by utilizing Lemma E.2. We define the weight matrices as

$$W_K = \sum_{i=1}^\rho p_i q_i^\top \in \mathbb{R}^{s \times d},$$

$$W_Q = \sum_{j=1}^\rho p_j' q_j'^\top \in \mathbb{R}^{s \times d},$$

where $p_i, p_j' \in \mathbb{R}^s$ and $q_i, q_j' \in \mathbb{R}^d$. In addition, let $\delta = 4\ln n$ and $p_1, p_1' \in \mathbb{R}^s$ be an arbitrary vector pair that satisfies

$$\left| p_1^\top p_1' \right| = (|\mathcal{V}|+1)^4 d \frac{\delta}{\epsilon \gamma_{\min}}. \tag{E.5}$$

Then by Lemma E.2, there is some unit vector $q_1, q_1'$ such that we have,

$$\left| (W_K v_a)^\top (W_Q v_c) - (W_K v_b)^\top (W_Q v_c) \right| > \delta, \tag{E.6}$$

for any $v_a, v_b, v_c \in \mathcal{V}$ with $v_a \neq v_b$. In addition, for the other two weight matrices $W_O \in \mathbb{R}^{d \times s}$ and $W_V \in \mathbb{R}^{s \times d}$, we set

$$W_V = \sum_{i=1}^{\rho} p_i'' q_i''^\top \in \mathbb{R}^{s \times d}, \tag{E.7}$$

where $q'' \in \mathbb{R}^d$, $q_1'' = q_1$ and $p_i'' \in \mathbb{R}^s$ is some nonzero vector that satisfies

$$\|W_O p_i''\| = \frac{\epsilon}{4\rho\gamma_{\max}}, \tag{E.8}$$

for any $i \in [\rho]$. As a result, we now bound $\Psi$ as:

$$
\begin{aligned}
\|\Psi\| &= \left\| W_O \left( W_V Z^{(i)} \right) \mathrm{Softmax} \left[ \left( W_K Z^{(i)} \right)^\top \left( W_Q Z_{:,k}^{(i)} \right) \right] \right\| \\
&= \left\| \sum_{k'=1}^{n} s_{k'}^k W_O \left( W_V Z^{(i)} \right)_{:,k'} \right\| \qquad \left( \text{ Denote } s_{k'}^k = \mathrm{Softmax} \left[ \left( W_K Z^{(i)} \right)^\top \left( W_Q Z_{:,k}^{(i)} \right) \right]_{k'} \right) \\
&= \sum_{k'=1}^{n} s_{k'}^k \left\| W_O \left( W_V Z^{(i)} \right)_{:,k'} \right\| \\
&\leq \max_{k' \in [n]} \left\| W_O \left( W_V Z^{(i)} \right)_{:,k'} \right\| \qquad \left( \sum_{k'=1}^{n} s_{k'}^k = 1 \right) \\
&= \max_{k' \in [n]} \left\| W_O \left( \sum_{i=1}^{\rho} p_i'' q_i''^\top \right) Z_{:,k'}^{(i)} \right\| \qquad \text{(By Lemma E.2)} \\
&= \sum_{i=1}^{\rho} \|W_O p_i''\| \cdot \max_{k' \in [n]} \left| q_i''^\top Z_{:,k'}^{(i)} \right| \qquad \text{(By (E.8))} \\
&= \frac{\epsilon}{4\gamma_{\max}} \cdot \max_{k' \in [n]} \left\| Z_{:,k'}^{(i)} \right\| \qquad \text{(By (E.8) and } \|q_i''\| = 1) \\
&< \frac{\epsilon}{4}.
\end{aligned}
$$

Next, for the second part, we prove that with the weight matrices $W_O, W_V, W_K, W_Q$ configured above, the attention layer distinguishes duplicate input tokens with different context, $Z_{:,k}^{(i)} = Z_{:,l}^{(j)}$ with $\mathcal{V}^{(i)} \neq \mathcal{V}^{(j)}$. We choose any $i, j \in [N]$ and $k, l \in [n]$ such that $Z_{:,k}^{(i)} = Z_{:,l}^{(j)}$ and $\mathcal{V}^{(i)} \neq \mathcal{V}^{(j)}$. In addition, we define $a^{(i)}, a^{(j)}$ as

$$
\begin{aligned}
a^{(i)} &= \left( W_K Z^{(i)} \right)^\top \left( W_Q Z_{:,k}^{(i)} \right) \in \mathbb{R}^n, \\
a^{(j)} &= \left( W_K Z^{(j)} \right)^\top \left( W_Q Z_{:,l}^{(j)} \right) \in \mathbb{R}^n.
\end{aligned}
$$

From (E.6) we have that $a^{(i)}$ and $a^{(j)}$ are tokenwise $(\gamma, \delta)$-separated where $\gamma$ is computed by

$$\left| a_{k'}^{(i)} \right| = \left| \left( W_K Z_{:,k'}^{(i)} \right)^\top \left( W_Q Z_{:,k}^{(i)} \right) \right|$$

$$
\begin{aligned}
&= \left| \left( \sum_{i=1}^{\rho} p_i q_i^\top Z_{:,k'}^{(i)} \right)^\top \left( \sum_{j=1}^{\rho} p_j' q_j'^\top Z_{:,k}^{(i)} \right) \right| \\
&= \left| \left( \sum_{i=1}^{\rho} Z_{:,k'}^{(i)\top} q_i p_i^\top \right) \left( \sum_{j=1}^{\rho} p_j' q_j'^\top Z_{:,k}^{(i)} \right) \right| \\
&= \left| \sum_{i=1}^{\rho} \sum_{j=1}^{\rho} Z_{:,k'}^{(i)\top} q_i p_i^\top p_j' q_j'^\top Z_{:,k}^{(i)} \right| \\
&= \sum_{i=1}^{\rho} \sum_{j=1}^{\rho} \left| Z_{:,k'}^{(i)\top} q_i \right| \left| p_i^\top p_j' \right| \left| q_j'^\top Z_{:,k}^{(i)} \right| \\
&\le (|\mathcal{V}| + 1)^4 d \frac{\delta}{\epsilon \gamma_{\min}} \gamma_{\max}^2. \qquad \text{(By (E.5) and } \|q_i\| = \|q_j'\| = 1)
\end{aligned}
$$

Therefore,

$$
\gamma = (|\mathcal{V}| + 1)^4 d \frac{\delta \gamma_{\max}^2}{\epsilon \gamma_{\min}}.
$$

Now, since $\mathcal{V}^{(i)} \neq \mathcal{V}^{(j)}$ and there is no duplicate token in $Z^{(i)}$ and $Z^{(j)}$ respectively, we use Lemma D.9 and obtain that

$$
\begin{aligned}
\left| \text{Boltz}\left( a^{(i)} \right) - \text{Boltz}\left( a^{(j)} \right) \right| &= \left| \left( a^{(i)} \right)^\top \text{Softmax}\left[ a^{(i)} \right] - \left( a^{(j)} \right)^\top \text{Softmax}\left[ a^{(j)} \right] \right| \quad \text{(E.9)} \\
&> \delta' \\
&= (\ln n)^2 e^{-2\gamma}.
\end{aligned}
$$

As we assumed $Z_{:,k}^{(i)} = Z_{:,l}^{(j)}$, we have

$$
\begin{aligned}
&\left| \left( a^{(i)} \right)^\top \text{Softmax}\left[ a^{(i)} \right] - \left( a^{(j)} \right)^\top \text{Softmax}\left[ a^{(j)} \right] \right| \quad\quad\quad\quad\quad\quad\quad \text{(E.10)} \\
&= \left| \left( Z_{:,k}^{(i)} \right)^\top (W_Q)^\top W_K \left( Z^{(i)} \text{Softmax}\left[ a^{(i)} \right] - Z^{(j)} \text{Softmax}\left[ a^{(j)} \right] \right) \right| \\
&= \left| \left( Z_{:,k}^{(i)} \right)^\top \left( \sum_{j=1}^{\rho} q_j' p_j'^\top \right) \left( \sum_{i=1}^{\rho} p_i q_i^\top \right) \left( Z^{(i)} \text{Softmax}\left[ a^{(i)} \right] - Z^{(j)} \text{Softmax}\left[ a^{(j)} \right] \right) \right| \\
&\quad\quad\quad\quad\quad\quad\quad\quad\quad\quad\quad\quad\quad\quad\quad\quad\quad\quad\quad\quad\quad\quad\quad\quad\quad\quad \text{(By Lemma E.2)} \\
&= \sum_{i=1}^{\rho} \sum_{j=1}^{\rho} \left| q_j'^\top Z_{:,k}^{(i)} \right| \cdot \left| p_j'^\top p_i \right| \cdot \left| \left( q_i^\top Z^{(i)} \right) \text{Softmax}\left[ a^{(i)} \right] - \left( q_i^\top Z^{(j)} \right) \text{Softmax}\left[ a^{(j)} \right] \right| \\
&\le \sum_{i=1}^{\rho} \gamma_{\max} \cdot (|\mathcal{V}| + 1)^4 \frac{\pi d}{8} \frac{\delta}{\epsilon \gamma_{\min}} \cdot \left| \left( q_i^\top Z^{(i)} \right) \text{Softmax}\left[ a^{(i)} \right] - \left( q_i^\top Z^{(j)} \right) \text{Softmax}\left[ a^{(j)} \right] \right|. \\
&\quad\quad\quad\quad\quad\quad\quad\quad\quad\quad\quad\quad\quad\quad\quad\quad\quad\quad\quad\quad\quad\quad\quad\quad\quad\quad\quad\quad \text{(By (E.5))}
\end{aligned}
$$

By combining (E.9) and (E.10), we have

$$
\sum_{i=1}^{\rho} \left| \left( q_i^\top Z^{(i)} \right) \text{Softmax}\left[ a^{(i)} \right] - \left( q_i^\top Z^{(j)} \right) \text{Softmax}\left[ a^{(j)} \right] \right| > \frac{\delta'}{(|\mathcal{V}| + 1)^4} \frac{\epsilon \gamma_{\min}}{d \delta \gamma_{\max}}. \quad \text{(E.11)}
$$

Now we arrive at the lower bound of the difference between the self-attention outputs of $Z^{(i)}, Z^{(j)}$ as:

$$
\left\| \mathcal{F}_S^{(\mathrm{SA})} \left( Z^{(i)} \right)_{:,k} - \mathcal{F}_S^{(\mathrm{SA})} \left( Z^{(j)} \right)_{:,l} \right\| \tag{E.12}
$$

$$
= \left\| W_O \left( W_V Z^{(i)} \right) \mathrm{Softmax} \left[ a^{(i)} \right] - W_O \left( W_V Z^{(j)} \right) \mathrm{Softmax} \left[ a^{(j)} \right] \right\|
$$

$$
= \sum_{i=1}^{\rho} \| W_O p_i'' \| \cdot \left| \left( q_i''^{\top} Z^{(i)} \right) \mathrm{Softmax} \left[ a^{(i)} \right] - \left( q_i''^{\top} Z^{(j)} \right) \mathrm{Softmax} \left[ a^{(j)} \right] \right|
$$

$$
\qquad\qquad\qquad\qquad\qquad\qquad\qquad\qquad\qquad\qquad \left( W_V = \textstyle\sum_{i=1}^{\rho} p_i'' q_i''^{\top} \right)
$$

$$
> \frac{\epsilon}{4\gamma_{\max}} \frac{\delta'}{(|\mathcal{V}| + 1)^4} \frac{\epsilon \gamma_{\min}}{d \delta \gamma_{\max}}. \qquad\qquad\qquad\qquad \left( \text{By (E.8) and (E.11)} \right)
$$

where $\delta = 4 \ln n$ and $\delta' = \ln^2(n) e^{-2\gamma}$ with $\gamma = (|\mathcal{V}| + 1)^4 d \delta \gamma_{\max}^2 / (\epsilon \gamma_{\min})$. Note that we are able to use (E.11) in the last inequality of (E.12) because (E.11) is guaranteed by $q_1$, and we set $q_1'' = q_1$ when constructing $W_V$ in (E.7). $\qquad\square$

# F    PROOFS OF SECTION 2.3

We consider the continuous sequence-to-sequence functions on a compact set of sequence as $f_{\text{seq2seq}} : [0,1]^{d \times L} \mapsto [0,1]^{d \times L}$. Furthermore, consider the function class of continuous sequence-to-sequence $\mathcal{F}_C$ which is $C$-Lipschitz in $\ell_\alpha$ norm. Explicitly, for any $f_{\text{seq2seq}} \in \mathcal{F}_C$ and two input embeddings $Z, Z'$, we have

$$\|f_{\text{seq2seq}}(Z) - f_{\text{seq2seq}}(Z')\|_\alpha \leq C \|Z - Z'\|_\alpha.$$

In addition, we consider simple transformers $\tau \in \mathcal{T}_A^{1,1,4}$ which consist of single-head single-layer size-one self-attention $f^{(\text{SA})} \in \mathcal{F}^{(\text{SA})}$ and $\ell_1 + \ell_2$ feed-forward layers $f^{(\text{FF})} \in \mathcal{F}^{(\text{FF})}$ each with 4 MLP hidden neurons:

$$\mathcal{T}_A^{1,1,4} := \{\tau : \mathbb{R}^{d \times L} \mapsto \mathbb{R}^{d \times L} | \tau = f_{\ell_1}^{(FF)} \circ \ldots \circ f_1^{(FF)} \circ f^{(\text{SA})} \circ f_{\ell_2}^{(FF)} \circ \ldots \circ f_1^{(FF)}\}.$$

Finally, define the approximation error for some given functions $f_1, f_2$ as:

$$d_\alpha(f_1, f_2) = \left( \int \|f_1(Z) - f_2(Z)\|_\alpha^\alpha dZ \right)^{\frac{1}{\alpha}}. \tag{F.1}$$

In this section, we prove the universality of prompt tuning by showing that there exists a simple transformer of single-layer self-attention $\tau \in \mathcal{T}_A^{1,1,4}$ such that for any $f_{\text{seq2seq}} \in \mathcal{F}_C$, prompt tuning on $g$ approximates this function up to some error $\epsilon > 0$.

The proof follows the construction base recipe of (Yun et al., 2020) and (Wang et al., 2023a). We start by quantizing the input and output domain of $\mathcal{F}_C$ such that — for each $f_{\text{seq2seq}} \in \mathcal{F}_C$, we obtain a quantized function $\bar{f}_{\text{seq2seq}} : \mathcal{G}_{\delta,L} \mapsto \mathcal{G}_{\delta,L}$ where $\mathcal{G}_{\delta,L} = \{0, \delta, 2\delta, \ldots, 1 - \delta\}^{d \times L}$. Here, $\bar{f}_{\text{seq2seq}}, \overline{\mathcal{F}}_C$ denote the seq2seq function and quantized function class, respectively. This is basically performing a piece-wise constant approximation, i.e., the values inside a quantized grid assume the same value. Next, we build a surrogate quantized sequence-to-sequence function $h_{\text{seq2seq}} : \mathcal{G}_{\delta,(L_p+L)} \to \mathcal{G}_{\delta,(L_p+L)}$ with $\mathcal{G}_{\delta,(L_p+L)} = \{0, \delta, 2\delta, \ldots, 1 - \delta\}^{d \times (L_p+L)}$ that takes the concatenation of prompts $P$ and embeddings $Z$ as inputs. Importantly, we let "the last $L$ tokes" of this quantized function $h_{\text{seq2seq}}$ approximates any $\bar{f}_{\text{seq2seq}} \in \overline{\mathcal{F}}_C$ by taking different prompts $P$. Finally, we construct some transformer $\tau \in \mathcal{T}_A^{1,1,4}$ to approximate $h_{\text{seq2seq}}$. This leads to a chaining reduction of approximations, which implies $\tau \in \mathcal{T}_A^{1,1,4}$ approximates $f_{\text{seq2seq}}$ up to any accuracy $\epsilon$.

## F.1    PROOF OF LEMMA 2.3

We start by building quantized sequence-to-sequence functions $h_{\text{seq2seq}} : \mathcal{G}_{\delta,(L_p+L)} \to \mathcal{G}_{\delta,(L_p+L)}$ with quantized prompts to approximate $\bar{f}_{\text{seq2seq}}$. Next, we approximate $h_{\text{seq2seq}}$ with transformer functions $\tau \in \mathcal{T}_A^{1,1,4}$. To achieve this, we use the feed-forward layer for quantizing the input and output domain of transformers. Also, we utilize self-attention layer as contextual mapping. As a result, we construct a transformer for prompt tuning to approximate any continuous sequence-to-sequence function.

First, we introduce the lemma below which shows that, the quantized sequence-to-sequence function $\bar{f}_{\text{seq2seq}}$ is approximated by some sequence-to-sequence function $h_{\text{seq2seq}} : \mathcal{G}_{\delta,(L_p+L)} \to \mathcal{G}_{\delta,(L_p+L)}$ where $\mathcal{G}_{\delta,(L_p+L)} = \{0, \delta, 2\delta, \ldots, 1 - \delta\}^{d \times (L_p+L)}$.

**Lemma F.1** (Lemma 2.3 Restated). *Let $\mathcal{F}_C$ be a class of $C$-Lipschitz sequence-to-sequence functions, where each function $f_{\text{seq2seq}} : [0,1]^{d \times L} \to [0,1]^{d \times L}$. Define the discrete function space $\mathcal{G}_{\delta,(L_p+L)} = \{0, \delta, 2\delta, \ldots, 1 - \delta\}^{d \times (L_p+L)}$. Then, there exists a function $h_{\text{seq2seq}} : \mathcal{G}_{\delta,(L_p+L)} \to \mathcal{G}_{\delta,(L_p+L)}$ such that for any $f_{\text{seq2seq}} \in \mathcal{F}_C$, there exists a prompt $P \in \mathbb{R}^{d \times L_p}$ satisfying*

$$d_p\left(h([P, \cdot])_{:,L_p:}, f_{\text{seq2seq}}\right) \leq \epsilon/2,$$

where the prompt sequence length $L_p$ satisfies $L_p \geq L\lambda$, with $\lambda = (2\epsilon^{-1}C(dL)^{1/\alpha})^{dL}$.

*Proof of Lemma F.1.* We first quantize the input and output sequence domain of $\mathcal{F}_C$ by quantizing $[0,1]^{d\times L}$ into a grid space $\mathcal{G}_{\delta,L} = \{0, \delta, 2\delta, \ldots, 1-\delta\}^{d\times L}$. Observe that there are $n = \left(\frac{1}{\delta}\right)^{dL}$ different matrices in the grid space $\mathcal{G}_{\delta,L}$. Now, consider all the possible input to output mappings, we have $m = n^n$ piece-wise constant functions $\bar{f}_{\text{seq2seq}} \in \overline{\mathcal{F}}_C$. We define $\bar{f}_{\text{seq2seq}} : \mathcal{G}_{\delta,L} \mapsto \mathcal{G}_{\delta,L}$ as

$$\bar{f}_{\text{seq2seq}}(Z) = \begin{cases} \bar{f}_{\text{seq2seq}}(Z) & Z \in \mathcal{G}_{\delta,L} \\ \bar{f}_{\text{seq2seq}}(Z^\star) & \text{otherwise} \end{cases},$$

where $k_{i,j}\delta < Z_{i,j}, Z^\star_{i,j} \leq (k_{i,j}+1)\delta$, while $Z^\star \in \mathcal{G}_{\delta,L}$ and $k_{i,j} \in \{0, 1, ..., 1/\delta - 1\}$. We set the function class for the quantized space as $\overline{\mathcal{F}}_C = \left\{ \bar{f}^{(1)}_{\text{seq2seq}}, \bar{f}^{(2)}_{\text{seq2seq}}, \ldots, \bar{f}^{(m)}_{\text{seq2seq}} \right\}$. Then, by utilizing the $C$-Lipschitzness, we have that for any $f_{\text{seq2seq}} \in \mathcal{F}_C$, there is a piece-wise constant approximation function $\bar{f}_{\text{seq2seq}} \in \overline{\mathcal{F}}_C$ that satisfies

$$d_\alpha(\bar{f}_{\text{seq2seq}}, f_{\text{seq2seq}}) = \left( \int \left\| \bar{f}_{\text{seq2seq}}(Z) - f_{\text{seq2seq}}(Z) \right\|_\alpha^\alpha dZ \right)^{1/\alpha} \qquad \text{(By (F.1))}$$

$$\leq \left( \int (C\delta)^\alpha \, dL \cdot dZ \right)^{1/\alpha} \qquad \text{(By } C\text{-Lipschitzness)}$$

$$= C\delta(dL)^{\frac{1}{\alpha}}.$$

By choosing $\delta = \delta^\star$ such that $C\delta(dL)^{\frac{1}{\alpha}} \leq \epsilon/2$, we have

$$d_\alpha(\bar{f}_{\text{seq2seq}}, f_{\text{seq2seq}}) \leq \frac{\epsilon}{2}. \qquad (\text{F.2})$$

Next, we quantize the prompts $P \in \mathbb{R}^{d\times L_p}$. We consider a set of quantized prompts in grid space $\mathcal{G}_{\delta,L_p} = \{0, \delta, 2\delta, \ldots, 1-\delta\}^{d\times L_p}$. This gives us $m_p = \left(\frac{1}{\delta}\right)^{dL_p}$ different quantized prompts. We denote this set of prompts as $\mathcal{P} = \left\{ P^{(1)}, P^{(2)}, \ldots, P^{(m_p)} \right\}$.

Since there are $m = n^n = \left(\frac{1}{\delta^{dL}}\right)^{\frac{1}{\delta^{dL}}}$ functions in $\overline{\mathcal{F}}_C$, the required prompt length $L_p$ to index all $m$ functions in $\overline{\mathcal{F}}_C$ is This gives

$$L_p \geq L \left(\frac{1}{\delta}\right)^{dL}$$

$$\geq L \left(\frac{1}{\epsilon} 2C(dL)^{\frac{1}{\alpha}}\right)^{dL}. \qquad (\text{Since we choose } \delta \text{ such that } C\delta(dL)^{\frac{1}{\alpha}} \leq \epsilon/2)$$

Finally, we define some quantized function $h_{\text{seq2seq}} : \mathcal{G}_{\delta,(L_p+L)} \to \mathcal{G}_{\delta,(L_p+L)}$ where $\mathcal{G}_{\delta,(L_p+L)} = \{0, \delta, 2\delta, \ldots, 1-\delta\}^{d\times(L_p+L)}$, and let

$$h_{\text{seq2seq}}\left(\left[P^{(i)}, Z\right]\right)_{:,L_p:} = \bar{f}^{(i)}_{\text{seq2seq}}(Z). \qquad (\text{F.3})$$

In addition, we set the first $L_p$ columns of $h_{\text{seq2seq}}$ to be zero, which is

$$h_{\text{seq2seq}}\left(\left[P^{(i)}, Z\right]\right)_{:,:L_p} = 0,$$

for all $Z \in [0,1]^{d \times L}, P \in \mathcal{G}_{\delta, L_p}$. Furthermore, let

$$h_{\text{seq2seq}}\left([P,Z]\right)_{:,L_p:} = \begin{cases} h_{\text{seq2seq}}\left([P,Z]\right)_{:,L_p:} & P \in \mathcal{P} \\ h_{\text{seq2seq}}\left([P^\star, Z]\right)_{:,L_p:} & \text{otherwise} \end{cases},$$

where $k_{i,j}\delta < P_{i,j}, P_{i,j}^\star \leq (k_{i,j}+1)\delta$, while $P^\star \in \mathcal{P}$ and $k_{i.j} \in \{0, 1, ..., 1/\delta - 1\}$.

As a result, we show that with a properly chosen grid granularity $\delta = \delta_1$, for any sequence-to-sequence function $f_{\text{seq2seq}} \in \mathcal{F}_C$, we build a quantized function $h$ with prompt $P$ that approximates $f_{\text{seq2seq}}$ with error $\epsilon/2$,

$$d_\alpha\left(h_{\text{seq2seq}}([P, \cdot])_{:,L_p:}, f_{\text{seq2seq}}\right) = d_\alpha\left(\bar{f}_{\text{seq2seq}}, f_{\text{seq2seq}}\right) \leq \epsilon/2.$$

This completes the proof. □

### F.2 PROOFS OF LEMMA 2.4

Here we show $\tau \in \mathcal{T}_A^{1,1,4}$ approximates the surrogate quantized seq2seq function $h_{\text{seq2seq}}$ up to any precision. To do this, we utilize Lemma 2.2 to construct a transformer $\tau \in \mathcal{T}_A^{1,1,4}$. Then we show that this transformer $\tau$ approximates quantized sequence-to-sequence functions $h_{\text{seq2seq}}([P, \cdot])$.

**Lemma F.2** (Lemma 2.4 Restated). *For any given quantized sequence-to-sequence function $h_{\text{seq2seq}}$ : $\mathcal{G}_{\delta,(L_p+L)} \to \mathcal{G}_{\delta,(L_p+L)}$ with $\mathcal{G}_{\delta,(L_p+L)} = \{0, \delta, 2\delta, \ldots, 1-\delta\}^{d \times (L_p+L)}$, there exists a transformer $\tau \in \mathcal{T}_A^{1,1,4}$ with positional encoding $E \in \mathbb{R}^{d \times (L_p+L)}$, such that $\tau = h([P, \cdot])_{:,L_p:}$.*

*Proof Sketch.* This lemma is inspired by (Wang et al., 2023a, Lemma 2). There are mainly three steps:

1. Given an input data with prompt $[P, Z] \in \mathbb{R}^{d \times (L_p+L)}$, we first apply positional encoding $E$, which is given as

$$E = \begin{bmatrix} 0 & 1 & 2 & \ldots & L_p + L - 1 \\ 0 & 1 & 2 & \ldots & L_p + L - 1 \\ \vdots & \vdots & \vdots & \ddots & \vdots \\ 0 & 1 & 2 & \ldots & L_p + L - 1 \end{bmatrix}.$$

Then a series of feed-forward layers in the modified Transformer network quantizes $[P, Z] + E$ to a quantized sequence $M \in \bar{\mathcal{G}}_{\delta,(L_p+L)}$. Here, we define the grid

$$\bar{\mathcal{G}}_{\delta,(L_p+L)} := [0:\delta:1-\delta]^d \times [1:\delta:2-\delta]^d \times \cdots \times [L_p+L-1:\delta:L_p+L-\delta]^d,$$

where $[a:\varepsilon:b] := \{a, a+\varepsilon, a+2\varepsilon, \ldots, b-\varepsilon, b\}$. Note that with the positional encoding, our contextual mapping through self-attention won't be limited to permutation equivalent functions.

2. Next, by utilizing Lemma 2.2, the single self-attention layer in the modified transformer takes the input $M$ and implements a contextual mapping $q : \mathbb{R}^{d \times (L+L_p)} \mapsto \mathbb{R}^{d \times (L+L_p)}$.

3. Finally, a series of feed-forward layers map elements of the contextual embedding $q(M)$ to the desired output value of $h_{\text{seq2seq}}([P, Z])$.

We remark that Step 2 distinguishes us from prior works by utilizing the fact that any-rank attention is a contextual mapping Lemma 2.2. This dramatically improves the result of (Wang et al., 2023a), which requires a depth of $dL/\epsilon$ layers, to just a single layer. □

*Proof of Lemma F.2.* First, we apply the positional encoding $E \in \mathbb{R}^{d \times (L_p+L)}$ on the input sequence with prompt sequence $[P, Z] \in \mathbb{R}^{d \times (L_p+L)}$, so that each token has a different domain. The positional encoding $E$ is given as

$$
E = \begin{bmatrix} 0 & 1 & 2 & \ldots & L_p + L - 1 \\ 0 & 1 & 2 & \ldots & L_p + L - 1 \\ \vdots & \vdots & \vdots & \ddots & \vdots \\ 0 & 1 & 2 & \ldots & L_p + L - 1 \end{bmatrix}.
$$

We next use feed-forward layers $f^{(\text{FF})}$ to implement a quantization map to quantize the input $[P, Z] + E$ in to its discrete version $M \in \bar{\mathcal{G}}_{\delta,(L_p+L)}$. The grid $\bar{\mathcal{G}}_{\delta,(L_p+L)}$ is defined as

$$
\bar{\mathcal{G}}_{\delta,(L_p+L)} := [0 : \delta : 1 - \delta]^d \times [1 : \delta : 2 - \delta]^d \times \cdots \times [L_p + L - 1 : \delta : L_p + L - \delta]^d,
$$

where $[a : \varepsilon : b] := \{a, a + \varepsilon, a + 2\varepsilon, \ldots, b - \varepsilon, b\}$. Note that the first column of $[P, Z] + E$ is in $[0, 1]^d$, the second is in $[1, 2]^d$, and so on. Here, we write the quantization mapping as

$$
[0, 1]^d \times \cdots \times [L_p + L - 1, L_p + L]^d \mapsto [0 : \delta : 1 - \delta]^d \times \cdots \times [L_p + L - 1 : \delta : L_p + L - \delta]^d,
$$

where $[a : \varepsilon : b] := \{a, a + \varepsilon, a + 2\varepsilon, \ldots, b - \varepsilon, b\}$. Inspired by the construction recipe by (Yun et al., 2020), this task is realized by $d(L_p + L)/\delta$ feed-forward layers. We add $d(L_p + L)/\delta$ layers of $f^{(\text{FF})}$ with the following form, for $k = 0, \delta, \ldots, (L_p + L) - \delta$ and $i = 1, \ldots, d$:

$$
Z \mapsto Z + e^{(i)} \phi \left( \left( e^{(i)} \right)^T Z - k\delta \mathbf{1}_n^T \right), \phi(t) = \begin{cases} 0 & t < 0 \text{ or } t \geq \delta \\ -t + 1 & 0 \leq t < \delta \end{cases}, \tag{F.4}
$$

where $e^{(1)} = (1, 0, 0, ..., 0) \in \mathbb{R}^d$ and $\phi(t) \in \Phi$ is an entrywise function, where the set of activation functions $\Phi$ consists of all piece-wise linear functions with at least one piece being constant and at most three pieces. Furthermore, any activation function $\phi \in \Phi$ is realized by 4 MLP neurons. Each layer in the form of (F.4) quantizes $X_{i,:}$ (the $i$-th row) in $[k\delta, k\delta + \delta)$ to $k\delta$. We denote output after the feed-forward layers as $M \in \bar{\mathcal{G}}_{\delta,(L_p+L)}$.

Next, in order to utilize Lemma 2.2, we observe that the quantized output $M$ from the previous step has no duplicate tokens, since each column has a unique domain. Also, we see that $M$ is token-wise $\left( \sqrt{d}, \sqrt{d}(L' - \delta), \sqrt{d}\delta \right)$-separated where $L' = L_p + L$. This is easily observed as we have, for any $k, l \in [L_p + L]$,

$$
\|M_{:,k}\| > \sqrt{d},
$$
$$
\|M_{:,k}\| < \sqrt{d}(L_p + L - \delta),
$$
$$
\|M_{:,k} - M_{:,l}\| > \sqrt{d}\delta.
$$

As a result, with Lemma 2.2, we arrive at a $(\Gamma, \Delta)$-contextual mapping $q : \mathbb{R}^{d \times (L_p+L)} \mapsto \mathbb{R}^{d \times (L_p+L)}$ where

$$
\Gamma = \sqrt{d}(L' - \delta) + \frac{\sqrt{d}\delta}{4} = \sqrt{d}\left(L' - \frac{3\delta}{4}\right),
$$
$$
\Delta = \exp\left(-5|\mathcal{V}|^4 d \ln(n) L'^2/\delta\right).
$$

Now we have successfully mapped each input sequence $[P, Z] + E$ to unique context ID $q(M) \in \mathbb{R}^{d \times (L_p+L)}$. We next associate each unique embeddings to a corresponding expected output of $h([P, \cdot])$.

Finally, we use feed-forward layers to map each token of $q(M)$ to the desired $[0, 1]^d$. As in (Yun et al., 2020, C.3), with a method similar to (F.4), we need one layer for each unique value of $q(M)$

for each $M \in \bar{\mathcal{G}}_{\delta,(L_p+L)}$. There are in total $(1/\delta)^{d(L_p+L)}$ possibilities of $M$ and each corresponds to some output of $h_{\text{seq2seq}}([P,\cdot])$. Since we only focus on the last $L$ tokens of output, we require $\mathcal{O}\left(L(1/\delta)^{d(L_p+L)}\right) = \mathcal{O}\left(\delta^{-d(L_p+L)}\right)$ layers to map these distinct numbers to expected outputs.

This completes the proof. □

### F.3 PROOFS OF THEOREM 2.3

With Lemma F.2, we are able to find a transformer $\tau \in \mathcal{T}_A^{1,1,4}$ such that $\tau([P,Z]) = h([P,Z])$. Finally, we arrive at the theorem that shows that a transformer of one single-head self-attention layer is a universal approximator for sequence-to-sequence functions.

**Theorem F.1** (Theorem 2.3 Restated). Let $1 \le p < \infty$ and $\epsilon > 0$. There exists a transformer $\tau \in \mathcal{T}_A^{1,1,4}$ with a single self-attention layer such that for any $f_{\text{seq2seq}} \in \mathcal{F}_C$, there exists a prompt $P \in \mathbb{R}^{d \times L_p}$ satisfying $d_\alpha\left(\tau([P,\cdot])_{:,L_p}, f_{\text{seq2seq}}\right) \le \epsilon$.

*Proof of Theorem 2.3.* Combining Lemma F.1 and Lemma F.2, we arrive at a transformer $\tau \in \mathcal{T}_A^{1,1,4}$, with prompt $P \in \mathcal{G}_{\delta,L_p}$, such that for any sequence-to-sequence $f_{\text{seq2seq}} \in \mathcal{F}_C$,

$$
\begin{aligned}
&d_\alpha\left(\tau\left([P,\cdot]\right)_{:,L_p:}, f_{\text{seq2seq}}\right) \\
&\le d_\alpha\left(\tau\left([P,\cdot]\right)_{:,L_p:}, h_{\text{seq2seq}}\left([P,\cdot]\right)_{:,L_p:}\right) + d_\alpha\left(h_{\text{seq2seq}}\left([P,\cdot]\right)_{:,L_p:}, f_{\text{seq2seq}}\right) \\
&\le \epsilon.
\end{aligned}
$$

This completes the proof. □

## G    PROOFS OF SECTION 2.4

### G.1    PROOF OF LEMMA 2.5

For the transformer $\tau \in \mathcal{T}_A^{1,1,4}$ in the previous section Appendix F, we compute the required number of FFN layers.

**Lemma G.1** (Lemma 2.5 Restated)**.** A transformer $\tau \in \mathcal{T}_A^{1,1,4}$, as defined in (2.3), requires $\mathcal{O}((1/\epsilon)^{d(L_p+L)})$ FFN layers to be a universal approximator through prompt tuning.

*Proof.* As shown in the final step of the proof for Lemma F.2, we require $\mathcal{O}\left(\delta^{-d(L_p+L)}\right)$ layers to map these distinct numbers to expected outputs. Recall that in (F.2), we have the relation of quantization granularity $\delta$ and function approximation error $\epsilon$ as $C\delta(dL)^{\frac{1}{\alpha}} \leq \epsilon/2$. We write the number of feed-forward layers as $\mathcal{O}\left(2L(C(dL)^{\frac{1}{\alpha}}/\epsilon)^{d(L_p+L)}\right) = \mathcal{O}\left(\epsilon^{-d(L_p+L)}\right)$, where $C$ is the Lipschitz constant and $\alpha$ is from the $\ell_\alpha$-norm we use for measuring the approximation error.    □

### G.2    PROOF OF THEOREM 2.4

In this section, we prove the universality of prompt tuning on another simple transformer architecture with a smaller depth than $\mathcal{T}_A^{1,1,4}$ from Section 2.3. This provides us a case for trade off between the depth and width of the transformer.

Consider transformers $\tau \in \mathcal{T}_B^{1,1,r}$ which consist of single-head single-layer size-one self-attention $f^{(\text{SA})}$ and two feed-forward layers $f_1^{(\text{FF})}, f_2^{(\text{FF})}$ each with $r$ MLP hidden neurons:

$$\mathcal{T}_B^{1,1,r} := \{g : \mathbb{R}^{d\times L} \mapsto \mathbb{R}^{d\times L} | \tau = f_2^{(\text{FF})} \circ f^{(\text{SA})} \circ f_1^{(\text{FF})}\}.$$

We prove the universality of prompt tuning by showing that there exists a transformer network $\tau \in \mathcal{T}_B^{1,1,r}$ such that for any $f_{\text{seq2seq}} \in \mathcal{F}_C$, prompt tuning on $\tau$ approximates this function up to some error $\epsilon > 0$.

Similar to the proof of Theorem F.1, we start by quantizing the input and output domain of $\mathcal{F}_C$ to obtain quantized functions

$$\bar{f}_{\text{seq2seq}} : \mathcal{G}_{\delta,L} \mapsto \mathcal{G}_{\delta,L},$$

where

$$\mathcal{G}_{\delta,L} = \{0, \delta, 2\delta, \ldots, 1-\delta\}^{d\times L}.$$

This is basically performing a piece-wise constant approximation. Next, we build a quantized sequence-to-sequence function

$$h_{\text{seq2seq}} : \mathcal{G}_{\delta,(L_p+L)} \to \mathcal{G}_{\delta,(L_p+L)} \quad \text{with} \quad \mathcal{G}_{\delta,(L_p+L)} = \{0, \delta, 2\delta, \ldots, 1-\delta\}^{d\times(L_p+L)},$$

that takes the concatenation of prompts $P$ and embeddings $Z$ as inputs. This quantized function $h_{\text{seq2seq}}$ approximates any $\bar{f}_{\text{seq2seq}} \in \bar{\mathcal{F}}_C$ by taking different prompts $P$. Finally, we construct some transformer $\tau \in \mathcal{T}_B^{1,1,r}$ to approximate $h_{\text{seq2seq}}$.

First, we utilize the results from Lemma F.1, which shows that the quantized sequence-to-sequence function $\bar{f}_{\text{seq2seq}}$ is approximated by some sequence-to-sequence function

$$h_{\text{seq2seq}} : \mathcal{G}_{\delta,(L_p+L)} \to \mathcal{G}_{\delta,(L_p+L)} \quad \text{with} \quad \mathcal{G}_{\delta,(L_p+L)} = \{0, \delta, 2\delta, \ldots, 1-\delta\}^{d\times(L_p+L)}.$$

Next, in Lemma G.2, we utilize Lemma 2.2 to construct a transformer $\tau \in \mathcal{T}_B^{1,1,r}$. Then, we use the transformer to approximate quantized sequence-to-sequence functions $h_{\text{seq2seq}}([P, \cdot])$.

---

**Lemma G.2** (Transformer Construction). For any given quantized sequence-to-sequence function

$$h_{\text{seq2seq}} : \mathcal{G}_{\delta,(L_p+L)} \to \mathcal{G}_{\delta,(L_p+L)} \quad \text{with} \quad \mathcal{G}_{\delta,(L_p+L)} = \{0, \delta, 2\delta, \ldots, 1-\delta\}^{d \times (L_p+L)},$$

there exists a transformer $\tau \in \mathcal{T}_B^{1,1,r}$ with positional embedding $E \in \mathbb{R}^{d \times (L_p+L)}$, such that

$$d_\alpha \left( \tau, h([P, \cdot])_{:,L_p:} \right) \le \epsilon/2.$$

---

*Proof Sketch.* The proof of this lemma follows a similar idea as Lemma F.2. Nonetheless, by applying the construction technique from (Kajitsuka and Sato, 2024), we employ a transformer configuration that utilizes just two feed-forward layers.

The proof consists of three steps:

1. Given an input data with prompt $[P, Z] \in \mathbb{R}^{d \times (L_p+L)}$, we first apply positional encoding $E$, which is given as

$$E = \begin{bmatrix} 0 & 1 & 2 & \ldots & L_p + L - 1 \\ 0 & 1 & 2 & \ldots & L_p + L - 1 \\ \vdots & \vdots & \vdots & \ddots & \vdots \\ 0 & 1 & 2 & \ldots & L_p + L - 1 \end{bmatrix}.$$

   Then a series of feed-forward layers in the modified Transformer network quantizes $[P, Z] + E$ to a quantized sequence $M \in \bar{\mathcal{G}}_\delta$. Here, we define the grid

$$\bar{\mathcal{G}}_\delta = [\delta : \delta : 1]^d \times [1 + \delta : \delta : 2]^d \times \cdots \times [L_p + L - 1 + \delta : \delta : L_p + L]^d,$$

   where $[a : \varepsilon : b] := \{a, a + \varepsilon, a + 2\varepsilon, \ldots, b - \varepsilon, b\}$. Note that with the positional encoding, our contextual mapping through self-attention won't be limited to permutation equivalent functions.

2. Next, by utilizing Lemma 2.2, the single self-attention layer in the modified transformer takes the input $M$ and implements a contextual mapping $q : \mathbb{R}^{d \times (L+L_p)} \mapsto \mathbb{R}^{d \times (L+L_p)}$.

3. Finally, a series of feed-forward layers map elements of the contextual embedding $q(M)$ to the desired output value of $h_{\text{seq2seq}}([P, Z])$.

$\square$

---

*Proof of Lemma G.2.* First, we apply the positional encoding $E \in \mathbb{R}^{d \times (L_p+L)}$ on the input sequence with prompt sequence $[P, Z] \in \mathbb{R}^{d \times (L_p+L)}$, so that each token of has a different domain. The positional encoding $E$ is given as

$$E = \begin{bmatrix} 0 & 1 & 2 & \ldots & L_p + L - 1 \\ 0 & 1 & 2 & \ldots & L_p + L - 1 \\ \vdots & \vdots & \vdots & \ddots & \vdots \\ 0 & 1 & 2 & \ldots & L_p + L - 1 \end{bmatrix}.$$

We next use the first feed-forward layer $f_1^{(\text{FF})}$ to implement a quantization map to quantize the input $[P, Z] + E$ into its discrete version $M \in \bar{\mathcal{G}}_\delta$ . Here, we define the grid

$$\bar{\mathcal{G}}_\delta = [\delta : \delta : 1]^d \times [1 + \delta : \delta : 2]^d \times \cdots \times [L_p + L - 1 + \delta : \delta : L_p + L]^d,$$

where $[a : \varepsilon : b] := \{a, a + \varepsilon, a + 2\varepsilon, \ldots, b - \varepsilon, b\}$. Note that the first column of $[P, Z] + E$ is in $[0, 1]^d$, the second is in $[1, 2]^d$, and so on. Here, we write the quantization mapping as

$$[0, 1]^d \times \cdots \times [L_p + L - 1, L_p + L]^d \mapsto [\delta : \delta : 1 - \delta]^d \times \cdots \times [L_p + L - 1 : \delta : L_p + L]^d,$$

where $[a : \varepsilon : b] := \{a, a + \varepsilon, a + 2\varepsilon, \ldots, b - \varepsilon, b\}$. Following (Kajitsuka and Sato, 2024), this quantization task is done by constructing the feed-forward layer as a $\theta$-approximated step function. Consider a real value piece-wise constant function $f^{(\text{Step})} : \mathbb{R} \mapsto \mathbb{R}$, for any small $\theta > 0$, $z \in \mathbb{R}$, we have the $\theta$-approximation as

$$f^{(\text{Step})}(z) \approx \sum_{t=0}^{(L_p+L)(1/\delta-1)} \left( \text{ReLU}\left(z/\theta - t\delta/\theta\right) - \text{ReLU}\left(z/\theta - 1 - t\delta/\theta\right) \right) \delta \quad \text{(G.1)}$$

$$= \begin{cases} 0 & z < 0 \\ \delta & 0 \le z < \delta \\ \vdots & \vdots \\ L + L_p & L + L_p - \delta \le z \end{cases},$$

which is a series of small step functions, each beginning their rise at $t\delta$ and ending at $\theta + t\delta$. Here, we show the first two terms $t = 0, 1$ for clarity:

$$t = 0 : \left( \text{ReLU}\left(z/\theta\right) - \text{ReLU}\left(z/\theta - 1\right) \right) \delta = \begin{cases} 0 & z < 0 \\ z\delta/\theta & 0 \le z < \theta \\ \delta & \theta \le z \end{cases},$$

$$t = 1 : \left( \text{ReLU}\left(z/\theta - \delta/\theta\right) - \text{ReLU}\left(z/\theta - 1 - \delta/\theta\right) \right) \delta = \begin{cases} 0 & z < \delta \\ z\delta/\theta & \delta \le z < \theta + \delta \\ \delta & \theta + \delta \le z \end{cases}.$$

With (G.1), it is straightforward that we extend it to $\mathbb{R}^{d \times L}$. As a result, we have the first feed-forward layer $f_1^{(\text{FF})}$ as

$$f_1^{(\text{FF})}(Z)_{i,j} = \sum_{t=0}^{(L_p+L)(1/\delta-1)} \left( \text{ReLU}\left(Z_{i,j}/\theta - t\delta/\theta\right) - \text{ReLU}\left(Z_{i,j}/\theta - 1 - t\delta/\theta\right) \right) \delta \quad \text{(G.2)}$$

$$\approx f^{(Step)}(Z_{i,j}),$$

where $i \in [d], j \in [L_p + L], 0 < \delta < 1$ and $\theta > 0$. With (G.2), we are able to quantize each sequence $[P, Z] + E$ to a quantized version $M \in \bar{\mathcal{G}}_\delta$.

Next, in order to utilize Lemma 2.2, we observe that the quantized input $M$ from the previous step has no duplicate tokens, since each column has a unique domain. Also, we see that $M$ is token-wise $\left(\sqrt{d}, \sqrt{d}(L' - \delta), \sqrt{d}\delta\right)$-separated where $L' = L_p + L$. This is easily observed as we have, for any $k, l \in [L_p + L]$,

$$\|M_{:,k}\| > \sqrt{d},$$
$$\|M_{:,k}\| < \sqrt{d}(L_p + L - \delta),$$
$$\|M_{:,k} - L_{:,l}\| > \sqrt{d}\delta.$$

As a result, with Lemma 2.2, the single self-attention layer implements a contextual mapping $q : \mathbb{R}^{d \times (L+L_p)} \mapsto \mathbb{R}^{d \times (L+L_p)}$, we arrive at a $(\Gamma, \Delta)$-contextual mapping where

$$\Gamma = \sqrt{d}(L' - \delta) + \frac{\sqrt{d}\delta}{4} = \sqrt{d}(L' - \frac{3\delta}{4}),$$

$$\Delta = \exp\left(-5|\mathcal{V}|^4 d \ln(n) L'^2/\delta\right).$$

Now we have successfully mapped each input sequence $[P, Z] + E$ to a unique context ID $q(M) \in \mathbb{R}^{d \times (L_p + L)}$. We next associate each unique embeddings to a corresponding expected output of $h_{\text{seq2seq}}([P, \cdot])$.

We associate each unique contextual embeddings to the corresponding output of $h([P, \cdot])$ using the second feed-forward layer $f_2^{(\text{FF})}$. As in (Kajitsuka and Sato, 2024, A.5), this is achieved by constructing a bump function $f_{\text{bump}} : \mathbb{R}^{d \times (L_p + L)} \mapsto \mathbb{R}^{d \times (L_p + L)}$ for each possible output from the last step $q(M^{(i)}), i \in [(1/\delta)^{d(L_p + L)}]$. Each bump function $f_{\text{bump}}$ is realized by $3d(L_p + L)$ MLP neurons. Therefore, we need $3d(L_p + L)(1/\delta)^{d(L_p + L)}$ MLP neurons to construct the feed-forward layer $f_2^{(\text{FF})}$, so that each contextual embedding is mapped to the expected output of $h_{\text{seq2seq}}([P, \cdot])$. A bump function $f_{\text{bump}}$ for a quantized sequence $A \in \bar{\mathcal{G}}_\delta$ is written as:

$$f_{\text{bump}}(Q) = \frac{h([P, A])}{d(L_p + L)} \sum_{i=1}^{d} \sum_{j=1}^{L_p + L} \left[\text{ReLU}\left(K(Q_{i,j} - A_{i,j}) - 1\right) - \text{ReLU}\left(K(Q_{i,j} - A_{i,j})\right) \right.$$
$$\left. + \text{ReLU}\left(K(Q_{i,j} - A_{i,j}) + 1\right)\right],$$

where $Q \in \mathbb{R}^{d \times (L_p + L)}$ is some context ID scalar $K > 0$. Furthermore, recall that in (F.2), we have the relation of quantization granularity $\delta$ and function approximation error $\epsilon$ as $C\delta(dL)^{\frac{1}{\alpha}} \leq \epsilon/2$. We express the number of neurons in terms of $\epsilon$ as $\mathcal{O}\left(d(L_p + L)(C(dL)^{\frac{1}{\alpha}}/\epsilon)^{d(L_p + L)}\right) = \mathcal{O}\left(\epsilon^{-d(L_p + L)}\right)$, where $C$ is the Lipschitz constant and $\alpha$ is from the $\ell_\alpha$-norm we use for measuring the approximation error.

As a result, by choosing the appropriate step function approximation $\theta$, we arrive at

$$d_p\left(h_{\text{seq2seq}}([P, \cdot])_{:, L_p:}, \tau\right) \leq \epsilon/2.$$

This completes the proof.

$\square$

Finally, we arrive at the theorem that shows that prompt tuning on some transformers with single-head single-attention layer and two feed-forward layers is a universal approximator for sequence-to-sequence functions.

**Theorem G.1** (Theorem 2.4 Restated)**.** Let $1 \leq p < \infty$ and $\epsilon > 0$, there exist a transformer $\tau \in \mathcal{T}_B^{1,1,r}$ with single self-attention layer, $r = \mathcal{O}(d(L_p + L))$ MLP neurons and quantization granularity $\delta$, such that for any $f_{\text{seq2seq}} \in \mathcal{F}_C$ there exists a prompt $P \in \mathbb{R}^{d \times L_p}$ with

$$d_\alpha\left(\tau([P, \cdot])_{:, L_p}, f_{\text{seq2seq}}\right) \leq \epsilon.$$

*Proof of Theorem 2.4.* Combining Lemma F.1 and Lemma G.2, we arrive at a transformer $\tau \in \mathcal{T}_B^{1,1,r}$, with prompt $P \in \mathcal{G}_{\delta, L_p}$, such that for any sequence-to-sequence $f_{\text{seq2seq}} \in \mathcal{F}_C$,

$$d_\alpha\left(\tau\left([P, \cdot]\right)_{:, L_p:}, f_{\text{seq2seq}}\right)$$
$$\leq d_\alpha\left(\tau\left([P, \cdot]\right)_{:, L_p:}, h\left([P, \cdot]\right)_{:, L_p:}\right) + d_\alpha\left(h_{\text{seq2seq}}\left([P, \cdot]\right)_{:, L_p:}, f_{\text{seq2seq}}\right)$$
$$\leq \epsilon.$$

This completes the proof. $\square$

# H    PROOFS OF SECTION 2.5

In this section, we show the memorization capacity of prompt tuning on transformer networks with single layer self attention. We now prove that there exist a transformer $\tau \in \mathcal{T}_B^{1,1,r}$, such that for any dataset $S$, the transformer $\tau$ memorizes $S$ through prompt tuning.

## H.1    PROOF OF THEOREM 2.5

**Theorem H.1** (Theorem 2.5 Restated). Consider a dataset $S = \{(X^{(i)}, Y^{(i)})\}_{i=1}^N$, where $X^{(i)}, Y^{(i)} \in [0,1]^{d \times L}$. Assume the coresponding embedding sequences $Z^{(1)}, \dots, Z^{(N)}$ are generated from a $C$-Lipschitz function. Then, there exists a single-layer, single-head attention transformer $\tau \in \mathcal{T}_B^{1,1,r}$ with $r = \mathcal{O}\left((1/\epsilon)^{d(L_p+L)}\right)$ and a soft-prompt $P \in \mathbb{R}^{d \times L_p}$ such that, for any $i \in [N]$:

$$\left\| \tau([P, Z^{(i)}])_{:, L_p} - Y^{(i)} \right\|_\alpha \leq \epsilon,$$

where $L_p \geq L\lambda$, with $\lambda = \left(2\epsilon^{-1}C(dL)^{1/\alpha}\right)^{dL}$.

*Proof Sketch.* We first find some sequence-to-sequence function $f_{\text{seq2seq}}^\star : [0,1]^{d \times L} \mapsto [0,1]^{d \times L}$, such that for any $i \in [N]$, $f_{\text{seq2seq}}^\star\left(Z^{(i)}\right) = Y^{(i)}$. Next, we complete the proof by utilizing the results of Theorem 2.4 to construct a transformer $\tau \in \mathcal{T}_B^{1,1,r}$ that is capable of approximating $f_{\text{seq2seq}}^\star$ through prompt tuning. $\qquad\square$

*Proof of Theorem 2.5.* From the sequence-to-sequence function class $\mathcal{F}_C$, there exist some function $f_{\text{seq2seq}}^\star : [0,1]^{d \times L} \mapsto [0,1]^{d \times L}$ such that, $f_{\text{seq2seq}}^\star\left(Z^{(i)}\right) = Y^{(i)}$ for any $i \in [N]$.

Next, since we utilize positional encoding, no information would be lost in the quantization step of Theorem 2.4. By utilizing the results of Theorem 2.4, we construct a transformer $\tau \in \mathcal{T}_B^{1,1,r}$ such that

$$d_\alpha\left(\tau([P, \cdot])_{:, L_p}, f_{\text{seq2seq}}^\star\right) = \left(\int \left\| \tau([P, Z])_{:, L_p} - f_{\text{seq2seq}}^\star(Z) \right\|_\alpha^\alpha dZ\right)^{\frac{1}{\alpha}} \leq \epsilon.$$

As a result, we arrive at

$$\max_{i \in [N]} \left\| \tau([P, Z^{(i)}])_{:, L_p:} - Y^{(i)} \right\|_\alpha \leq \epsilon.$$

$\qquad\square$

## I  PROOFS OF COMPUTATIONAL LIMITS OF PROMPT TUNING (SECTION 3)

We first introduce some helper definition and lemmas from fine-grained complexity theory (Alman and Song, 2023).

**Definition I.1** (Approximate Attention Computation $\mathrm{AttC}(n, d, B, \epsilon_a)$, Definition 1.2 in (Alman and Song, 2023)). Let $\epsilon_a > 0$ and $B > 0$ be parameters. Given three matrices $Q, K, V \in \mathbb{R}^{n \times d}$, with the guarantees that $\|Q\|_{\max} \leq B$, $\|K\|_{\max} \leq B$, and $\|V\|_{\max} \leq B$, $\mathrm{AttC}(n, d, B, \epsilon_a)$ outputs a matrix $T \in \mathbb{R}^{n \times d}$ which is approximately equal to $\mathrm{Att}(Q, K, V) := D^{-1}AV$, meaning,

$$\|T - D^{-1}AV\|_{\max} \leq \epsilon_a, \quad \text{with } A := \exp(QK^\top) \text{ and } D := \mathrm{diag}(A\mathbb{1}_n)$$

Here, for a matrix $M \in \mathbb{R}^{n \times n}$, we write $\|M\|_{\max} := \max_{i,j} |M_{i,j}|$.

**Lemma I.1** (Fine-Grained Upper bound, Theorem 1.4 in (Alman and Song, 2023)). $\mathrm{AAttC}(n, d = \mathcal{O}(\log n), B = o(\sqrt{\log n}), \epsilon_a = 1/\mathrm{poly}(n))$ can be solved in time $\mathcal{T}_{\mathrm{mat}}(n, n^{o(1)}, d) = n^{1+o(1)}$.

**Lemma I.2** (Fine-Grained Lower bound, see Theorem 1.3 in (Alman and Song, 2023)). Assuming SETH, for every $q > 0$, there are constants $C, C_a, C_b > 0$ such that: there is no $\mathcal{O}(n^{2-q})$ time algorithm for the problem $\mathrm{AAttC}(n, d = C \log n, B = C_b \sqrt{\log n}, \epsilon_a = n^{-C_a})$.

### I.1  PROOF OF THEOREM 3.1

*Proof of Theorem 3.1.* Recall the Prompt Tuning Inference Problem APTI from Problem 1.

**Problem 1** (Approximate Prompt Tuning Inference $\mathrm{APTI}(d, L, L_p, \delta_F)$). Let $\delta_F > 0$ and $B > 0$. Given three $Q_p, K_p, V_p \in \mathbb{R}^{d \times (L + L_p)}$ with guarantees that $\|Q_p\|_{\max} \leq B$, $\|K_p\|_{\max} \leq B$ and $\|V_p\|_{\max} \leq B$, we aim to study an approximation problem $\mathrm{APTI}(d, L, L_p, B, \delta_F)$, that approximates $V_p \mathrm{Softmax}(K_p^\top Q_p)$ with a matrix $\widetilde{Z}$ such that $\|\widetilde{Z} - V_p \mathrm{Softmax}(K_p^\top Q_p)\|_{\max} \leq \delta_F$, where, for a matrix $M \in \mathbb{R}^{a \times b}$, we write $\|M\|_{\max} := \max_{i,j} |M_{i,j}|$.

We rewrite

$$V_p \mathrm{Softmax}(K_p^\top Q_p) = VD^{-1} \exp(K_p^\top Q_p).$$

By transpose-invariance property of $\|\cdot\|_{\max}$, we observe $\left\|\widetilde{Z} - V_p \mathrm{Softmax}(K_p^\top Q_p)\right\|_{\max} \leq \delta_F$ is equivalent to $\|T - D^{-1}AV\|_{\max}$ with the following identifications between APIT and ATTC:

- $(L_p + L) = n$, $d = d$, $B = B$, $\delta_F = \epsilon_a$

- $\widetilde{Z} = T$, $V_p = V$, $K_p = K$, $Q_p = Q$

By $\left\|[\cdot]_{:,L_p:}\right\|_{\max} \leq \|\cdot\|_{\max}$, we complete the proof via a simple reduction from fine-grained upper bound result Lemma I.1. $\square$

### I.2  PROOF OF THEOREM 3.2

*Proof of Theorem 3.2.* Using the same identifications as in the proof of Theorem 3.1, we complete the proof with Lemma I.2. $\square$

## J LIMITATIONS OF PROMPT TUNING TRANSFORMERS

In Section 2, we demonstrate that through prompt tuning, even a transformer with the simplest architecture can serve as a universal approximator. However, to achieve this, it is necessary to construct a specific transformer tailored for the task. In this section, we explore how prompts influence the output of a pretrained transformer model. Additionally, we investigate the boundaries of prompt tuning on arbitrary pretrained transformer model by analyzing its underlying mechanisms.

### J.1 DISCUSSION ON THE LIMITATIONS OF PROMPT TUNING

For simplicity, consider a single-layer transformer function class with 1 head of size $s$ and $r$ MLP hidden neurons:

$$\mathcal{T}_C^{1,s,r} := \{\tau : \mathbb{R}^{d \times L} \mapsto \mathbb{R}^{d \times L} | \tau = f^{(\mathrm{FF})}\left(f^{(\mathrm{SA})}(\cdot)\right)\}.$$

The tokenwise output of the transformer $\tau$ with input $[P, X] \in \mathbb{R}^{d \times (L_p + L)}$ is

$$\tau([P, X])_{:,i} = f^{(\mathrm{FF})}\left(f^{(\mathrm{Att})}([P, X]_{:,i}, [P, X]) + [P, X]_{:,i}\right),$$

where $[P, X]$ is the concatenation of a prompt $P \in \mathbb{R}^{d \times L_p}$ and a data $X \in \mathbb{R}^{d \times L}$. By taking the inverse of feed-forward function $f^{(\mathrm{FF}^{-1})} : \mathbb{R}^d \mapsto \mathbb{R}^d$, we have

$$f^{(\mathrm{Att})}(x, [P, X]) \in f^{(\mathrm{FF}^{-1})}(y) - x, \tag{J.1}$$

where $x = X_{:,i}$ and $y$ is the corresponding label token for $x$.

Next, to better understand how the prompt $P$ affect the output of the transformer, we focus on the output token of the attention layer corresponding to some data token $x = X_{:,i}$,

$$f^{(\mathrm{Att})}(x, [P, X]) \tag{J.2}$$
$$= W_O(W_V[P, X]) \operatorname{Softmax}\left[(W_K[P, X])^\top (W_Q x)\right]$$
$$= W_O(W_V[P, X]) \frac{\begin{bmatrix} \exp\left[(W_K[P, X]_{:,1})^\top (W_Q x)\right] \\ \vdots \\ \exp\left[(W_K[P, X]_{:,(L+L_p)})^\top (W_Q x)\right] \end{bmatrix}}{\sum_{j=1}^{L+L_p} \exp\left[(W_K[P, X]_{:,j})^\top (W_Q x)\right]}$$
$$= \frac{\sum_{i=1}^{L+L_p} W_O(W_V[P, X]_{:,i}) \exp\left[(W_K[P, X]_{:,i})^\top (W_Q x)\right]}{\sum_{j=1}^{L+L_p} \exp\left[(W_K[P, X]_{:,j})^\top (W_Q x)\right]}$$
$$= \frac{\sum_{i=1}^{L_p} \exp\left[(W_K P_{:,i})^\top (W_Q x)\right] f^{(\mathrm{Att})}(x, P)}{\sum_{j=1}^{L+L_p} \exp\left[(W_K[P, X]_{:,j})^\top (W_Q x)\right]} + \frac{\sum_{i=1}^{m} \exp\left[(W_K X_{:,i})^\top (W_Q x)\right] f^{(\mathrm{Att})}(x, X)}{\sum_{j=1}^{L+L_p} \exp\left[(W_K[P, X]_{:,j})^\top (W_Q x)\right]}$$
$$= \frac{\Psi(P, x)}{\Psi([P, X], x)} f^{(\mathrm{Att})}(x, P) + \frac{\Psi(X, x)}{\Psi([P, X], x)} f^{(\mathrm{Att})}(x, X),$$

where $\Psi(\cdot, \cdot, \cdot)$ is a positive scalar and defined as

$$\Psi(A, z) = \sum_i \exp\left((W_K A_{:,i})^\top (W_Q z)\right).$$

Combining (J.1) and (J.2), we have

$$\left( \frac{\Psi\left(P, x\right)}{\Psi\left([P, X], x\right)} f^{(\text{Att})}\left(x, P\right) + \frac{\Psi\left(X, x\right)}{\Psi\left([P, X], x\right)} f^{(\text{Att})}\left(x, X\right) \right) \in f^{(FF)-1}\left(y\right) - x. \qquad \text{(J.3)}$$

Essentially, with all parameters for the feed-forward and self-attention layers fixed, prompt tuning finds the prompt $P^\star$ such that (J.3) holds for each input-label pair $(x, y)$. In (J.3), note that while $\Psi(\cdot, \cdot, \cdot)$ are positive scalars, the attention terms $f^{(\text{Att})}(\cdot)$ are vectors. The initial term $\frac{\Psi(P,x)}{\Psi([P,X],x)} f^{(\text{Att})}(x, P)$ depends entirely on $P$, highlighting the strong effect of prompt tuning on shaping the model's outputs by guiding the attention mechanism. In contrast, $P$'s influence on the second term $\frac{\Psi(X,x)}{\Psi([P,X],x)} f^{(\text{Att})}(x, X)$ is limited to scaling, preserving the original attention pattern between $x$ and $X$. Thus, prompt tuning biases the attention function's output but does not alter the intrinsic attention pattern between $x$ and $X$.

This manipulation highlights prompt tuning's ability to subtly refine and leverage the pretrained model's knowledge without disrupting its core attention dynamics. However, it constrains prompt tuning's expressiveness, as it cannot change the direction of the attention output vector $f^{(\text{Att})}(x, X)$. Thus, prompt tuning is limited to realigning latent knowledge within the model, failing to learn new knowledge, which would require altering the model's core attention dynamics.

In Section 2.5, we discuss the cases where prompt tuning is able to memorize some general data set. Here, on the other hand, we also provide an example where prompt tuning on some general transformers fails to memorize some simple data set.

## J.2 EXAMPLES OF PROMPT TUNING FAILURES

The memorization ability in Theorem 2.5 is based on some specific transformers we carefully constructed for the memorization task. However, as we discussed in Appendix J, there exists limitations for prompt tuning on when learning new knowledge. Here, we provide an example where prompt tuning on some arbitrary transformers fails to memorize. We first introduce some assumptions on the relation between our transformer and dataset.

**Assumption J.1.** We assume that all output tokens $\left(Y^{(i)}\right)_{:,k}$ are in the range set of $f^{(\text{FF})}$. We assume that $W_Q, W_K, W_V, W_O$ are full rank matrices and that $f^{(\text{SA})}\left(X^{(i)}\right)$ are distinct for $i = 1, 2, \ldots, n$.

Now, we show that transformers through prompt tuning fails to memorize some simple data set.

**Corollary J.0.1** (Prompt Tuning Fails to Memorize, Theorem 2 of (Wang et al., 2023a)). For any pretrained single layer transformer $\tau \in \mathcal{T}$, there exist a sequence-to-sequence dataset $S = \left\{ \left( X^{(1)} = \left[x_1^{(1)}, x^\star\right], Y^{(1)} = \left[y_1^{(1)}, y_2^{(1)}\right] \right), \left( X^{(2)} = \left[x_1^{(2)}, x^\star\right], Y^{(2)} = [y_1^{(2)}, y_2^{(2)}] \right) \right\}$, and we cannot find a prompt $P \in \mathbb{R}^{d \times L_p}$ with any $L_p > 0$ such that $\tau\left([P, x_i]\right) = y_i$ holds for any $i = 1, 2$. The vectors $x_0, x_1, x_2$ are denoted post positional encodings.

**Remark J.1.** The most important aspect of this dataset is the shared token $x^\star$. As shown in Appendix J.1, to learn the first example $\left(X^{(1)}, Y^{(1)}\right)$, we are able to find a prompt $P$, such that

$$\left( \frac{\Psi\left(P, x^\star\right)}{\Psi\left([P, X^{(1)}], x^\star\right)} f^{(\text{Att})}\left(x^\star, P\right) + \frac{\Psi\left(X^{(1)}, x^\star\right)}{\Psi\left([P, X^{(1)}], x^\star\right)} f^{(\text{Att})}\left(x^\star, X^{(1)}\right) \right) \in f^{(FF)-1}\left(y_2^{(1)}\right) - x^\star.$$

However, now the vector $f^{(\text{Att})}\left(x^\star, P\right)$ is fixed as prompt $P$ has been chosen. This prevents us from finding a prompt to cater to the second example, which is written as

$$\left( \frac{\Psi\left(P, x^\star\right)}{\Psi\left([P, X^{(2)}], x^\star\right)} f^{(\text{Att})}\left(x^\star, P\right) + \frac{\Psi\left(X^{(2)}, x^\star\right)}{\Psi\left([P, X^{(2)}], x^\star\right)} f^{(\text{Att})}\left(x^\star, X^{(2)}\right) \right) \in f^{(FF)-1}\left(y_2^{(2)}\right) - x^\star.$$

Thus, the expressive power of prompt tuning is limited.

# K  Supplementary Proofs for Appendix D

Here we restate some proofs of the properties of Boltzmann operator from (Kajitsuka and Sato, 2024) for completeness.

## K.1  Lemma D.1

*Proof of Lemma D.1.* By taking $\ln$ on $p_i$ defined in Definition D.1, we see

$$\ln p_i = z_i - \ln \sum_{j=1}^{n} e^{z_j} = z_i - \ln \mathcal{Z}(z). \tag{K.1}$$

Also, by the definition of Boltz, we have

$$
\begin{aligned}
\mathrm{Boltz}(z) &= \sum_{i=1}^{n} z_i p_i \\
&= \sum_{i=1}^{n} p_i \ln \left( p_i \mathcal{Z}(z) \right) && \text{(By (K.1))} \\
&= \sum_{i=1}^{n} p_i \ln p_i + \sum_{i=1}^{n} p_i \ln \mathcal{Z}(z) \\
&= -\mathcal{S}(p) + \ln \mathcal{Z}(z).
\end{aligned}
$$

This completes the proof. $\qquad\square$

## K.2  Lemma D.2

*Proof of Lemma D.2.* We restate the proof from (Kajitsuka and Sato, 2024) for completeness.

We first observe that

$$
\begin{aligned}
\frac{\partial}{\partial z_j} p_i &= \frac{\partial}{\partial z_j} \left( \frac{e^{z_i}}{\sum_{k=1}^{n} e^{z_k}} \right) && \text{(K.2)} \\
&= \frac{\delta_{ij} e^{z_j} \left( \sum_{k=1}^{n} e^{z_k} \right) - e^{z_i} e^{z_j}}{\left( \sum_{k=1}^{n} e^{z_k} \right)^2} \\
&= \frac{\delta_{ij} e^{z_j}}{\sum_{k=1}^{n} e^{z_k}} - \frac{e^{z_i} e^{z_j}}{\left( \sum_{k=1}^{n} e^{z_k} \right)^2} \\
&= p_j \left( \delta_{ij} - p_i \right),
\end{aligned}
$$

where $\delta_{ij}$ is the delta function, i.e., $\delta_{ij} = 1$ only when $i = j$.

Next we have

$$
\begin{aligned}
\frac{\partial}{\partial z_i} \mathrm{Boltz}(z) &= \frac{\partial}{\partial z_i} \left( \sum_{j=1}^{n} z_j p_j \right) \\
&= \sum_{j=1}^{n} \frac{\partial z_j}{\partial z_i} p_j + \sum_{j=1}^{n} z_j \frac{\partial p_j}{\partial z_i} \\
&= p_i + \sum_{j=1}^{n} z_j p_i \left( \delta_{ji} - p_j \right) && \text{(By (K.2))} \\
&= p_i \left( 1 + z_i - \mathrm{Boltz}(z) \right) && \text{(By (D.1))} \\
&= p_i \left( 1 + z_i + \mathcal{S}(p) - \ln \mathcal{Z}(z) \right). && \text{(By Lemma D.1)}
\end{aligned}
$$

Since $p_i > 0$, we only need to focus on the second term

$$1 + z_i + \mathcal{S}(p) - \ln \mathcal{Z}(z) < 0.$$

This means

$$z_i < \ln \mathcal{Z}(z) - \mathcal{S}(p) - 1$$

By using $\max_{j \in [n]} z_j \le \ln \mathcal{Z}(z)$ (Boyd and Vandenberghe, 2004, p. 72) and $\mathcal{S}(p) \le \ln n$, we have that, when

$$z_i < \ln \mathcal{Z}(z) - \mathcal{S}(p) - 1,$$

is satisfied, the Boltzmann operator $\mathrm{Boltz}(z)$ monotonically decreases in the direction of $z_i$. $\quad\square$

## K.3  LEMMA D.3

*Proof of Lemma D.3.*  We restate the proof from (Kajitsuka and Sato, 2024) for completeness.

Observe that

$$\frac{\partial \mathcal{S}(p)}{\partial z_i} = \frac{\partial}{\partial z_i} \left( -\sum_{j=1}^{n} p_j \ln p_j \right) \tag{K.3}$$

$$= -\sum_{j=1}^{n} \frac{\partial p_j}{\partial z_i} \ln p_j + p_j \frac{\partial}{\partial z_i} \ln p_j$$

$$= -\sum_{j=1}^{n} p_i \left( \delta_{ji} - p_j \right) \ln p_j + p_i \left( \delta_{ji} - p_j \right) \qquad (\text{By (K.2)})$$

$$= -p_i \sum_{j=1}^{n} \left[ \delta_{ji} \left( \ln p_j + 1 \right) - p_j \ln p_j - p_j \right]$$

$$= -p_i \left( \ln p_i + 1 + \mathcal{S}(p) - 1 \right) \qquad (\text{By } \delta_{ii} = 1, \mathcal{S}(p) = \sum p_j \ln p_j, \sum p_j = 1)$$

$$= -p_i \left( \ln p_i + \mathcal{S}(p) \right).$$

Now, we prove the concavity by taking the derivative once again from Lemma D.2, which is

$$\frac{\partial^2}{\partial z_i^2} \mathrm{Boltz}(z) = \frac{\partial}{\partial z_i} p_i \left( 1 + \ln p_i + \mathcal{S}(p) \right) \qquad (\text{By Lemma D.2})$$

$$= \frac{\partial p_i}{\partial z_i} \cdot \left( 1 + \ln p_i + \mathcal{S}(p) \right) + p_i \cdot \frac{\partial}{\partial z_i} \left( 1 + \ln p_i + \mathcal{S}(p) \right)$$

$$= p_i \left( 1 - p_i \right) \left( 1 + \ln p_i + \mathcal{S}(p) \right) + p_i \left[ \frac{p_i \left( 1 - p_i \right)}{p_i} - p_i \left( \ln p_i + \mathcal{S}(p) \right) \right]$$

$$\qquad\qquad (\text{By (K.2) and (K.3)})$$

$$= p_i \left[ \left( 1 - 2p_i \right) \left( \ln p_i + \mathcal{S}(p) + 1 \right) + 1 \right]$$

$$= p_i \left[ \left( 1 - 2p_i \right) \left( z_i - \ln \mathcal{Z}(z) + \mathcal{S}(p) + 1 \right) + 1 \right] \qquad (\text{By (K.1)})$$

Since $p_i > 0$, we analyze the second term. Consider $p_i < \frac{1}{2}$, we have

$$z_i - \ln \mathcal{Z}(z) + \mathcal{S}(p) + 1 < \frac{-1}{1 - 2p_i}.$$

By using $\max_{j \in [n]} z_j \leq \ln \mathcal{Z}(z)$ (Boyd and Vandenberghe, 2004, p. 72) and $\mathcal{S}(p) \leq \ln n$, we have

$$z_i < \max_{j \in [n]} z_j - \ln n + \frac{-2 + 2p_i}{1 - 2p_i}.$$

Since $\frac{-2+2p_i}{1-2p_i}$ is unbounded below in domain $\frac{1}{2} > p_i > 0$, we focus on discussing cases where $\frac{1}{4} > p_i > 0$. We now have

$$-2 > \frac{-2 + 2p_i}{1 - 2p_i} < -3.$$

As a result, the Boltzmann operator $\mathrm{Boltz}(z)$ is concave with respect to $z_i$ for any

$$z_i < \max_{j \in [n]} z_j - \ln n - 3.$$

This completes the proof. $\qquad\square$

## K.4 LEMMA D.4

*Proof of Lemma D.4.* From Lemma D.2, we know that $\mathrm{Boltz}(z)$ monotonically decreases in the direction of $z_i$ when $z_i < z_1 - \ln n - 1$. Since $z$ is tokenwise $(\delta)$-separated and has no duplicate entry, given $z_1$, the minimum of $\mathrm{Boltz}(z)$ happens at $z^\star = (z_1, z_1 - \delta, z_1 - 2\delta, \ldots, z_1 - (n-1)\delta)$ where $\delta > \ln n + 1$. By Lemma D.2, we see that

$$\mathrm{Boltz}(z) > \mathrm{Boltz}(z^\star) > \mathrm{Boltz}(z').$$

$\qquad\square$

## K.5 LEMMA D.5

*Proof of Lemma D.5.* For any $z'$, we find some $z^\star \in \mathbb{R}^m$, where

$$z^\star = \left( z'_1, \ldots, z'_{m-1}, -\infty \right).$$

By Lemma D.2, we have

$$\mathrm{Boltz}(z^\star) > \mathrm{Boltz}(z').$$

In addition, for any $n$, we are able to find some $z^\star$ with last $(m - n)$ entries being $(-\infty)$. As a result, we have

$$\mathrm{Boltz}(z) = \mathrm{Boltz}(z^\star) > \mathrm{Boltz}(z').$$

$\qquad\square$

## K.6 LEMMA D.6

*Proof of Lemma D.6.* We restate the proof from (Kajitsuka and Sato, 2024) for completeness.

Let $a' \in \mathbb{R}^n$ be

$$a' = (a_1, a_1 - \delta, \ldots, a_1 - \delta). \tag{K.4}$$

From Lemma D.4, we know that $\mathrm{Boltz}(a) > \mathrm{Boltz}(a')$. In addition, we have:

$$\mathrm{Boltz}(a')$$

$$= \sum_{i=1}^{n} \left( a'_i \frac{e^{a'_i}}{\sum_{j=1}^{n} e^{a'_j}} \right)$$

$$= \frac{a_1 e^{a_1} + (n-1)(a_1 - \delta) e^{a_1 - \delta}}{e^{a_1} + (n-1)e^{a_1 - \delta}} \qquad \text{(By (K.4))}$$

$$= \frac{a_1 + (n-1)(a_1 - \delta) e^{-\delta}}{1 + (n-1)e^{-\delta}}$$

$$= a_1 - \frac{(n-1)\delta e^{-\delta}}{1 + (n-1)e^{-\delta}}.$$

Also, we know that $\text{Boltz}(b) \leq b_1$, since entries of $b$ is sorted in a decreasing order. Therefore,

$$\text{Boltz}(a) - \text{Boltz}(b)$$

$$\geq \text{Boltz}(a') - b_1$$

$$> a_1 - \frac{(n-1)\delta e^{-\delta}}{1 + (n-1)e^{-\delta}} - (a_1 - \delta) \qquad \text{(By } b_1 < a_1 - \delta \text{ )}$$

$$= \delta - \frac{(n-1)\delta e^{-\delta}}{1 + (n-1)e^{-\delta}}$$

$$= \frac{\delta}{1 + (n-1)e^{-\delta}} \qquad \text{(By } \delta > 2\ln n + 3.\text{)}$$

$$\geq \ln n.$$

Note that $\ln n > (\ln n)^2 e^{-(a_1 - b_1)}$, because $a_1 - b_1 > \ln n$ implies $\ln n \cdot e^{-(a_1 - b_1)} < 1$. $\qquad \square$

## K.7 LEMMA D.7

*Proof of Lemma D.7.* We restate the proof from (Kajitsuka and Sato, 2024) for completeness.

With the concavity given in Lemma D.3 and first-order Taylor approximation, we have

$$\text{Boltz}(b_1, \ldots, b_{n-1}, t) + (a_n - t) \cdot \frac{\partial}{\partial t} \text{Boltz}(b_1, \ldots, b_{n-1}, t) > \text{Boltz}(b_1, \ldots, b_{n-1}, a_n),$$

for $t < a_n$.

Then, by setting $t = b_n$, we obtain

$$\text{Boltz}(b_1, \ldots, b_{n-1}, t) - \text{Boltz}(b_1, \ldots, b_{n-1}, a_n)$$

$$= \text{Boltz}(b) - \text{Boltz}(a)$$

$$> (a_n - b_n) \left( -\frac{\partial}{\partial t} \text{Boltz}(b_1, \ldots, b_{n-1}, t) \Big|_{t=b_n} \right)$$

$$= (a_n - b_n) \left[ -p_n (1 + \ln p_n + \mathcal{S}(p)) \right] \qquad \text{(By Lemma D.2)}$$

$$> (a_n - b_n) \left[ -p_n \left( 1 + b_n - \max_{i \in [n]} b_i + \ln n \right) \right]$$

$$> (a_n - b_n) p_n (\delta + a_n - b_n - \ln n - 1)$$

$$= (a_n - b_n) \frac{e^{b_n}}{\sum_{i=1}^{n} e^{b_i}} (\delta + a_n - b_n - \ln n - 1).$$

This completes the proof. $\qquad \square$

## K.8  LEMMA D.8

*Proof of Lemma D.8.* We restate the proof from (Kajitsuka and Sato, 2024) for completeness.

Let

$$a_{\mathrm{up}} := (a_1, a_2, \ldots, a_k, a_{k+1}) \in \mathbb{R}^{k+1},$$
$$b_{\mathrm{lo}} := (a_1, a_2, \ldots, a_k, b_{k+1}, b_{k+1}, \ldots, b_{k+1}) \in \mathbb{R}^n.$$

Then, Lemma D.2 implies that

$$\mathrm{Boltz}(a) < \mathrm{Boltz}(a_{\mathrm{up}}),$$
$$\mathrm{boltz}(b) > \mathrm{Boltz}(b_{\mathrm{lo}}).$$

Thus we only have to bound $\mathrm{Boltz}(b_{\mathrm{lo}}) - \mathrm{Boltz}(a_{\mathrm{up}})$.

Let

$$\gamma_k := \sum_{l=1}^{k} a_l e^{a_l} \quad \text{and} \quad \xi_k := \sum_{l=1}^{k} e^{a_l}.$$

Next, decompose $\mathrm{Boltz}(b_{\mathrm{lo}})$:

$$
\begin{aligned}
\mathrm{Boltz}(b_{\mathrm{lo}}) &= \frac{\gamma_k + (n-k)b_{k+1}e^{b_{k+1}}}{\xi_k + (n-k)e^{b_{k+1}}} \\
&= \frac{\gamma_k + b_{k+1}e^{b_{k+1}+\ln(n-k)}}{\xi_k + e^{b_{k+1}+\ln(n-k)}} \\
&= \frac{\gamma_k + \left(b_{k+1}+\ln(n-k)\right)e^{b_{k+1}+\ln(n-k)}}{\xi_k + e^{b_{k+1}+\ln(n-k)}} - \frac{\ln(n-k)\cdot e^{b_{k+1}+\ln(n-k)}}{\xi_k + e^{b_{k+1}+\ln(n-k)}} \\
&= \mathrm{Boltz}\left(a_1, \ldots, a_k, b_{k+1}+\ln(n-k)\right) - \frac{\ln(n-k)\cdot e^{b_{k+1}+\ln(n-k)}}{\xi_k + e^{b_{k+1}+\ln(n-k)}}.
\end{aligned}
$$

Therefore, we have

$$
\begin{aligned}
&\mathrm{Boltz}(b_{\mathrm{lo}}) - \mathrm{Boltz}(a_{\mathrm{up}}) \\
&= \mathrm{Boltz}\left(a_1, \ldots, a_k, b_{k+1}+\ln(n-k)\right) - \mathrm{Boltz}(a_{\mathrm{up}}) - \frac{\ln(n-k)\cdot e^{b_{k+1}+\ln(n-k)}}{\xi_k + e^{b_{k+1}+\ln(n-k)}}.
\end{aligned}
\tag{K.5}
$$

Note that by Lemma D.7, we also have

$$
\begin{aligned}
&\mathrm{boltz}\left(a_1, \ldots, a_k, b_{k+1}+\ln(n-k)\right) - \mathrm{Boltz}(a_{\mathrm{up}}) \\
&> \left(a_{k+1} - b_{k+1} - \ln(n-k)\right)\left(\delta + a_{k+1} - b_{k+1} - \ln(n-k) - \ln(k+1) - 1\right) \\
&\quad\cdot \frac{e^{b_{k+1}+\ln(n-k)}}{\xi_k + e^{b_{k+1}+\ln(n-k)}} \\
&> (\delta - \ln n)(2\delta - 2\ln n - 1)\cdot \frac{e^{b_{k+1}+\ln(n-k)}}{\xi_k + e^{b_{k+1}+\ln(n-k)}}. &&(\text{By } \delta\text{-separatedness}) \\
&> 4\ln^2(n)\cdot \frac{e^{b_{k+1}+\ln(n-k)}}{\xi_k + e^{b_{k+1}+\ln(n-k)}}. &&(\text{By assumption } \delta > 4\ln n)
\end{aligned}
\tag{K.6}
$$

Now we plug (K.6) into (K.5) to obtain

$$\mathrm{Boltz}(b_{\mathrm{lo}}) - \mathrm{Boltz}(a_{\mathrm{up}})$$

$$= \text{Boltz}\left(a_1, \ldots, a_k, b_{k+1} + \ln(n-k)\right) - \text{Boltz}(a_{\text{up}}) - \frac{\ln(n-k) \cdot e^{b_{k+1} + \ln(n-k)}}{\xi_k + e^{b_{k+1} + \ln(n-k)}}$$

$$> \frac{e^{b_{k+1} + \ln(n-k)}}{\xi_k + e^{b_{k+1} + \ln(n-k)}} \cdot \left(4 \ln^2(n) - \ln(n-k)\right)$$

$$> \frac{e^{b_{k+1} + \ln(n-k)}}{\xi_k + e^{b_{k+1} + \ln(n-k)}} \cdot 2 \ln^2(n).$$

Also, for the denominator, we have

$$\xi_k + e^{b_{k+1} + \ln(n-k)} < \sum_{l=1}^{k+1} e^{a_l} \qquad\qquad (\text{By } a_{k+1} > b_{k+1} + \ln(n-k))$$

$$< e^{a_1} \sum_{l=1}^{k+1} e^{-(l-1)\delta} \qquad\qquad (\text{By } a_l < a_1 - (l-1)\delta)$$

$$< 2 e^{a_1}. \qquad\qquad (\text{By } \delta > \ln 2)$$

Therefore, we arrive at

$$\text{Boltz}(\bar{b}) - \text{Boltz}(a_{\text{up}}) > \frac{e^{b_{k+1} + \ln(n-k)}}{\xi_k + e^{b_{k+1} + \ln(n-k)}} \cdot 2(\ln n)^2$$

$$> \frac{e^{b_{k+1} + \ln(n-k)}}{2 e^{a_1}} \cdot 2(\ln n)^2$$

$$> (\ln n)^2 e^{-(a_1 - b_{k+1})}.$$

This implies that

$$\text{Boltz}(b) - \text{Boltz}(a) > (\ln n)^2 e^{-(a_1 - b_{k+1})}.$$

This completes the proof. $\qquad\qquad\qquad\qquad\qquad\qquad\qquad\qquad\qquad\qquad\qquad\qquad \square$

## K.9 LEMMA D.9

*Proof of Lemma D.9.* We restate the proof from (Kajitsuka and Sato, 2024) for completeness.

First, we observe that $\text{Boltz}$ is permutation invariant by definition. In addition, there are no duplicate entries in each vector $z_i$. Therefore, w.l.o.g. we write the vectors in entrywise decreasing order $z_1^{(i)} > \ldots > z_n^{(i)}$ for any $i \in [N]$. We prove (D.3) by utilizing the first constraint of $(\gamma, \delta)$-tokenwise separateness of $z^{(i)}$, which is

$$\left| z_s^{(i)} \right| < \gamma,$$

for any $i \in [N]$ and $s \in [n]$. Since $z_n^{(i)} < \text{Boltz}(z^{(i)}) < z_1^{(i)}$, we have

$$\left| \text{Boltz}(z^{(i)}) \right| < \max\left( \left| z_1^{(i)} \right|, \left| z_n^{(i)} \right| \right) < \gamma.$$

Next, we prove the $\delta'$-separateness. Consider $i \in [N]$ and $s \in [n]$, w.l.o.g. we assume that there exists $k \in \{0, \ldots, n-1\}$ such that

$$\left( z_1^{(i)}, \ldots, z_k^{(i)} \right) = \left( z_1^{(j)}, \ldots, z_k^{(j)} \right) \quad \text{and} \quad a_{k+1} > b_{k+1}.$$

Then, by combining Lemma D.8 and Lemma D.6, we have

$$
\begin{aligned}
&|\mathrm{Boltz}(z^{(i)}) - \mathrm{Boltz}(z^{(j)})| \\
&> (\ln n)^2 e^{-\left(z_1^{(i)} - z_{k+1}^{(j)}\right)} \\
&> (\ln n)^2 e^{-2\gamma}. \qquad\qquad (a_1 - b_{k+1} < 2r \text{ since } (\gamma, \delta)\text{-separated})
\end{aligned}
$$

This completes the proof. □

## K.10   LEMMA E.1

*Proof of Lemma E.1.* We restate the proof from (Park et al., 2021) for completeness.

We first note that the second inequality is simple because $u$ is a unit vector. Next, we prove the first inequality. We focus on the cases where $|\mathcal{X}| = N \geq 2$ and $d \geq 2$. We first prove that for any vector $v \in \mathbb{R}^d$, a unit vector $u \in \mathbb{R}^d$ uniformly randomly drawn from the hypersphere $\mathbb{S}^{d-1}$ satisfies

$$
\Pr\left(|u^\top v| < \frac{\|v\|}{N^2}\sqrt{\frac{8}{\pi d}}\right) < \frac{2}{N^2}. \tag{K.7}
$$

With (K.7), we define $\mathcal{V} := \{x - x' : x, x' \in \mathcal{X}\}$. Then, the union bound implies

$$
\begin{aligned}
\Pr\left(\bigcup_{v \in \mathcal{V}}\left\{|u^\top v| < \frac{\|v\|}{N^2}\sqrt{\frac{8}{\pi d_x}}\right\}\right) &\leq \sum_{v \in \mathcal{V}} \Pr\left(|u^\top v| < \frac{\|v\|}{N^2}\sqrt{\frac{8}{\pi d_x}}\right) \\
&< \frac{N(N-1)}{2} \cdot \frac{2}{N^2} < 1,
\end{aligned}
$$

and thus there exists at least one unit vector $u$ that satisfies the lower bound.

We start the prove with

$$
\begin{aligned}
&\Pr\left(|u^\top v| < \frac{\|v\|}{N^2}\sqrt{\frac{8}{\pi d}}\right) \\
&= \Pr\left(|u_1| < \frac{1}{N^2}\sqrt{\frac{8}{\pi d}}\right) \\
&= 2\Pr\left(0 < u_1 < \frac{1}{N^2}\sqrt{\frac{8}{\pi d}}\right) \qquad\text{(By symmetry of the uniform distribution)} \\
&= \frac{2}{\mathrm{Area}\left(\mathbb{S}^{d-1}\right)} \cdot \int_{\cos^{-1}\left(\frac{1}{N^2}\sqrt{\frac{8}{\pi d}}\right)}^{\frac{\pi}{2}} \mathrm{Area}\left(\mathbb{S}^{d-2}\right) \cdot (\sin(\phi))^{d-2}\mathrm{d}\phi \\
&= 2 \cdot \frac{\mathrm{Area}\left(\mathbb{S}^{d-2}\right)}{\mathrm{Area}\left(\mathbb{S}^{d-1}\right)} \cdot \int_{\cos^{-1}\left(\frac{1}{N^2}\sqrt{\frac{8}{\pi d}}\right)}^{\frac{\pi}{2}} (\sin(\phi))^{d-2}\mathrm{d}\phi \\
&= \frac{2}{\sqrt{\pi}} \cdot \frac{(d-1)\,\Gamma\left(\frac{d}{2}+1\right)}{d\,\Gamma\left(\frac{d}{2}+\frac{1}{2}\right)} \cdot \int_{\cos^{-1}\left(\frac{1}{N^2}\sqrt{\frac{8}{\pi d}}\right)}^{\frac{\pi}{2}} (\sin(\phi))^{d-2}\mathrm{d}\phi \\
&< \sqrt{\frac{2}{\pi}} \cdot \frac{(d-1)\sqrt{d+2}}{d} \cdot \int_{\cos^{-1}\left(\frac{1}{N^2}\sqrt{\frac{8}{\pi d}}\right)}^{\frac{\pi}{2}} 1\mathrm{d}\phi \qquad\text{(By Gautschi inequality and } \sin(\pi) \leq 1) \\
&\leq \sqrt{\frac{2d}{\pi}} \int_{\cos^{-1}\left(\frac{1}{N^2}\sqrt{\frac{8}{\pi d}}\right)}^{\frac{\pi}{2}} 1\mathrm{d}\phi \qquad\text{(Since } d \geq 1)
\end{aligned}
$$

$$= \sqrt{\frac{2d}{\pi}} \left( \frac{\pi}{2} - \cos^{-1} \left( \frac{1}{N^2} \sqrt{\frac{8}{\pi d}} \right) \right)$$

$$= \sqrt{\frac{2d}{\pi}} \sin^{-1} \left( \frac{1}{N^2} \sqrt{\frac{8}{\pi d}} \right)$$

$$\leq \sqrt{\frac{2d}{\pi}} \cdot \frac{\pi}{2} \cdot \frac{1}{N^2} \sqrt{\frac{8}{\pi d}}$$

$$= \frac{2}{N^2}. \qquad\qquad \left( \phi \leq \frac{\pi}{2} \sin(\phi), \forall 0 \leq \phi \leq \frac{\pi}{2} \right)$$

This completes the proof. $\qquad\qquad\square$

