# OpenReview forum: "Fundamental Limits of Prompt Tuning Transformers: Universality, Capacity and Efficiency"
_ICLR.cc/2025/Conference — ICLR 2025 Poster_

### Official Review · Reviewer_8DjM · 2024-10-31

**Soundness:** 3
**Presentation:** 2
**Contribution:** 2
**Rating:** 8
**Confidence:** 3

**Summary:**

This paper considers the approximation power of the transformer architecture. The authors prove that single-head, single-layer attention is a uniform approximator of a certain class of Lipschitz functions over the space of sequences of tokens. The authors also provide results regarding the ability of transformers to memorize a dataset, where they improve on previous results regarding the number of fully connected layers needed to memorize. Finally, they provide a complexity-theory style result regarding the efficiency of prompt tuning. Their result indicates that prompt tuning is possible in nearly linear time under several assumptions.

**Strengths:**

* The paper is well-written, other from a few minor typos. There is clear effort to clearly lay out the problems and their solutions, including statements of intermediate lemmas.
* The problem clearly addresses an important problem. It's worth knowing the approximation power of the transformer architecture, given its widespread use.
* The results also improve on the previous work of (Wang et al., 2023) in a non-trivial way.

**Weaknesses:**

**Organization and presentation.**
* In general, it's inarguable that this is not an easy paper to read. This is the nature of universal approximation style papers; the technical assumptions needed for this work can be daunting. However, there are a few things that I think the authors could do to make this paper appealing to more "casual" readers.
    * Rather than trying to state the theorems in full detail, opt for informal statements, with the rigorous version deferred to the appendix. This would likely lighten the load in the main text.
    * No matter how tempting, do not use \vspace{} with a negative argument. The paper is hard enough to read as is; decreasing the skips between lines does not help this whatsoever.
    * Try, whenever possible, to include equations on their own lines. There too many long-ish equations displayed in-line in this paper. As a result, I had a hard time reading this paper; I expect others will feel similarly.
* The paper could also benefit from slightly more thoughtfulness around the notation used.
    * Define notation **before** it is used. The authors use a lot of notation in the introduction that is defined in the next section.
    * Similarly, there is confusion about the indexing variable $p$. My understanding is that this variable is used both as an indexing variable for token sequences of length or index $p$, and as the norm-type (e.g., for norms on an $L^p$ functional space). Using separate variables would improve readability.
    * Using (capital) $Z$ and (lower-case) $s$ to denote the hidden dimension is confusing. Perhaps it would be better to stick to either capital or lower-case letters when they play the same role.
   * This may just be a typo, but in Problem 2, $d_p$ takes three arguments the first time it's used, but it is defined later on (in the next sentence) as having only two arguments. I think perhaps there is a missing underscore before the $L_p$, so it ends up being $\tau([P,\cdot])_{:,L_p}$.
    * The word vocabulary is a bit overloaded. It generally is used to mean the codomain of the tokenizer. But here it's more like the set of columns of the datasets $Z^(i)$. It would be worth disambiguating this.

**Assumptions.**
* This paper would benefit from expanding on why each of its assumptions is reasonable. This is lacking throughout the paper. One could argue that this doesn't matter for universal approximation results. And yet, the fact remains that these results would be stronger and more useful if the authors made clear whether the assumptions hold in cases closer to real-world problems.  Some further comments:
    * Is the class of seq2seq Lipschitz functions a reasonable class of functions to consider? The definition under Def. 2.3 indicates that the functions are Lipschitz over the space of tokens unless I've misunderstood something. But why should we expect the space of tokens to admit local structure that makes Lipschitz meaningful? In general, tokens in the Euclidean neighborhood of other tokens may not share any semantic similarity. It makes me wonder whether there might be a better class of functions for a universality result.
    * I have a similar concern about the token separation assumption in Def. 2.5. This seems a bit unreasonable given the possibility that the dataset may contain repeated queries. It would be worth explaining how reliant the theory is on these results.

**Miscellaneous.**
* The section on computational limits (Section 3) felt relatively rushed to me. It wasn't clear why this problem is related to $k$-SAT, or what the reduction is that powered Theorem 3.2. The original limit was 10 pages, so perhaps the additional page of content could be used to expand on this part.
* The line at the end that says "By the formal nature of this work, our results do not lead to practical implementations. However, we anticipate that our findings will offer valuable insights to future prompt tuning methods" was disappointing. Not only is there no attempt to ground these ideas in real world settings, but the statement regarding the impact of this work is, in a word, vague. Is there not more that the authors can say here? This is an opportunity to say why you think that this research is impactful. And if you don't think that this paper will be impactful, it's worth asking why it was carried out in the first place. Of course, no one is expecting a paper on uniform approximation to be immediately applicable to modern LLMs. However, it would be worth speculating for at least a paragraph about why you think these results will be helpful moving forward, i.e., what kinds of "valuable insights" do you think this work will yield?

**Questions:**

See above

---

> ### Author Response · Authors · 2024-11-19
> **Response 1 (Organization and presentation)**
>
> ### Thank you for your detailed review.
> ### We have addressed all your comments and questions in this and the following responses.
> ### We have also modified the draft accordingly and marked the changes in **blue** in the latest revision.
> ### Please see the updated PDF for details.
>
> ---
>
> > **Reviewer's Comment:** Rather than trying to state the theorems in full detail, opt for informal statements, with the rigorous version deferred to the appendix. This would likely lighten the load in the main text.
>
> Thank you for the suggestion. We agree that informal statements enhance clarity.
>
> In response, **we have revised the “Overview of Our Results” on page 3 into informal statements of our main results (see `line133-148` of the latest revision).** This change offers a clearer high-level overview and makes the subsequent discussion, “Comparing with Prior Works,” more accessible. We believe this reduces the load for readers to follow the subsequent sections.
>
> > **Reviewer's Comment:**  No matter how tempting, do not use \vspace{} with a negative argument. The paper is hard enough to read as is; decreasing the skips between lines does not help this whatsoever.
>
> Thanks for the suggestion. Our original intention was to keep the paper within 9 pages to avoid it feeling too lengthy as a 10-page submission.  However, we now realize this approach backfired, affecting readability. **We have removed all the \vspace now.**
>
> We believe the overall readability of the latest revision is much better now. Please see updated PDF for details.
>
> > **Reviewer's Comment:**   Try, whenever possible, to include equations on their own lines. There too many long-ish equations displayed in-line in this paper. As a result, I had a hard time reading this paper; I expect others will feel similarly.
>
> Thanks for the suggestion. We have incorporated this change where possible. However, given the dense nature of our results—especially with the newly added page of extended discussion in response to your later question—we couldn’t apply it uniformly. Nevertheless, we agree that these adjustments improve the overall readability of the draft.
>
> > **Reviewer's Comment:**   Define notation before it is used. The authors use a lot of notation in the introduction that is defined in the next section.
>
> Thanks for pointing this out. This issue arose from rearranging sections during preparation. We apologize for the oversight. **We have conducted three additional rounds of proofreading and addressed this with our utmost effort.**
>
> > **Reviewer's Comment:**  Similarly, there is confusion about the indexing variable $p$. My understanding is that this variable is used both as an indexing variable for token sequences of length or index $p$, and as the norm-type (e.g., for norms on an $L^p$ functional space). Using separate variables would improve readability.
>
> Thanks for pointing this out. This shared notation is indeed ambiguous. **We have change $p$-norm to $\alpha$-norm and modified the draft accordingly.**
>
> > **Reviewer's Comment:**  Using (capital) $Z$ and (lower-case) $s$ to denote the hidden dimension is confusing. Perhaps it would be better to stick to either capital or lower-case letters when they play the same role.
>
> Sorry for the confusion caused.  **We meant to denote input embeddings with $Z\in d \times L$ and $s$ for hidden dimension.** We have clarified these notations in `line 183`.
>
> > **Reviewer's Comment:**  This may just be a typo, but in Problem 2, …takes three arguments the first time it's used, but it is defined later on (in the next sentence) as having only two arguments. I think perhaps there is a missing underscore before the, so it ends up being …
>
> Thanks for your careful proofreading. **This is indeed a typo. We have updated accordingly.**
>
> > **Reviewer's Comment:**  The word vocabulary is a bit overloaded. It generally is used to mean the codomain of the tokenizer. But here it's more like the set of columns of the datasets. It would be worth disambiguating this.
>
> Sorry for the confusion. We follow the conventions used in [Yun et al., 2020], [Wang et al., 2023a], and [Kajitsuka and Sato, 2024]. However, we agree with your point and have added a note at `line 239` to clarify this.

---

> ### Author Response · Authors · 2024-11-19
> **Response 2 (Assumptions)**
>
> > **Reviewer's Comment:** Is the class of seq2seq Lipschitz functions a reasonable class of functions to consider? The definition under Def. 2.3 indicates that the functions are Lipschitz over the space of tokens unless I've misunderstood something. But why should we expect the space of tokens to admit local structure that makes Lipschitz meaningful? In general, tokens in the Euclidean neighborhood of other tokens may not share any semantic similarity. It makes me wonder whether there might be a better class of functions for a universality result.
> >  **Reviewer's Comment:** I have a similar concern about the token separation assumption in Def. 2.5. This seems a bit unreasonable given the possibility that the dataset may contain repeated queries. It would be worth explaining how reliant the theory is on these results.
>
>
> Thank you for your question. Here are some clarifications:
>
> Yes, the **Lipschitz function class is reasonable to consider for seq2seq models**, because token embeddings are designed to capture semantic proximity, making **Lipschitz continuity meaningful in embedding space.** Sorry for the confusion as our notation design is not ideal. **We need to differentiate between embeddings $Z$ and raw data sequence $X$ and the Lipschitz assumption applies to $Z$, not $X$.** We have modified the draft to distinct the two.
>
> **The tokenwise separatedness is achieved by adding positional encoding to the input sequence.**  The assumption is made on input of attention layers (i.e., $Z$), instead of raw data $X$. We do not make extra assumption on the data $X$.

---

> ### Author Response · Authors · 2024-11-19
> **Response 3 (Miscellaneous)**
>
> > **Reviewer's Comment:** The section on computational limits (Section 3) felt relatively rushed to me. It wasn't clear why this problem is related to k-SAT, or what the reduction is that powered Theorem 3.2. The original limit was 10 pages, so perhaps the additional page of content could be used to expand on this part.
>
>
> Since our computational analysis is build on fine-grained complexity theory, in which, SETH is a fundamental conjecture. It’s a key hypothesis for the lower bound results of Sec 3. Specifically, We require SETH to facilitate fine-grained reduction to prove the inefficiency threshold Theorem 3.1.
>
>
> The key idea is to show that, outside a certain norm-bound region, **prompt tuning inference is at least as hard as the k-SAT problem and therefore cannot be solved in quadratic time.** To show this, we do not use the k-SAT problem directly. Instead, we rely on prior results from [Alman and Zhao (2023)] — attention is at least as hard as k-SAT. We demonstrate that our Problem 1 is equally challenging  outside the specified norm-bound region, confirming the quadratic time limitation.
>
> To provide more background for k-SAT, k-SAT (k-Satisfiability) problems involve determining if there is an assignment of boolean variables that satisfies a Boolean formula in conjunctive normal form, where each clause has exactly $k$ literals. **k-SAT is NP-complete for $k \geq 3$.**
>
> **Importance in SETH:** The Strong Exponential Time Hypothesis (SETH) posits that **no algorithm can solve k-SAT in time $O(2^{(1-\varepsilon)n})$** for any $\varepsilon > 0$ and sufficiently large $k$. k-SAT is crucial in SETH because:
> * Complexity Bounds: SETH provides strong lower bounds for the time complexity of k-SAT, which influences the complexity assumptions for many other problems.
> * Reductions: Many hardness results are derived by reducing k-SAT to other problems, using SETH as a basis to show these problems cannot be solved faster.
> * Algorithmic Impact: Improved k-SAT algorithms could refute SETH, reshaping our understanding of computational complexity.
>
> Therefore, it’s common and natural to state SETH with k-SAT problems in literature.
>
> In response to your suggestion about adding the SETH discussion to page 10, we opted to use that space for remarks on practical implications and made room by removing all \vspace commands.
>
> If you find it appropriate, we are happy to include the discussion in the appendix instead. Thank you!

---

> ### Author Response · Authors · 2024-11-19
> **Response 3-2 (Miscellaneous 2)**
>
> > **Reviewer's Comment:** The line at the end that says "By the formal nature of this work, our results do not lead to practical implementations. However, we anticipate that our findings will offer valuable insights to future prompt tuning methods" was disappointing. Not only is there no attempt to ground these ideas in real world settings, but the statement regarding the impact of this work is, in a word, vague. Is there not more that the authors can say here? This is an opportunity to say why you think that this research is impactful. And if you don't think that this paper will be impactful, it's worth asking why it was carried out in the first place. Of course, no one is expecting a paper on uniform approximation to be immediately applicable to modern LLMs. However, it would be worth speculating for at least a paragraph about why you think these results will be helpful moving forward, i.e., what kinds of "valuable insights" do you think this work will yield?
>
>
> Thank you for your question.
>
> **We have extended concluding remarks discussing the practical implications of our theoretical results.** Please also see **`page 10`** of the revised PDF for more details. We quote the below for your reference.
>
> ### **Practical Implications from Statistical Limits (Section 2).**
>
>  We analyze the universality of prompt tuning transformers with minimal structures, and its memorization capacity on general datasets.
>
> * **Universality (Theorem 2.2).** Our results show that the universality of prompt tuning pretrained transformer is achievable on as simple as a single-layer, single-head attention transformers. This demonstrates that universality in prompt-tuning isn’t limited to large, complex foundation models.
>
> * **Width-Depth Tradeoff (Section 2.4).** Our results highlight a trade-off in the design choices for the depth and width of FFN (MLP) layers:
>     1. $\mathcal{O}((1/\epsilon)^{d(L+L_p)}$ FFN layers of width $4$ or
>     2. $2$ FFN layers of width  $\mathcal{O}((1/\epsilon)^{d(L+L_p)}$.
>
>     In practice, 1. and 2. differ in memory usage, parallelization, and optimization preferences, leading to distinct application scenarios.
>
> * **Memorization (Section 2.5).** Our memorization results apply to general datasets, whereas prior results are limited to specialized cases. This makes our results go beyond specialized theoretical analysis and align more with practical applications with a suggested *long* soft-prompt length.
>
>
> ### **Practical Implications from Computational Limits (Section 3).**
>
> We analyze the $\mathcal{O}(L^2)$ bottleneck of prompt tuning transformers and provides useful guidance for designing efficient prompt tuning (approximation) methods with precision guarantees. Let $Q_p=W_QX_p$, $K_p=W_KX_p$, and $V_p=W_VX_p$ with $X_p=[P,X]\in\mathbb{R}^{d\times (L_p+L)}$. Here $L$ and $L_p$ are the input and soft-prompt length.
>
> * **Self- and Cross-Attention.** Our computational results apply to both self-attention and cross-attention prompt tuning. This is because the norm bound conditions depend on $\max\{|Q_p|, |K_p|, |V_p|\}$, which are valid for both self- and cross-attention inputs.
>
> * **Necessary Conditions for Subquadratic Prompt Tuning (Theorem 3.1).** Our result suggests proper normalization on soft-prompt and weight matrices are required to ensure subquadratic prompt tuning inference, i.e.,  $\max\{\|Q_p\|, \|K_p\|,\|V_p\|\}\le \mathcal{O}(\sqrt{\log(L_p+L)})$.
>
> * **Necessary Conditions for Almost Linear Time Prompt Tuning (Theorem 3.2).** Our result suggests more strict normalization on soft-prompt and weight matrices are required to ensure almost linear time prompt tuning inference, i.e.,$\max\{\|Q_p\|, \|K_p\|,\|V_p\|\}\le o(\sqrt{\log(L_p+L)})$.
>
> ---
>
> We sincerely appreciate the time and effort you invested in reviewing our paper, especially your attention to detials! Your feedback helped us resolve ambiguities in our notation.
>
> We have carefully considered all your comments and made the necessary revisions to address your concerns.
>
> Please feel free to reach out if you have any further questions or need additional clarification about our work. Thank you!

---

> ### Author Response · Authors · 2024-11-24
> **A Gentle Reminder**
>
> Dear Reviewer,
>
> As the rebuttal phase nears its end, we wanted to gently remind you of our responses.
>
> We have addressed all your comments and questions with the utmost care and hope our efforts meet your expectations.
>
> If you have any remaining concerns, we would be happy to discuss them further. Otherwise, we kindly invite you to consider raising your score if you find our updates satisfactory.
>
> Thank you for your time and thoughtful review!

---

> ### Comment · Reviewer_8DjM · 2024-11-28
> **Response**
>
> Thanks so much for your thorough rebuttal. This addressed nearly all of my concerns. I still don't agree with this: "Yes, the Lipschitz function class is reasonable to consider for seq2seq models, because token embeddings are designed to capture semantic proximity." Yes, ideally, embeddings would be contextual. But I don't think it reasonable to assume that a neighborhood in embedding space would *really* capture all of the semantic nuances of which pieces of language are related to one another. That being said, I don't disagree that for the sake of theory, this is a fairly reasonable way to study transformers.
>
> Overall, since the authors fixed the problems identified in my review, there's no reason to jam this paper up any longer. They have a clear set of results that seem correct/rigorous, and the practical implications included in the rebuttal are interesting. I'm bumping from 5 --> 7 (correction: 8, since 7 isn't an option in this weird, non-continuous score scaling that ICLR insists on). Let's accept and move on with our lives :)

---

> > ### Author Response · Authors · 2024-11-29
> >
> > We are very glad to hear that our responses and revisions have met your satisfaction.
> >
> > We agree that the Lipschitz assumption remains a bit distant from some practical scenarios and is primarily included to ensure theoretical feasibility.
> >
> > Thank you once again for your constructive feedback, attention to detail, and encouraging kind words. Your review, particularly your suggestions regarding presentation and organization, has made this work more robust and accessible to a broader audience.

---

### Official Review · Reviewer_VLhX · 2024-11-03

**Soundness:** 3
**Presentation:** 2
**Contribution:** 2
**Rating:** 6
**Confidence:** 3

**Summary:**

This paper studies the theoretical properties of 1-layer 1-head transformers in terms of their approximation power and also the existence of efficient algorithms to approximate arbitrary 1-layer 1-head transformers with weights of bounded norms. There are 3 main theoretical results in the paper. The first result (theorem 2.1) shows that a single self-attention layer with sufficiently large number of MLP layers can approximate any Lipschitz function that maps one sequence of real-value vectors to another sequence of real-valued vectors. The second result (theorem 2.3) shows that a 1-layer 1-head transformers with sufficiently wide 2-layer MLPs can memorize any dataset generated by Lipschitz functions with prompt tuning with sufficiently long prompt. Finally, the third result (Theorem 3.1) shows that the Prompt Tuning Inference problem can be approximated efficiently in nearly linear time (in terms of the sequence length) through a reduction to a previously known result (AttC).

**Strengths:**

The paper is technically strong. It provides several improvements over previous results on the topic.
While I did not verify every single proof, the overall argument seems sound and the conclusion plausible.
The paper is fairly well-written for such a technically involved paper, and I believe the results in this paper will be useful for future work studying the approximation properties of transformers.

**Weaknesses:**

Overall, while the paper is technically solid, I have a hard time seeing it being impactful for communities beyond those studying the representational power of transformers.

I consider myself to be reasonably well-informed about the development of deep learning theory. These kinds of approximation results are certainly nice to know, but they rarely make any impact in practice or even inform any algorithmic design. For example, the authors claim that:

> These fundamental limits provide important necessary conditions for designing expressive and efficient prompt tuning methods for practitioners.

I do not see how these results will have any impact on how we do prompt tuning right now. What does it tell us that we do not already know? If there are, I would like to see them discussed in more detail in the paper. For example, does this result says about why prompt tuning fails in certain scenarios?

I would also like to see related work in the main body of the paper and a more detailed discussion of how this result relates to prior works. The page limit has been raised to 10 so there is no reason to leave them in the appendix.

**Questions:**

> We study the fundamental limits of prompt tuning transformer-based pre-trained models (i.e., foundation models) in two aspects

Where does the pretraining affect this analysis? As far as I can tell, the paper only considers the representational power of the transformer. Why does it matter if the transformer is pre-trained or not? If pretraining matters in the analysis, I would expect to see some assumptions about the representation from pretraining but I am not sure if there is any.

---

> ### Author Response · Authors · 2024-11-19
> **Response 1**
>
> ### Thank you for your detailed review.
> ### We have addressed all your comments and questions in this and the following responses.
> ### We have also modified the draft accordingly and marked the changes in **blue** in the latest revision.
> ### Please see the updated PDF for details.
>
> ---
>
> > **Reviewer's Comment:**
> I do not see how these results will have any impact on how we do prompt tuning right now. What does it tell us that we do not already know? If there are, I would like to see them discussed in more detail in the paper.
>
>
> Thank you for your question.
>
> In response to **practical impact, we have extended concluding remarks discussing the practical implications of our theoretical results.** We quote in below for your reference. Please also see `page 10` of the revised PDF for more details.
>
>
>
> ---
>
> ### **Practical Implications from Statistical Limits (Section 2).**
>
>  We analyze the universality of prompt tuning transformers with minimal structures, and its memorization capacity on general datasets.
>
> * **Universality (Theorem 2.2).** Our results show that the universality of prompt tuning pretrained transformer is achievable on as simple as a single-layer, single-head attention transformers. This demonstrates that universality in prompt-tuning isn’t limited to large, complex foundation models.
>
> * **Width-Depth Tradeoff (Section 2.4).** Our results highlight a trade-off in the design choices for the depth and width of FFN (MLP) layers:
>     1. $\mathcal{O}((1/\epsilon)^{d(L+L_p)}$ FFN layers of width $4$ or
>     2. $2$ FFN layers of width  $\mathcal{O}((1/\epsilon)^{d(L+L_p)}$.
>
>     In practice, 1. and 2. differ in memory usage, parallelization, and optimization preferences, leading to distinct application scenarios.
>
> * **Memorization (Section 2.5).** Our memorization results apply to general datasets, whereas prior results are limited to specialized cases. This makes our results go beyond specialized theoretical analysis and align more with practical applications with a suggested *long* soft-prompt length.
>
>
> ### **Practical Implications from Computational Limits (Section 3).**
>
> We analyze the $\mathcal{O}(L^2)$ bottleneck of prompt tuning transformers and provides useful guidance for designing efficient prompt tuning (approximation) methods with precision guarantees. Let $Q_p=W_QX_p$, $K_p=W_KX_p$, and $V_p=W_VX_p$ with $X_p=[P,X]\in\mathbb{R}^{d\times (L_p+L)}$. Here $L$ and $L_p$ are the input and soft-prompt length.
>
> * **Self- and Cross-Attention.** Our computational results apply to both self-attention and cross-attention prompt tuning. This is because the norm bound conditions depend on $\max\{|Q_p|, |K_p|, |V_p|\}$, which are valid for both self- and cross-attention inputs.
>
> * **Necessary Conditions for Subquadratic Prompt Tuning (Theorem 3.1).** Our result suggests proper normalization on soft-prompt and weight matrices are required to ensure subquadratic prompt tuning inference, i.e.,  $\max\{\|Q_p\|, \|K_p\|,\|V_p\|\}\le \mathcal{O}(\sqrt{\log(L_p+L)})$.
>
> * **Necessary Conditions for Almost Linear Time Prompt Tuning (Theorem 3.2).** Our result suggests more strict normalization on soft-prompt and weight matrices are required to ensure almost linear time prompt tuning inference, i.e.,$\max\{\|Q_p\|, \|K_p\|,\|V_p\|\}\le o(\sqrt{\log(L_p+L)})$.
>
> ---
>
> > For example, does this result says about why prompt tuning fails in certain scenarios?
>
>
>
> In response to **failures of prompt tuning in practical scenarios**, we kindly invite reviewers to check Appendix I of the submitted draft. We also provide a summary below.
>
> **Summary of Appendix I.**
> Our “there exist” type of theoretical results show that, with prompt tuning, even a pretrained transformer with a minimal architecture can act as a universal approximator. However, **achieving this requires a *task-specific* pretrained model weights tailored precisely to the task.** In Appendix I, we discuss the limitations of prompt tuning on arbitrary pretrained transformer models.
>
> In addition, we provide 3 examples of scenarios for failures:
> 1. **Sec 2:** Our universality results imply soft-prompt length is exponentially dependent on input dimension $d$ and length $L$. This suggests that **short soft-prompts have very limited expressiveness for prompt tuning transformers.**
> 2. **Appendix I:** Our universality results also imply prompt tuning transformers with arbitrary pretrained weights have limited expressiveness.
> 3. **Sec 3:** Our computational results imply **proper normalization of soft-prompt+input prompt and pretrained attention weights is crucial for efficient designs.**

---

> ### Author Response · Authors · 2024-11-19
> **Response 2**
>
> > **Reviewer's Comment:** I would also like to see related work in the main body of the paper and a more detailed discussion of how this result relates to prior works. The page limit has been raised to 10 so there is no reason to leave them in the appendix.
>
> Thank you for the suggestion. However, after careful consideration, we decided to use page 10 for **discussions of practical implications** and **to adjust the layout for improved readability.** Therefore, we believe it is more appropriate to keep the related work in the appendix.
>
> We kindly invite reviewers to refer to Appendix B of the latest revision. The relevant discussions are quoted below for your reference.
>
> * [Wang et al., 2023a] proves the universality of prompt tuning a model that requires deep transformers of $\mathcal{O}((L_p+L) (1/\epsilon)^{d})$ attention layers with two heads and $\mathcal{O}((1/\epsilon)^{d (L_p+L)})$ FFN layers. In contrast, we prove that prompt tuning transformers with the simplest configurations — single-head, single-layer attention, 2 FFN layers — are universal approximators. In addition, in the memorization capacity aspect,  [Wang et al., 2023a] considers datasets with only two-token sequences and focus solely on memorizing the final token. Our work shows that prompt tuning simple transformers with 1-head, 1-layer attention and 2 FNN layers is capable of complete memorization of general datasets without any constraint on the number of tokens. Moreover, we establish a lower bound on the required soft-prompt tokens for any dataset.
>
> * [Petrov et al., 2024] prove the universality of prompt tuning on transformers, with the number of layers linear in the input sequence length. Furthermore, they focus on approximating functions with domain of hypersphere. On the other hand, we focus on transformers with the simplest configuration — single-layer-single-head self-attention — and a more general continuous sequence-to-sequence functions.
>
> * Other prior works: [Petrov et al., 2023] discuss different kinds of context-based learning, and experimentally show when prompt tuning is successful in adapting to new tasks. In this work, we tackle the prompt tuning problem from a theoretical perspective.  [Oymak et al., 2023] identifies the cases where attention layer with prompt tuning is more expressive than a self-attention layer. They utilize prompt tokens dependent to weight matrices. In addition, they require weight matrices to be full rank. Conversely, our study explores the expressive power of prompt tuning under more general conditions, without relying on such assumptions.
>
>
> > **Reviewer's Question:** “We study the fundamental limits of prompt tuning transformer-based pre-trained models (i.e., foundation models) in two aspects”
> Where does the pretraining affect this analysis? As far as I can tell, the paper only considers the representational power of the transformer. Why does it matter if the transformer is pre-trained or not? If pretraining matters in the analysis, I would expect to see some assumptions about the representation from pretraining but I am not sure if there is any.
>
>
> Sorry for the confusion caused.
>
> Our proof demonstrates the existence of a transformer capable of universal approximation, but achieving this requires specific model weights. Therefore, **proper pretraining** to obtain these specific weights is essential to realize the universality of prompt tuning.
>
> ---
>
> Thank you for your thorough review!
>
> We're open to any further questions or clarifications you might have about our work.

---

> ### Author Response · Authors · 2024-11-24
> **A Gentle Reminder**
>
> Dear Reviewer,
>
> As the rebuttal phase nears its end, we wanted to gently remind you of our responses.
>
> We have addressed all your comments and questions with the utmost care and hope our efforts meet your expectations.
>
> If you have any remaining concerns, we would be happy to discuss them further. Otherwise, we kindly invite you to consider raising your score if you find our updates satisfactory.
>
> Thank you for your time and thoughtful review!

---

### Official Review · Reviewer_BCYB · 2024-11-03

**Soundness:** 3
**Presentation:** 3
**Contribution:** 3
**Rating:** 6
**Confidence:** 2

**Summary:**

This paper studies the fundamental limits of prompt tuning and derives several interesting theoretical properties on single-head transformers with a single self-attention layer. The authors 1) show that prompt tuning can universally approximate any sequence-to-sequence function, subject to some continuity assumptions,  2) show that prompt tuning can memorize arbitrary datasets and 3) prove the existence of almost-linear time prompt tuning algorithms, which builds the theoretical foundation on the computational time required for prompt tuning.

**Strengths:**

Overall, given the theoretical nature of the paper, it is a dense read by design, but the good presentation has helped a lot by clearly outlining the different contributions (e.g., universality results, memorization properties and the computational time analysis). The paper is quite solid in terms of the theoretical contributions. The contributions are outlined cleanly and the flow is great. The theoretical results also represent a significant contribution over existing works, such as [1].

References

[1] Yihan Wang, Jatin Chauhan, Wei Wang, and Cho-Jui Hsieh. Universality and limitations of prompt tuning. Advances in Neural Information Processing Systems (NeurIPS), 36, 2023a.

**Weaknesses:**

To begin with my feedback, I'd like to mention that while I have a grasp about the high-level ideas and contributions of the paper, I have not checked the math and derivations in detail although they seem reasonable; I am not an expert in this area, so I might not have sufficient knowledge in judging the merits and significance of the work w.r.t. the previous works, so I defer the assessment of these parts to other reviewers and the AC.

I understand the paper outlines many "there exists" type of results which cannot be verified easily with practical algorithms, however, I do think the authors could give add some discussions on in what circumstances their theoretical results could lead to practical implications, especially given that prompt tuning itself is a very *practical* low-cost adaptation technique, so for the results to be relevant and of interest beyond the theoretical community, some kind of discussions about practical impact are essential. There also seems to be some contradictions given the paper suggests that even a shallow, single-head transformer is already universal yet there is some empirical evidence that prompt tuning is more effective with deeper transformers -- while I understand some gap between theory and practice can be acceptable especially given that many theoretical properties of prompt tuning are not well-known thus far, I wonder whether the authors could give any comments on why that is the case? Is it possible that there are unrealistic assumptions in the theoretical analysis?

**Questions:**

Please refer to "Weaknesses" above.

---

> ### Author Response · Authors · 2024-11-20
> **Response 1**
>
> ### Thank you for your detailed review.
> ### We have addressed all your comments and questions in this and the following responses.
> ### We have also modified the draft accordingly and marked the changes in **blue** in the latest revision.
> ### Please see the updated PDF for details.
>
> ---
>
> > **Reviewer's Comment:** I understand the paper outlines many "there exists" type of results which cannot be verified easily with practical algorithms, however, I do think the authors could give add some discussions on in what circumstances their theoretical results could lead to practical implications, especially given that prompt tuning itself is a very practical low-cost adaptation technique, so for the results to be relevant and of interest beyond the theoretical community, some kind of discussions about practical impact are essential.
>
>
> Thank you for your suggestion.
>
> **We have added extended discussions about practical implications of our theoretical results in our concluding remarks (Sec 4).**
> Please also see **`page 10`** of the revised PDF for more details. We quote the below for your reference.
>
> ---
>
> ### **Practical Implications from Statistical Limits (Section 2).**
>
>  We analyze the universality of prompt tuning transformers with minimal structures, and its memorization capacity on general datasets.
>
> * **Universality (Theorem 2.2).** Our results show that the universality of prompt tuning pretrained transformer is achievable on as simple as a single-layer, single-head attention transformers. This demonstrates that universality in prompt-tuning isn’t limited to large, complex foundation models.
>
> * **Width-Depth Tradeoff (Section 2.4).** Our results highlight a trade-off in the design choices for the depth and width of FFN (MLP) layers:
>     1. $\mathcal{O}((1/\epsilon)^{d(L+L_p)}$ FFN layers of width $4$ or
>     2. $2$ FFN layers of width  $\mathcal{O}((1/\epsilon)^{d(L+L_p)}$.
>
>     In practice, 1. and 2. differ in memory usage, parallelization, and optimization preferences, leading to distinct application scenarios.
>
> * **Memorization (Section 2.5).** Our memorization results apply to general datasets, whereas prior results are limited to specialized cases. This makes our results go beyond specialized theoretical analysis and align more with practical applications with a suggested *long* soft-prompt length.
>
>
> ### **Practical Implications from Computational Limits (Section 3).**
>
> We analyze the $\mathcal{O}(L^2)$ bottleneck of prompt tuning transformers and provides useful guidance for designing efficient prompt tuning (approximation) methods with precision guarantees. Let $Q_p=W_QX_p$, $K_p=W_KX_p$, and $V_p=W_VX_p$ with $X_p=[P,X]\in\mathbb{R}^{d\times (L_p+L)}$. Here $L$ and $L_p$ are the input and soft-prompt length.
>
> * **Self- and Cross-Attention.** Our computational results apply to both self-attention and cross-attention prompt tuning. This is because the norm bound conditions depend on $\max\{|Q_p|, |K_p|, |V_p|\}$, which are valid for both self- and cross-attention inputs.
>
> * **Necessary Conditions for Subquadratic Prompt Tuning (Theorem 3.1).** Our result suggests proper normalization on soft-prompt and weight matrices are required to ensure subquadratic prompt tuning inference, i.e.,  $\max\{\|Q_p\|, \|K_p\|,\|V_p\|\}\le \mathcal{O}(\sqrt{\log(L_p+L)})$.
>
> * **Necessary Conditions for Almost Linear Time Prompt Tuning (Theorem 3.2).** Our result suggests more strict normalization on soft-prompt and weight matrices are required to ensure almost linear time prompt tuning inference, i.e.,$\max\{\|Q_p\|, \|K_p\|,\|V_p\|\}\le o(\sqrt{\log(L_p+L)})$.

---

> ### Author Response · Authors · 2024-11-20
> **Response 2**
>
> > **Reviewer's Comment:**  There also seems to be some contradictions given the paper suggests that even a shallow, single-head transformer is already universal yet there is some empirical evidence that prompt tuning is more effective with deeper transformers -- while I understand some gap between theory and practice can be acceptable especially given that many theoretical properties of prompt tuning are not well-known thus far, I wonder whether the authors could give any comments on why that is the case? Is it possible that there are unrealistic assumptions in the theoretical analysis?
>
> Thanks for the comment. To clarify, **there is no contradiction.**
>
> While it’s true that larger and deeper pretrained transformers perform better in prompt tuning, our work focuses on establishing the theoretical limits of "prompt tuning transformers." We demonstrate that even simple, shallow transformer blocks can approximate any sequence-to-sequence Lipschitz function through prompt tuning. This suggests that **the universality of prompt tuning is inherent to the transformer block itself, rather than dependent on deep layers.** This insight aligns with the empirical success of prompt tuning discussed in the introduction.
>
> Specifically, as detailed in `Appendix I` (Limitations of Prompt Tuning Transformers), achieving universality with shallow transformers requires task-specific pretrained model weights. **This indirectly explains why deeper transformers excel in practice: their layers can accommodate more "task-specific weights."**
>
> Regarding the **Lipschitz continuous assumption, it is both reasonable and practical.** Real-world transformer inputs, such as vector embeddings, generally exhibit continuity in Euclidean space, making this assumption suitable for theoretical analysis and providing meaningful bounds on approximation and convergence rates.
>
> ---
>
> Thank you for your detailed review. We hope the responses above have addressed your concerns. We are open to any further discussions!

---

> ### Author Response · Authors · 2024-11-24
> **A Gentle Reminder**
>
> Dear Reviewer,
>
> As the rebuttal phase nears its end, we wanted to gently remind you of our responses.
>
> We have addressed all your comments and questions with the utmost care and hope our efforts meet your expectations.
>
> If you have any remaining concerns, we would be happy to discuss them further. Otherwise, we kindly invite you to consider raising your score if you find our updates satisfactory.
>
> Thank you for your time and thoughtful review!

---

> > ### Comment · Reviewer_BCYB · 2024-11-25
> >
> > I thank the authors for their feedback. I think the paper represents solid theoretical advances and can be of interest to the ICLR community and thus I'll keep my recommendation to accept. While I appreciate the new content about practical applications in Page 10, some of the findings are either somewhat vague and require more controlled experiments to show empirically (e.g., "suggesting "long-prompt" length -- where do you draw the line of "long"? If there is some theoretically suggested value, I also think some experiments will be needed to validate its optimality) or (unsurprising) results that we already know (e.g., "short soft embeddings are not very expressive"). While I do *not* think these points made the paper less valuable, I feel less confident about strongly advocating for this acceptance especially as I mentioned in the original review that I am not an expert in judging its theoretical value. I will keep my score.

---

> > > ### Author Response · Authors · 2024-11-25
> > > **Thank You and Reference to Proof-of-Concept Experiments in Previous Responses**
> > >
> > > Dear Reviewer BCYB,
> > >
> > > Thank you for your comment. We are glad to hear that you appreciate the solid theoretical contributions of this work.
> > >
> > > In response to the **control experiments** you mentioned:
> > >
> > > > "suggesting 'long-prompt' length -- where do you draw the line of 'long'? If there is some theoretically suggested value, I also think some experiments will be needed to validate its optimality) or (unsurprising) results that we already know (e.g., 'short soft embeddings are not very expressive')."
> > >
> > > By **long soft-prompt length**, we refer to the exponential lower bound of the **required soft-prompt length** to memorize data of dimension $d$ and length $L$. Specifically, we mean:
> > >
> > > $$
> > > L_p \geq L \cdot \left( 2(1/\epsilon)C(dL)^{1/\alpha} \right)^{dL},
> > > $$
> > >
> > > as suggested in **`Theorem 2.5`**.
> > >
> > > We agree that while it is unsurprising that a longer soft-prompt length is more expressive, control experiments are valuable for validating our results.
> > >
> > > In response, **we have included proof-of-concept experiments in **[this response](https://openreview.net/forum?id=jDpdQPMosW&noteId=2KMUPDoqX4)**.** These experiments verify the exponential lower bound of the required soft-prompt length using two Lipschitz datasets.
> > >
> > > Key results include:
> > > 1. We confirm the linear relationship between $ \log L_p $ and $\log \epsilon$. This relation is derived from the proposed soft-prompt length lower bound.
> > > 2. Prompt tuning on a single-head, single-layer transformer effectively approximates Lipschitz functions.
> > > 3. As $dL $ increases, the required prompt length $ L_p $ also increases, consistent with our theoretical results.
> > >
> > > **To summarize, our results, detailed in the figure [here](https://imgur.com/a/PbnlIdX), align well with our theoretical analysis.**
> > >
> > > We hope these experimental justifications address your concerns. If so, we kindly invite you to advocate for this work more strongly and consider raising your score to acceptance.
> > >
> > > Thank you for your thoughtful feedback!

---

> > > > ### Author Response · Authors · 2024-11-29
> > > >
> > > > Dear Reviewer BCYB,
> > > >
> > > > It seems our discussion was paused midway. We were wondering if you’ve had a chance to review our response to your previous comments.
> > > >
> > > > Specifically, regarding your question, “Where do you draw the line of 'long'?”, we refer to the **required soft-prompt length, $L_p$, for prompt tuning memorization**, provided in Thm 2.5 of the submitted draft.
> > > >
> > > > Moreover, our last response (quoted in the newly added Appendix K) includes **controlled experiments** to validate our results.
> > > >
> > > > We look forward to continuing the discussion. Thank you!

---

### Official Review · Reviewer_CLcT · 2024-11-04

**Soundness:** 2
**Presentation:** 1
**Contribution:** 2
**Rating:** 5
**Confidence:** 3

**Summary:**

The paper investigates the limits of prompt tuning in transformer-based models by focusing on the simplest architectures with a single self-attention layer and head. The authors prove that such simple transformers can universally approximate any Lipschitz continuous sequence-to-sequence function through prompt tuning. They also demonstrate that these transformers can memorize any dataset via prompt tuning by establishing an exponential lower bound on the required number of soft-prompt tokens based on data dimension, sequence length, and desired error tolerance. Additionally, they identify a computational efficiency phase transition determined by the norms of the soft-prompt-induced keys and queries. They establish an efficiency threshold beyond which no sub-quadratic algorithm exists under the SETH, but within this threshold, they prove the existence of almost linear-time inference algorithms for prompt tuning.

**Strengths:**

1. The paper advances the theory of prompt tuning by proving that even the simplest transformer architectures—with a single attention head and a single self-attention layer—can universally approximate any Lipschitz continuous sequence-to-sequence function.

2. The paper notably discovers a phase transition in computational efficiency based on the norms of the soft-prompt-induced keys and queries. It provides a criterion under which prompt tuning can be performed in sub-quadratic time and demonstrates the existence of almost linear time algorithms under certain conditions.

3. The paper introduces new applications, such as the chained reduction of piece-wise constant approximations, to prove universality results. It also extends the contextual mapping property to any-rank weight matrices in single-layer attention mechanisms, improving the rank-1 limitation of previous work.

**Weaknesses:**

1. The paper contains numerous grammatical errors, and some sentences are poorly constructed. A thorough proofreading is required to enhance the readability of the text.

2. While the paper brings new applications of previous concepts to function approximation, the technical novelty appears limited, with many proofs in Section 2 overlapping significantly with those used in "Are Transformers with One Layer Self-Attention Using Low-Rank Weight Matrices Universal Approximators?" and Universality and Limitations of Prompt Tuning" papers.

3. The paper does not include experiments to showcase any of the theoretical results. The authors could demonstrate the approximation capabilities of single-head, single-layer transformers for sequence-to-sequence functions in Section 2. The authors can even use synthetic datasets designed to have known Lipschitz properties. Deeper transformers with more heads and layers might be used for comparison.

4. Another empirical comparison might be examining the relationship between the number of soft-prompt tokens and the model's ability to memorize datasets, as suggested by the exponential lower bound. Again, one might utilize synthetic datasets where the complexity (e.g., vocabulary size, sequence length) can be controlled.

5. I think the authors should perhaps discuss and more on the results and analysis of the “Are Transformers with One Layer Self-Attention Using Low-Rank Weight Matrices Universal Approximators?” and compare the results as they prove that transformers with one self-attention layer are universal approximators for permutation equivariant continuous functions. Also, explaining how the results in the current paper relate to or differ from theirs might provide insights.

**Questions:**

1. The paper "Are Transformers with One Layer Self-Attention Using Low-Rank Weight Matrices Universal Approximators?" stated that they easily extended their analysis of contextual mapping to masked self-attention, which is what's actually used in practice. Since such an extension might be beneficial for completeness, could the authors elaborate on why they haven't addressed it? Was it more challenging in this case?

2. The identified efficiency criterion (Theorem 3.1) requires controlling the norms of the soft-prompt-induced keys and queries to be $O(\log(L_p + L))$. In practical applications, particularly with large-scale models and datasets, how feasible is it to meet this criterion?

**Minor Remarks**

1. In order to make them multiplication-compatible, $W_O^i$ should be $d \times s$, and this doesn’t propagate to later sections.

2. The infinity norm is already defined in matrices, which gives the maximum row sum, and perhaps a better notation would be $||M||_{max}$ for clarity.

---

> ### Author Response · Authors · 2024-11-22
> **Response 1**
>
> ### Thank you for your detailed review.
> ### Except for the experiments, we have addressed all your comments and questions in this and the following responses.
> ### **Numerical validation will be provided in a separate response once completed.**
> ### We have also updated the draft, marking changes in blue in the latest revision.
> ### Please refer to the updated PDF for details.
>
> ---
> > **Reviewer's Comment:** The paper contains numerous grammatical errors, and some sentences are poorly constructed. A thorough proofreading is required to enhance the readability of the text.
>
>
> Thank you for careful proofreading. In response, we had conducted 3 more rounds of proofreading and corrected all identified typos and grammatical errors, as spotted by both the reviewers and the authors.
>
> > **Reviewer's Comment:**  While the paper brings new applications of previous concepts to function approximation, the technical novelty appears limited, with many proofs in Section 2 overlapping significantly with those used in "Are Transformers with One Layer Self-Attention Using Low-Rank Weight Matrices Universal Approximators?" and Universality and Limitations of Prompt Tuning" papers.
>
> Thank you for your feedback.
>
> In response to the **concern of credit, the reason for the mentioned overlap is for completeness.**
>
> In the submitted version,  we had emphasized in the opening of Appendix C that **we need to introduce non-original but still necessary auxiliary lemmas to facilitate later proofs.**
> Moreover, we had intentionally arranged these non-original proofs in Appendix J (Supplementary Proofs for Appendix C.)
>
> In response to **novelty**, while our proof builds upon prior research, we would like to highlight several distinct contributions of our work:
>
> 1. **More General Attention Weights:** While the attention contextual mapping of [Kajitsuka and Sato, 2024a] holds only for rank-1 weight matrices, our results hold for any rank. This makes our results more practical. That is, it’s almost impossible to find attention layers with rank-1 matrices in any pre-trained transformer. Please see **`Remark 2.1`** in `line 286-291` for detailed discussion.
>
> 2. **Simplest Possible Transformer:** Our research is the first to formally demonstrate the universality of prompt tuning in transformers using the simplest possible configuration with single layer single head attention, while  [Wang et al., 2023a] requires $\mathcal{O}(L (1/ \delta)^d)$ attention layers to achieve $\delta$-accuracy. This result suggests that prompt tuning universality is not a consequence of deep layers.
>
> 3. **More General Memorization Results:** Our memorization results hold for general datasets, while [Wang et al., 2023a] only discuss memorizing the last token of some very specialized datasets.
>
> We emphasize that the above results cannot be trivially deduced from [Kajitsuka and Sato, 2024a; Wang et al., 2023a]. We believe these advancements represent a non-trivial step beyond incremental improvements, making a fair contribution to the field.
>
> > **Reviewer's Comment:** I think the authors should perhaps discuss and more on the results and analysis of the “Are Transformers with One Layer Self-Attention Using Low-Rank Weight Matrices Universal Approximators?” and compare the results as they prove that transformers with one self-attention layer are universal approximators for permutation equivariant continuous functions. Also, explaining how the results in the current paper relate to or differ from theirs might provide insights.
>
> Sorry for the confusion caused, but we believe there might be some slight oversight.
>
> **This paper is about universality of prompt tuning transformers, while [Kajitsuka and Sato, 2024a] is about universality of transformers.** The primary difference is that the transformer in our setting is fixed. We show there exists a soft-prompt $P$ capable of approximating any seq2seq Lipschitz continuous function with precision guarantee.
>
> We kindly invite reviewers to check **`Appendix A`** for discussion on universality of transformers.
>
> We provide a summary here for your reference:
>
> >> The previous paper “Are Transformers with One Layer Self-Attention Using Low-Rank Weight Matrices Universal Approximators?” explores the universality of transformers, illustrating that achieving universal approximation requires weight updates. In contrast, our findings demonstrate the universality of prompt tuning. Specifically, our results show that by adjusting only the prompts of a pretrained model—without any weight updates—a transformer with the simplest architecture achieves universal approximation. This shows the potential of prompt tuning as a powerful, efficient alternative to traditional weight-updating based tuning.

---

> ### Author Response · Authors · 2024-11-22
> **Response 2 (Questions)**
>
> > **Reviewer's Question:** The paper "Are Transformers with One Layer Self-Attention Using Low-Rank Weight Matrices Universal Approximators?" stated that they easily extended their analysis of contextual mapping to masked self-attention, which is what's actually used in practice. Since such an extension might be beneficial for completeness, could the authors elaborate on why they haven't addressed it? Was it more challenging in this case?
>
> Thank you for the question.
>
> **The application of masked self-attention is also straightforward in this work.** Since transformers with masked self-attention mechanisms also exhibit universality, we can adopt the same approach outlined in our paper:
>
> 1. Quantize the target function class and prompts,
> 2. Construct a surrogate function to correspond the prompts to each target function.
> 3. Utilize the universality of masked self-attention transformers to approximate the surrogate function.
>
>
> > **Reviewer's Question:**  The identified efficiency criterion (Theorem 3.1) requires controlling the norms of the soft-prompt-induced keys and queries to be $\mathcal{O}{\sqrt{\log (L_p+L)}}$. In practical applications, particularly with large-scale models and datasets, how feasible is it to meet this criterion?
>
> Thanks for the question.
>
> In response, suitable normalizations for these norms can be implemented using pre-activation layer normalization [1,2] to control $\|X_p\|$, or outlier-free attention activation functions [3] to control pretrained weights $\|W_K\|, \|W_Q\|, \|W_V\|$.
>
> We acknowledge the submitted draft was too vague about practical implications. In response, **we have extended our conclusion with 2 more concluding remarks on our theoretical results.** Please see **`page 10`** of the latest revision for details.
>
> [1] Xiong  et al. "On layer normalization in the transformer architecture." International Conference on Machine Learning. PMLR, 2020.
>
> [2] Wang, Qiang, et al. "Learning deep transformer models for machine translation." ALC 2019
>
> [3] Hu et al. "Outlier-efficient hopfield layers for large transformer-based models." ICML 2024
>
> > **Reviewer's Question:**  In order to make them multiplication-compatible, $W_O^i$ should be $d\times s$, and this doesn’t propagate to later sections.
>
>
> Thank you for taking close attention. This is a typo. We have corrected it and make sure the notation is consistent throughout the paper.
>
>
> > **Reviewer's Question:**   The infinity norm is already defined in matrices, which gives the maximum row sum, and perhaps a better notation would be $\|...\|_{max}$ for clarity.
>
> Yes you are absolutely correct. Sorry for confusing notation. **We have changed it to the standard  $\|...\|_{max}$ notation as you suggested.**
>
> ---
>
> Thank you for your detailed review and careful proofreading. We hope the above responses have adequately addressed your concerns.
>
> We are looking forward to further discussions!

---

> ### Author Response · Authors · 2024-11-23
> **Response 3: Proof-of-Concept Experiments**
>
> > **Reviewer's Comment:** The authors could demonstrate the approximation capabilities of single-head, single-layer transformers for sequence-to-sequence functions in Section 2. The authors can even use synthetic datasets designed to have known Lipschitz properties.
>
> > **Reviewer's Comment:** Another empirical comparison might be examining the relationship between the number of soft-prompt tokens and the model's ability to memorize datasets, as suggested by the exponential lower bound.
>
> # Proof-of-Concept Experiments
>
> Here we provide minimally sufficient numerical results to back up our theory.
>
> ### **Figure: $\log L_p$ vs $\log \epsilon$ Plots for Different Data Types.**
> https://imgur.com/a/PbnlIdX
>
> **Caption.** The numerical results align with prompt tuning universality (`Thm 2.3`) and memorization (`Thm 2.5`) results.
> We verify that prompt tuning on a single-head, single-layer transformer can approximate Lipschitz functions.  For Lipschitz data of dimension $d$ and length $L$, we observe that as $dL$ increases, the required prompt length $L_p$ also increases.  In particular, we confirm the lower bound for the soft prompt: $\log L_p \propto -\log \epsilon$.
>
> ### **Objective: Memorization of Prompt Tuning on Single-Layer Single-Head Attention Transformer**
>
> We verify the required soft-prompt length of prompt tuning memory capacity (`Thm 2.5`):
>
> $$
> L_p \ge L \cdot \left( 2(1/\epsilon)C(dL)^{1/\alpha} \right)^{dL},
> $$
>
> where $\epsilon$ is the maximum error in retrieving a sequential data point with Lipschitz constant $C$, length $L$, and dimension $d$. For simplicity, we verify the linear relation:
>
> $$
> \log L_p \propto - \log \epsilon.
> $$
>
> Besides the memorization result in `Thm 2.5`, this setting also illustrates two additional points:
>
> 1. **Verifying the Universality Results (`Thm 2.3, 2.4`)**:
>    In this setting, the target function of prompt tuning approximation is the identity function mapping $C$-Lipschitz data to themselves.
>
> 2. **Verifying the Contextual Mapping Results (`Lemma 2.2`)**:
>    In the proof of `Thm 2.5`, we utilize the concept of contextual mapping to determine the required soft-prompt length. Thus, verifying the above relation also verifies `Lemma F.2` and `Lemma 2.2`.
>
> ---
>
> ### **Setup**
> We perform prompt tuning on a single-head, single-layer transformer with a hidden size of 1, following `Sec. 2`,
> Memorization is defined as in `Def. 2.7`.
> We use this transformer model to demonstrate the memorization capacity of prompt tuning, as shown in `Thm 2.5`.
> We verify the linear relation for different $(d, L)$ values: $(d=1, L=2)$, $(d=1, L=3)$, and $(d=2, L=3)$.
>
> ---
>
> ### **Data**
> We generate Lipschitz sequential data $X \in \mathbb{R}^{d \times L}$ using:
>
> - The sigmoid function on the interval $[0,1]$ with dimension $d$ and length $L$.
> - The $x^2$ function on the interval $[0,1]$ with dimension $d$ and length $L$.
>
> ---
>
> ### **Optimizer**
> We use the Adam optimizer to optimize the prompt $P \in \mathbb{R}^{d \times L_p}$ while keeping the transformer model weights fixed.
> Training continues until the maximum error $\epsilon$ decreases by no more than 0.00001 for 10 consecutive $L_p$ epochs.
>
> ---
>
> ### **Computational Resources**
> All experiments are conducted using a single NVIDIA A100 GPU with 80GB of memory.
> The code is based on standard PyTorch and the Hugging Face Transformers library.
>
> ---
>
> ### **Results: Alignment with Theory (i.e., the suggested soft-prompt length lower bound or $\log L_p \propto - \log \epsilon.$)**
> Our results are presented in figure: https://imgur.com/a/PbnlIdX
>
> Key observations include:
>
> 1. We confirm the linear relationship between $\log L_p$ and $\log \epsilon$.
> 2. Prompt tuning on a single-head, single-layer transformer approximates Lipschitz functions.
> 3. We verify that with a larger $dL$, the required prompt length $L_p$ increases.
>
> ---
>
> We hope these responses and revisions address the reviewers' concerns and improve the overall quality of our paper.
>
> Thank you again for your review! We are looking forward to further discussions!

---

> ### Author Response · Authors · 2024-11-24
> **A Gentle Reminder**
>
> Dear Reviewer,
>
> As the rebuttal phase nears its end, we wanted to gently remind you of our responses.
>
> We have addressed all your comments and questions with the utmost care and hope our efforts meet your expectations.
>
> If you have any remaining concerns, we would be happy to discuss them further. Otherwise, we kindly invite you to consider raising your score if you find our updates satisfactory.
>
> Thank you for your time and thoughtful review!

---

> > ### Comment · Reviewer_CLcT · 2024-11-27
> >
> > I thank the authors for efforts during rebuttal. The authors have addressed some of my concerns such as where to locate this work among the previous efforts on universality of transformers and prompt tuning. Yet, I still have the following questions:
> >
> > *  I want to ask the authors why they haven't put their experiments in the paper. I am aware of the the page limitations, but I believe it can be valuable to incorporate in Appendix to verify their theoretical results. Additionally, the figure that the authors demonstrate in this rebuttal does not quite reflect the linear expected behavior, particularly because the authors use only four data points. I suggest the authors to add more data points in the figure to show the linear behavior between $\log(L_p)$ and $-\log(\epsilon)$.
> >
> > * For the novelty, I understand that the works build upon the prior results and have some extension to them. Even though there are some contributions in this work, they are incremental and the novelty is limited as most of their theoretical results are highly inspired by the previous work. Could the authors highlight specific contributions and technical novelty that set this work apart by introducing a genuinely new idea or perspective?
> >
> > * The authors extend the previous effort on contextual mapping property of attention from rank-1 to any rank weight matrices. I understand that there is not rank-1 matrix in practical settings, but I could not understand the theoretical value of extension to any rank matrix. Isn't it obvious that if rank-1 matrix satisfy Lemma D.2, then it is easy to construct a matrix with any other ranks because rank-1 is more constrained compared to other ranks? Indeed, in the proof of Lemma D.2, the authors build upon the previous work by adding rank-1 matrices all of which already satisfy the condition. Actually, I am not sure if the proof of Lemma D.2 is correct if the set $\mathcal{Q}$ has only one element. Could the authors explain how to extend the matrix from rank-1 to rank-$\rho$ when the set $\mathcal{Q}$ has only one element?

---

> ### Author Response · Authors · 2024-11-28
>
> Thank you for your attention to detail and additional questions. Below are our clarifications:
>
>
> 1. **Regarding the case where the set $\mathcal{Q}$ has only one element:**
>    We confirm that **the inequality in Lemma D.2 holds in this scenario.** Specifically, if $\mathcal{Q}$ contains only one element $v$, we can construct the weight matrices $W_K$ and $W_Q$ by using $v$ as $q_0$ and $q'_0$.
>
> 2. **Regarding the novelty:**
> The novelty lies in how to characterize the Lipschitz seq2seq function space in the prompt tuning setting.
>
>    We acknowledge that our work (problem setting and techniques) builds on [Kajitsuka and Sato, 2024a; Wang23; Yun19]. However, we believe that appropriately attributing credit to their foundational contributions does not diminish the novelty of our work. As explained earlier, our results go beyond these prior works and cannot be trivially deduced from [Kajitsuka and Sato, 2024a; Wang et al., 2023a]. Here, we highlight an example of a genuinely novel idea/technique:
>     * our memorization analysis is purely novel (original) as [Wang23] only consider a very specialized 2-token sequence data setting. Our innovation is adopting a contextual mapping characterization to establish the correspondence between the soft-prompt $P \in \mathbb{R}^{d \times L_p}$ and the target Lipschitz seq2seq function space. This approach enables us to prove the generic memorization capacity and derive a lower bound on the required soft-prompt length. We deem this a genuinely new idea, perspective, and technique for handling prompt tuning memorization.
>
> 3. **Regarding the theoretical value of extending from rank-1 to higher-rank matrices:**
>    While rank-1 matrices represent a more constrained case, the theoretical value lies in rigorously proving that the contextual mapping property holds for any rank-$\rho$ matrix. This ensures that the inequality in Lemma D.2 is satisfied under the interactions of multiple rank-1 components. This proof is necessary as it explicitly demonstrates that the inequality holds for any rank-$\rho$, relaxing the rank-1 constraint and going beyond a straightforward assumption.
>
> 4. **Regarding including experiments in the draft:**
>    We initially omitted these experiments because we felt they were straightforward and incremental, contributing limited scientific value to the literature. However, based on your suggestion, we have now included these proof-of-concept experiments in `Appendix K` of the latest revision.
>
> 5. **Regarding the data points:**
>    We remind the reviewer that the plots are on a $\log_{10}$ scale. Specifically, by plotting 0, 1, 2, and 3, we have considered soft-prompt lengths ranging from $1$ to $10^3$ on 6 different settings (2 datasets * 3 configurations). We believe is sufficient for proof-of-concept experiments. Additionally, reproducible code has been provided in the supplementary material with our previous rebuttal response. Interested readers can easily add more data points at their convenience. For these reasons, we believe the current setting sufficiently demonstrates our points.
>
> Thank you again for your time and attention to detail. Please let us know if there is anything else we can clarify further.

---

> ### Comment · Reviewer_CLcT · 2024-11-28
>
> I do not agree with the authors on the construction of the rank$-\rho$ matrix.
>
> The authors claim that when there exists only one element $v \in \mathcal{Q}$, then they construct the matrix by $p_1 v^\top + p_2 v^\top = (p_1 + p_2) v^\top$, which is still rank-1 matrix. I suggest the authors to explain their construction in detail. I think the current construction is not correct.

---

> ### Author Response · Authors · 2024-11-29
>
> Thank you for your attention to detail, persistence for correctness and opportunity for further clarifications.
>
> Sorry for any confusion caused. We confirm that our construction is correct.
>
> We construct the weight matrices as
>
> $$
> W_K = \sum_{i=1}^\rho p_i q_i^\top ,\quad \text{and} \quad
> W_Q = \sum_{j=1}^\rho p^\prime_j q_j^{\prime \top}.
> $$
>
> If the set $\mathcal{Q}$ contains only one element $v$, we use $v$ as both $q_1$ and $q^\prime_1$. To construct rank-$\rho$ matrices $W_K$ and $W_Q$, we choose the remaining vectors $q_i$ and $q^\prime_i$ for $i > 1$ to be linearly independent of $q_1$ and $q^\prime_1$. This ensures that each additional term increases the rank of the matrices by one, resulting in full rank $\rho$.
>
> Furthermore, it suffices that at least one pair $(q_i, q^\prime_i)$ for some $i \in [\rho]$ belongs to $\mathcal{Q}$.
>
> **In plain language, our construction shows that the above sums of outer products can produce matrices of any rank, and it only requires a single pair $(q, q') \in \mathcal{Q}$ to validate our derivation.** Please refer to `line 1379-1383` for detailed derivations.
>
> We acknowledge that the current draft could better convey these details and will refine this explanation in the final version.
>
> Finally, we would like to confirm whether the other 4 concerns in your last response have been satisfactorily addressed. We are open to further discussion if needed.
>
> Thank you again for your time and constructive comments.

---

> ### Author Response · Authors · 2024-12-02
> **A Gentle Remider**
>
> Dear Reviewer CLcT,
>
> As the rebuttal phase nears its end and our discussion seems to have paused midway, we would be delighted to continue the conversation.
>
> We have made every effort to address all your concerns and additional questions in our responses.
>
> We would greatly appreciate any further input. If you feel your concerns have been adequately addressed, with utmost respect, we invite you to consider a score adjustment. If not, we hope to make the most of the remaining two days to provide further clarifications.
>
> Thank you for your time and consideration!
>
> Best,
>
> Authors

---

### Author Response · Authors · 2024-11-23
**Global Response**

Dear Reviewers,

We thank the reviewers for the insightful questions and reviews. We have answered all the questions and addressed all the comments in detail in rebuttal and revision.

In response to the reviewers' suggestions, we have made revisions to improve the overall readability of the paper. We have conducted 3 more rounds of proofreading and have fixed all typos. Several sections have been updated for clarity and completeness. These revisions include notation corrections, additional explanations, and paragraphs and sections to help readers build intuition.

**Most importantly, we extend the page 10 with concluding remarks about practical implications of our results.** [`BCYB`, `VLhX`, `8DjM`]

### **Revision Details**

* We unified the notation for the $l_p$-norm to $l_\alpha$-norm to avoid ambiguity with the soft-prompt length $L_p$. [`8DjM`]
* We refine the wording of many lemmas and theorems to ensure clarity and eliminate potential confusions. [`CLcT`, `BCYB`, `VLhX`, `8DjM`]
* We unified the notation of the input embeddings of the attention layer with $Z$. This is important as we need to distinguish input data $X$ and $Z$ in our analysis. [`8DjM`]
* In `line 66`, we provide the definition of our $\ell_\alpha$-norm. [`8DjM`]
* In `line 94`, we provide the definition for $Q_p, K_p, V_p$. [`8DjM`]
* In `line 246`, we provide clarification for the definition of vocabulary. [`8DjM`]
* In `line 311`, we provide definitions for our surrogate function $h$. [`8DjM`]
* In `line 452`, we provide an intuitive explanation for a better understanding of Theorem 3.1. [`CLcT`]

### **Summary of Contribution for Future Reader**

For the convenience of reviewers and future readers, we provide a high-level overview of our contributions here.

* **Universality of Prompt Tuning**
    * Analyze the universality of prompt tuning transformers with minimal structures, and its memorization capacity on general datasets.
    * First to simplify model architecture to a single-layer, single-head attention transformer.
    * Two designs design choices for the depth and width of FFN (MLP) layers.
    * First prompt tuning memorization of general datasets.

* **Prompt Tuning Efficiency**
    * Efficient prompt tuning is possible.
    * There exists an efficiency phase transition depending on the norm of soft-prompt-induced queries and keys.
    * Efficient prompt tuning can be as fast as almost linear time.


We hope these revisions address the reviewers' concerns and improve the overall quality of our paper.

Thank you again for your review! Please let us know if anything requires further clarification!

Best regards,

Authors

---

### Meta-Review · Area_Chair_zM9j · 2024-12-17

**Metareview:**

The paper studies the fundamental limit of prompt tuning for single-head transformer models with a single self-attention layer. The paper proves that prompt tuning for such transformer models is universal approximators. The paper provides a lower bound on the number of required soft prompts to memorize a dataset.

The reviewers generally agree that the paper makes important theoretical contributions to the community, is well presented (especially for such a theoretical paper), could be useful for future works along this line. Although the original submission contained some grammatical errors, the authors have made additional thorough proofreading during rebuttal.

A common concern shared by multiple reviewers is the practical implication of the theoretical contributions from the paper. During rebuttal, the authors provided some empirical results which indeed align with the theoretical results and also also added some extensive discussions on the practical implications of the theoretical results. Reviewer CLcT also asked about the technical novelty compared to most relevant prior work, and the authors gave thorough explanations in the rebuttal. Reviewer VLhX also asked a question about whether the theoretical results reflect the impact of pre-training, and the authors commented that the "there exists" type of results in the paper does need to assume that pre-training has found some reasonable weights. I think this explanation makes sense because accounting for the effect of pre-training can be an independent topic of its own and hence may be out of scope.

I think the discussions regarding the practical implications added by the authors make sense to me to some degree. But even without considering these, I think the theoretical contributions of the paper are also significant enough to warrant acceptance.

All things considered, acceptance is recommended.

**Additional Comments On Reviewer Discussion:**

During author-reviewer discussion, Reviewer CLcT raised a technical question during the rebuttal, and the authors gave multiple rounds of responses to explain it. The authors' explanation looks reasonable to me. Reviewer BCYB also expressed reservations about the significance of the practical implications added by the authors, but the reviewer also commented that this does not diminish the theoretical contributions of the paper.

---

### Decision · Program_Chairs · 2025-01-22

Accept (Poster)